human-computer interaction/computer modelling and simulation/artificial intelligence

forecasting, human judgment, quantitative forecasting methods, forecast combination

**Author for correspondence:**
Maximilian Zellner
e-mail: mzellner@usc.edu

# A survey of human judgement and quantitative forecasting methods

Maximilian Zellner[1], Ali E. Abbas[1], David V. Budescu[2] and Aram Galstyan[1]

[1]University of Southern California, Los Angeles, CA, USA
[2]Fordham University, Bronx, NY, USA

MZ, 0000-0002-4318-2663

This paper's top-level goal is to provide an overview of research conducted in the many academic domains concerned with forecasting. By providing a summary encompassing these domains, this survey connects them, establishing a common ground for future discussions. To this end, we survey literature on human judgement and quantitative forecasting as well as hybrid methods that involve both humans and algorithmic approaches. The survey starts with key search terms that identified more than 280 publications in the fields of computer science, operations research, risk analysis, decision science, psychology and forecasting. Results show an almost 10-fold increase in the application-focused forecasting literature between the 1990s and the current decade, with a clear rise of quantitative, data-driven forecasting models. Comparative studies of quantitative methods and human judgement show that (1) neither method is universally superior, and (2) the better method varies as a function of factors such as availability, quality, extent and format of data, suggesting that (3) the two approaches can complement each other to yield more accurate and resilient models. We also identify four research thrusts in the human/machine-forecasting literature: (i) the choice of the appropriate quantitative model, (ii) the nature of the interaction between quantitative models and human judgement, (iii) the training and incentivization of human forecasters, and (iv) the combination of multiple forecasts (both algorithmic and human) into one. This review surveys current research in all four areas and argues that future research in the field of human/machine forecasting needs to consider all of them when investigating predictive performance. We also address some of the ethical dilemmas that might arise due to the combination of quantitative models with human judgement.

# 1. Introduction

To predict an uncertain quantity or to determine its distribution, people (and organizations) often seek the advice of human experts and/or apply algorithmic procedures. The choice of algorithmic procedures and how to combine algorithmically derived forecasts with human expertise is contentious across academic disciplines. One hypothesis, mostly found in the operations research and computer science communities, is that with the wide availability of data and advances in computing technology, algorithmic forecasts offer the opportunity to support humans by mining large datasets and learning patterns and trends from data. Critics of this view point out that the use of machine learning or 'big data methods'—such as stepwise regression and neural nets—that use statistical procedures to discover apparent patterns without recourse in theory and prior knowledge are akin to alchemy (e.g. [1]). They emphasize that these procedures fare worse in forecasting competitions [2,3], and violate two important forecasting principles [4]: the Golden Rule of Forecasting, which reminds the forecaster to be conservative by being consistent with cumulative knowledge about the present and past and by seeking out all knowledge relevant to the problem [5], and Occam's razor [6].

This divide between fields is also evident when comparing the choice of terminology. Computer science, for example, refers to the use of human judges[1] and algorithmic methods in forecasting as human and machine forecasting. Researchers in the forecasting domain use the terms human judgement and quantitative/algorithmic methods instead. A similar observation can be made when investigating the creation of a joint forecast from multiple experts. Computer scientists label this step aggregation, while forecasting researchers refer to it as combination. Depending on the academic discipline, one can also find different perceptions of how individual forecasts should be aggregated or combined. These range from linear combinations to Bayesian updating of the decision maker's prior belief.

These disagreements between academic disciplines and traditions about what constitutes valid methods and the use of heterogeneous terminology motivate our survey of publications originating from the most common fields involved in forecasting. For the remainder of this survey, we adopt the terminology used in the forecasting domain. Human judgement refers to the derivation of a forecast by a human judge, and quantitative methods describe algorithmic or machine models. When discussing the creation of one joint forecast from many individual sources we use the term combination.

Several survey papers have previously reviewed the literature on human judgement. For example, Lawrence *et al.* [7] offers a comprehensive view of judgemental forecasting. Clemen & Winkler [8] also review a variety of methods to combine human judgement. Other literature focuses on comparing forecasts derived by human judgement and quantitative models. For example, Grove *et al.* [9]; Kuncel *et al.* [10]; Ægisdóttir *et al.* [11]; and Meehl [12] review findings comparing human and quantitatively derived predictions in clinical settings. They found that quantitative prediction methods outperform mental health practitioners, although prediction accuracy varied by several factors including the type of prediction, how and where predictor data were gathered, which statistical or algorithmic procedure was used, and how much information was available.

These studies delineate cases in which either human judgement or quantitative forecasting methods proved superior. Combining the two fields could mean that their inherent shortcomings balance each other out, thereby increasing forecasting accuracy and reliability. For this reason, we review both human judgement and quantitative forecasting methods as well as methods to combine their outputs. We evaluate research on human judgement, such as incentivization, scoring, calibration and group forecasting. The quantitative methods include common approaches such as regression models and smoothing of time series, and more advanced methods such as neural and Bayesian networks, ARIMA and simulations. At the intersection of human judgement and algorithmic forecasting, we discuss research issues such as algorithm aversion, belief updating and human trust in algorithmically derived forecasts. We also update and expand reviews of the combination of forecasts, which applies to both human judgement and quantitative methods, by considering more recent methods.

The objective of this review is threefold: (i) we survey literature on quantitative forecasting and human judgement and identify current and future trends. (ii) In the process, we revisit and update the previous literature reviews to include new literature in the field of forecast combination. (iii) The resulting high-level view of the field will provide the basis for a dialogue between fields that have been previously disconnected.

---

[1]We use the terms judge and forecaster interchangeably throughout the paper.

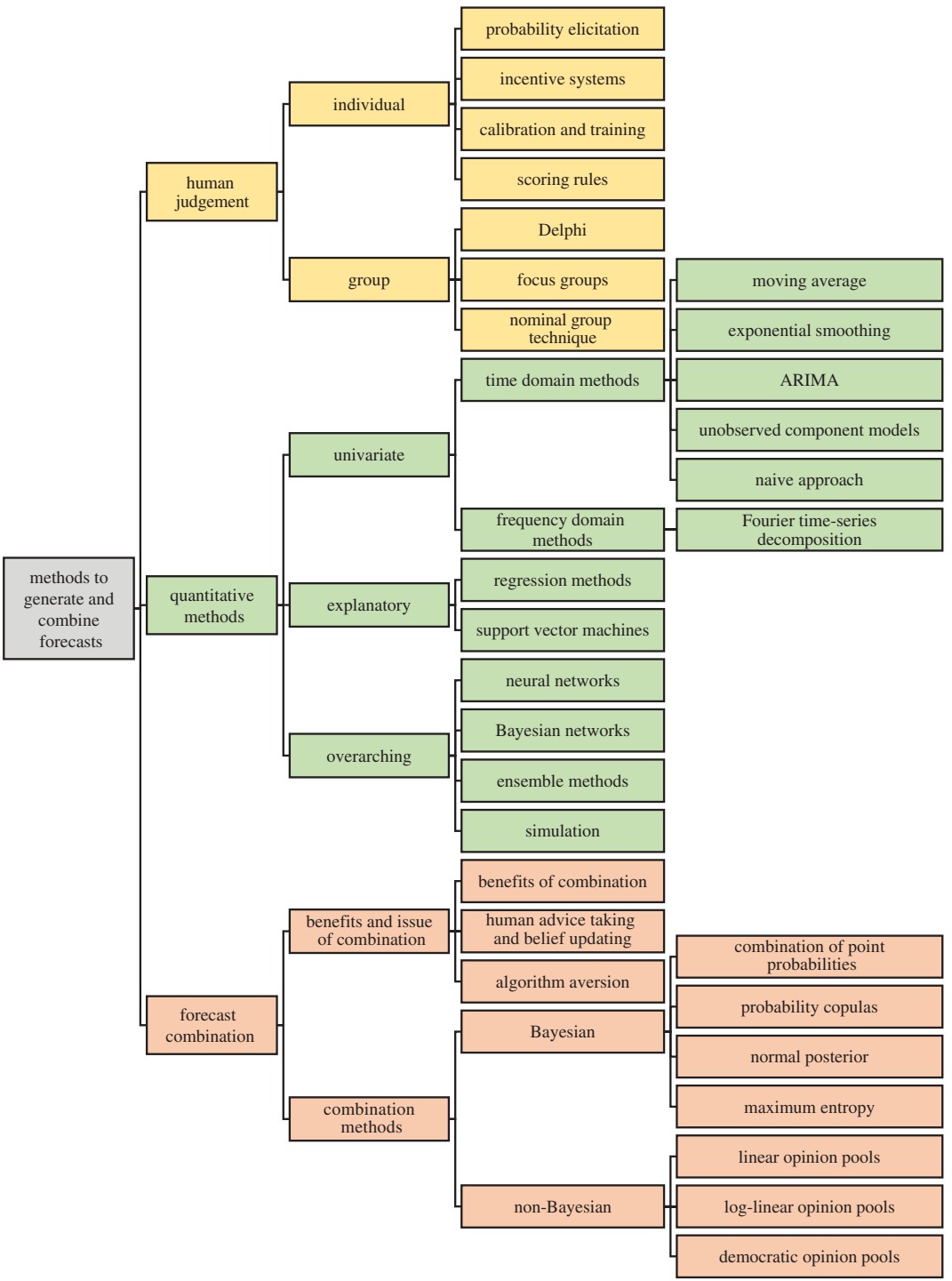

**Figure 1.** Structural overview of surveyed methods.

The entirety of methods and research problems in the forecasting domain are too numerous to be discussed in one survey paper. As a result, this paper addresses the most common research problems, as well as the most prominent techniques and methods, and groups them into three distinct areas. The first area concerns methods and problems relating to human judgement, whereas the second one discusses different quantitative methods to produce predictions. How to combine the forecasts produced by humans and quantitative methods constitutes the third area. Figure 1 provides a visual representation of the surveyed methods.

The remainder of this paper is structured as follows. Section 2 offers an overview of the research methodology we used to identify relevant studies. In §3, we review the literature on human judgement, and §4 investigates quantitative models for forecasting. Section 5 discusses issues arising

| definition of goal and key terms | forecasting fields and methods (§2.1) | determination of journals (§2.2) | review journals<br>human judgement (§3)<br>quantitative forecasting methods (§4)<br>forecast combination (§5) | summary and further research (§6) |
|---|---|---|---|---|

**Figure 2.** Methodology of literature review.

**Table 1.** Key search terms.

| | |
|---|---|
| — forecasting | — machine forecasting |
| — human forecasting | — artificial intelligence for prediction |
| — human judgement | — human computer interaction inforecasting |
| — forecasting using experts | — aggregation of expert judgement |
| — causal and time-series forecasting | — combination of expert judgement |
| — quantitative forecasting methods | |

from the combination of human judgement with predictions derived by quantitative models, as well as concrete procedures to combine multiple forecasters' judgements. Section 6 summarizes the results and our main conclusions.

# 2. Research methodology

Figure 2 summarizes the approach used to select literature for this review. First, we identified the main goal of the study, which was to review the literature on human judgement, quantitative forecasting models and their combination. The key terms to determine relevant literature are given in table 1.

We then performed a broad search of the main forecasting methods and the main application areas. These areas were used to conduct more in-depth searches, which helped identify relevant journals and publications. The journals, and referenced materials, were then searched and their findings condensed to derive main trends.

## 2.1. Forecasting fields and methods

Using the search term 'forecasting' to determine common forecasting application fields and methods, Google Scholar yielded approximately 2.7 million results when a search was conducted in June 2018. The forecasting fields were identified by conducting a search using the key term 'forecasting' and then using the results given by the 'related search' feature. In a second step, we took these results to conduct a search to determine the number of publications on each topic. Figure 3 shows that forecasting demand and weather are the fields most frequently covered by scientific publications, followed by electricity forecasting.

In the past 38 years, the number of publications in these areas increased more than 10-fold, from 111 400 publications in the period from 1980–1989 to 1 280 200 publications in the period 2010–2018.

Using the search term 'forecasting methods' and the 'related search' feature of Google Scholar, we identified the most prominent forecasting methods. Figure 4 shows the trend in publications concerning the most searched forecasting techniques. It shows that time-series forecasting has been the most prominent forecasting approach, followed by neural networks, and the specific time-series forecasting method ARIMA (short for autoregressive integrated moving average). Affective forecasting was also a prominent search term.

Figure 5 shows the share of publications in several subareas of forecasting. We distinguished quantitative forecasting methods along univariate and explanatory approaches, as well as models inherent to artificial intelligence. Artificial intelligence applications increased their share in publications, mostly at the expense of statistical forecasting methods. Over the four periods from 1980–1989 to 2010–2018, the total number of publications across the areas surveyed increased from 43 390 to 442 800.

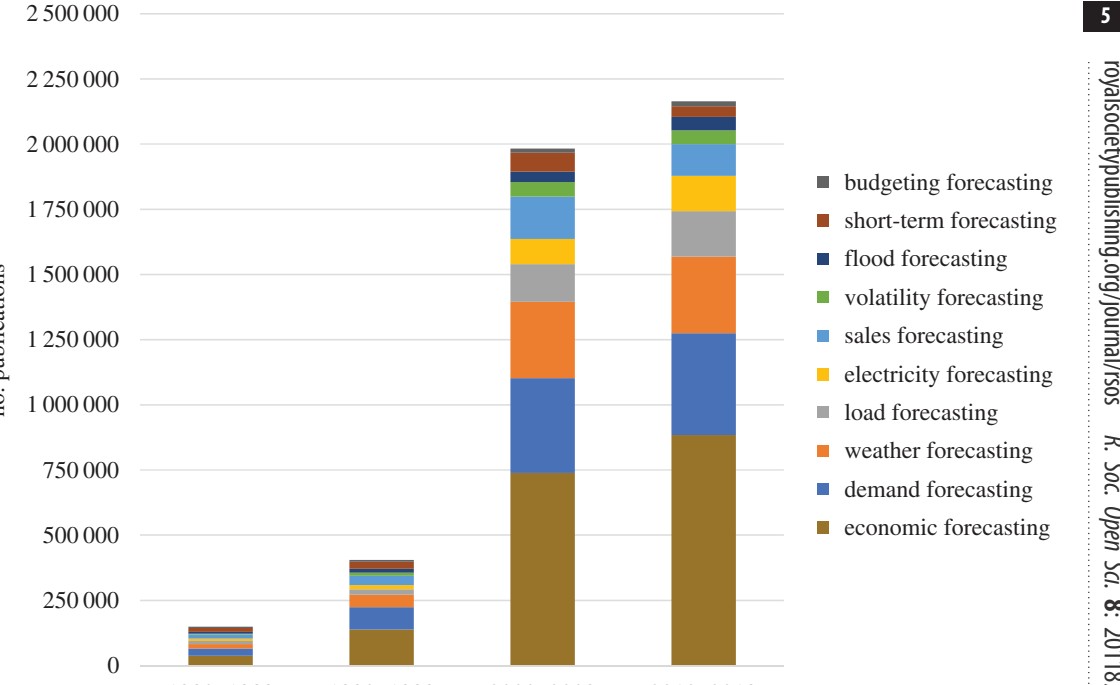

**Figure 3.** Distribution of most frequently searched forecasting topics according to Google Scholar over time.

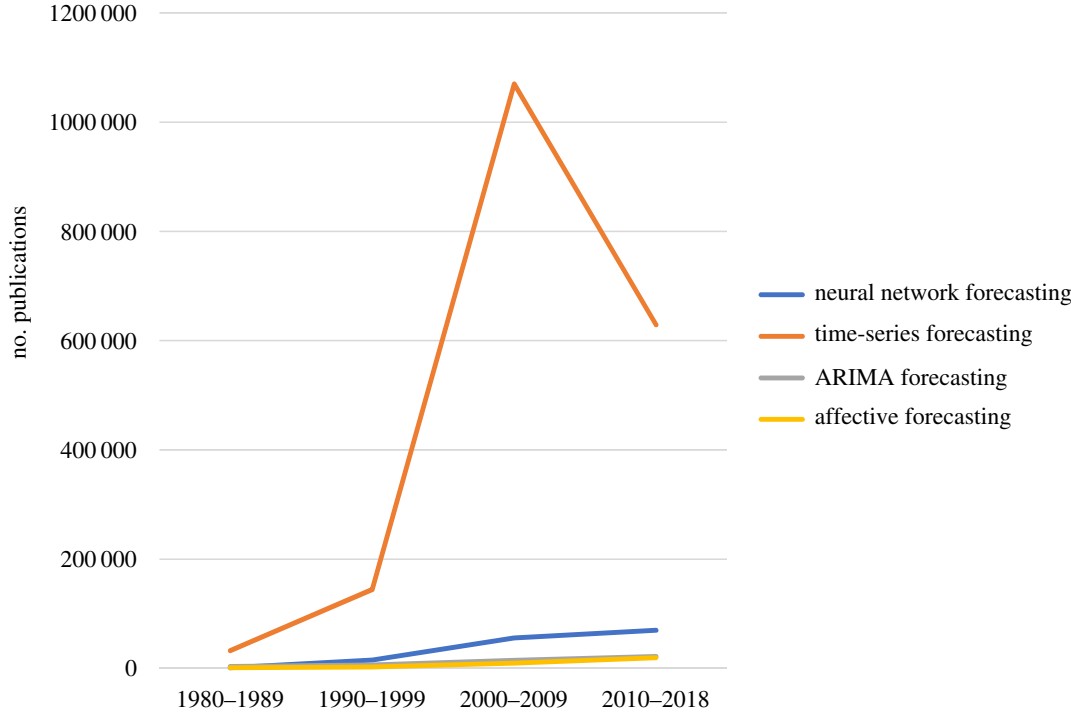

**Figure 4.** Most frequently searched forecasting methods according to Google Scholar over time.

Note that the search terms used to derive figures 2–4 are not mutually exclusive, which can result in an optimistic view of publication numbers. However, the overall trend is likely to persist despite the potential for over-representation of some publications.

### 2.1.1. Determination of relevant journals

Table 2 lists the most cited peer-reviewed journals that appeared most frequently when searching for the specific search key terms of table 1 on Google Scholar. Additionally, we determined the most prestigious

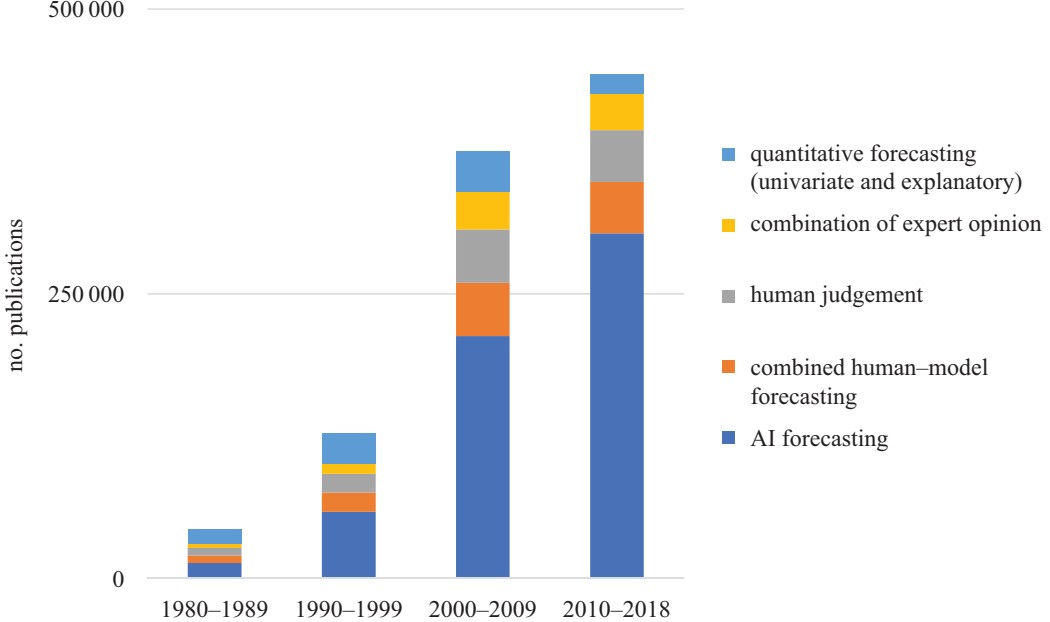

**Figure 5.** Distribution of publications in various forecasting fields using Google Scholar over time.

**Table 2.** Relevant journals for each field.

| human judgement | quantitative forecasting |
| --- | --- |
| *International Journal of Forecasting* | *International Journal of Forecasting* |
| *Journal of Forecasting* | *Journal of Forecasting* |
| *Journal of Behavioral Decision Making* | *Management Science* |
| *Psychometrika* | *Neurocomputing* |
| *Psychological Assessment* | *Computers & Operations Research* |
| | *Association for Uncertainty in Artificial Intelligence* |
| | *Journal of Machine Learning Research* |
| | *Proceeding of the AAAI Conference on Artificial Intelligence* |
| | *Transactions of the IEEE Computer Society* |
| | *ACM SIGKDD Conference on Knowledge Discovery and Data Mining* |
| human–machine interaction | combination of forecasts |
| *Computers in Human Behavior* | *Risk Analysis* |
| *Journal of Behavioral Decision Making* | *International Journal of Forecasting* |
| *Ergonomics* | *European Journal of Operational Research* |
| *International Journal of Forecasting* | *Management Science* |
| *International Journal of Industrial Ergonomics* | *Operations Research* |
| *Journal of Business Research* | |
| *Technological Forecasting and Social Change* | |

conferences in the field of machine learning and artificial intelligence using the automatic H5-index ranking provided by Google Scholar. In the domain of forecasting with quantitative models, we also used the *Transactions of the IEEE Computer Society*, which bundles the proceedings of relevant conferences (*IEEE Transactions on Pattern Analysis and Machine Intelligence, IEEE International Conference on Big Data, IEEE Conference on Data Mining, IEEE Transactions on Knowledge and Data Engineering*). These journals and conference proceedings provided an initial starting point. Publications from other sources were included if they were cross-referenced in the original set of journals and were considered relevant to this review.

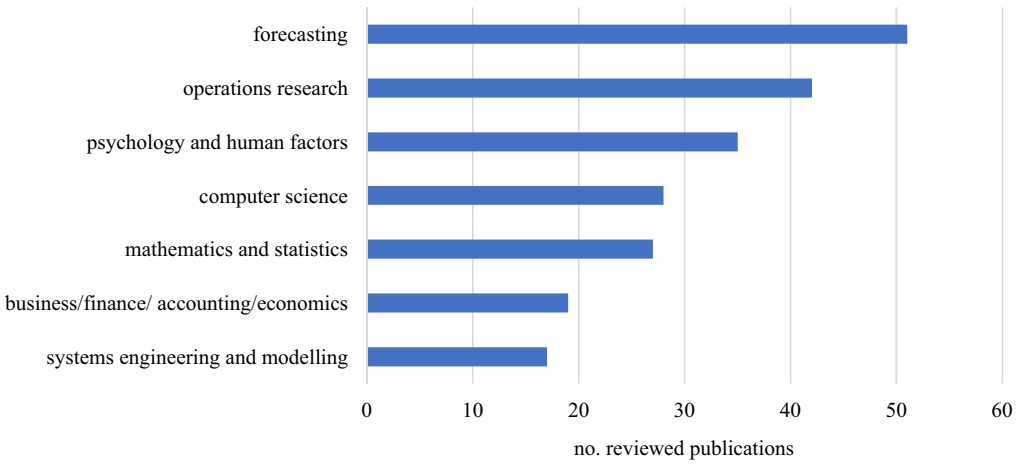

**Figure 6.** Distribution of sources along categories of initial search.

After determining the relevance of the pre-filtered literature by reading the abstract and conclusion, we identified 288 sources to be relevant to the final version of the literature review. Initially, we started with approximately 200 sources, which then 'snowballed' into the larger number of reviewed publications. Figure 6 depicts the different categories and fields in which the assessed literature of the initial search was published. This was achieved by assigning a journal to a field if the name of the journal or the title carried certain terms.

All the journal publications and most of the books were downloaded to be analysed using the library of the University of Southern California. The databases and websites used for this review were: Google Scholar, Library of the University of Southern California, Mendeley, ScienceDirect, EBSCO, IEEEXplore, Emerald, WISO and INFORMS.

# 3. Human judgement

Human judgement, which is also referred to as *judgemental forecasting* in the literature, employs one or multiple forecasters to provide an opinion. In this section, we distinguish between aspects arising in individual human judgement and judgement using groups. The first section discusses issues concerning using individuals when generating a forecast, while the second section elaborates on group techniques.

## 3.1. Individual human judgement

This section reviews individual human judgement, and related topics including probability elicitation, the impact of incentive schemes, forecaster calibration, training, as well as scoring rules.

### 3.1.1. Probability elicitation

Using human judgement for forecasting purposes, a decision maker who requires a forecast on a specific problem is faced with eliciting probability forecasts from an expert and combining multiple judgements [8]. We distinguish between eliciting probability distributions over the space of all outcomes, which requires methods to construct the continuous distribution, and point forecasts.

Concerning individual human forecasters, research focuses on elicitation and evaluation methods (e.g. [13,14]), and multiple ways to construct continuous probability distributions from the elicited point estimates (e.g. [15–18]).

There exist three main methods of eliciting expert judgement. The first method provides a fixed probability and asks the expert for the corresponding variable value (FP). The second approach provides a fixed value and elicits the corresponding probability (FV), and the third method is a combination of the two [13]. Other methods rely on pairwise comparisons of events [19] or subsets of the continuous variable [20].

Abbas *et al*. [21] assessed the FP and FV methods along several dimensions including monotonicity, accuracy and precision of estimated fractiles in a behavioural experiment, and found a slight superiority of the fixed variable value approach. They also found that participants preferred this approach, alleging

that fixed value estimates were more familiar to them from their everyday life. Despite the results of the elicitation methods being similar, the insights suggest that how forecasting questions are presented to experts impacts the speeds and accuracy of the resulting forecast.

Not all tasks require the forecaster to provide a probability distribution over the set of possible outcomes. Instead, forecasters might have to provide point estimates or intervals in which the quantity to be forecast will be with a certain probability. Lawrence *et al.* [7] provide an extensive overview of research investigating the performance of human point forecasts. Their meta-study found that with point forecasts, the accuracy of the elicited forecast depends on the used data. While the point estimates provided by statistical methods were more accurate when using random data, human forecasters performed well when using time-series data [7]. Desanctis & Jarvenpaa [22] and Harvey & Bolger [23] found evidence that whether visualization improves forecasting accuracy depends on the used data. If the data contained trends, visualization led to an improvement of point forecasts. In the case that the data did not feature any trends, there was no clear evidence supporting the representation of data by visualization over table format.

The next subsections investigate factors that impact the quality of the elicited forecast, as well as methods to control for biases and past forecasting performance.

### 3.1.2. Incentive systems

Incentive schemes in forecasting are supposed to foster truthful reporting by human judges [24–27]. Using game theoretic terminology, truthful reporting refers to an agent revealing his, or her, true opinion or assessment to the less-informed principal. Ottaviani & Sørensen [28] and Lichtendahl & Winkler [29] have shown that forecasters who compete to be the most accurate have an incentive to not report their true opinion. Lichtendahl *et al.* [30] have shown that an advice seeker benefits from setting up an incentive scheme that rewards strategic behaviour. The rationale is that the variance of the quantity predicted by the experts is greater under a competitive scheme. Because the experts emphasize private information only available to them, the individual forecasts are less correlated, which was shown to improve the accuracy of the joint forecast. According to Morris [31], dependence between forecasters due to common sources of information, educational background, etc. is one of the biggest challenges when combining individual judgements into a single forecast.

Using this game theoretic insight, researchers in the field of artificial intelligence have developed an algorithm to prevent untruthful reporting and to determine the human forecaster who gives the most truthful assessment [32].

In an extensive review, Lerner & Tetlock [33] surveyed research discussing the effects of holding people accountable for their predictions and forecasts. They found that accountability is not a panacea because it can amplify certain biases. These insights imply that incentives have to be carefully tailored to the forecasting problem at hand, such that they attenuate and not amplify existing biases [33]. Camerer & Hogarth [34] also used a meta-analysis to investigate the effects of payments to judges. They found that incentives sometimes improve performance, but often do not. Whether incentives worked was found to be task specific. If the judge's task was effort-responsive, i.e. the outcome could be improved by putting in more effort, incentives were found to work well. Examples of effort-responsive tasks provided by those tasks included judgement, prediction, problem-solving or recalling items from memory. Once the problems became more complex and intricate, incentives were found to hurt performance, supposedly because it led judges to neglect available heuristics [34]. Given the ambiguous nature of these results, more research into the effectiveness of incentives for forecasting purposes appears necessary.

### 3.1.3. Forecaster calibration and training

Tetlock [35] introduced the distinction between *hedgehogs* and *foxes*, where term hedgehog is used for domain specific or niche forecasters, who deliberately do not consider other domains when making forecasts, while foxes are trying to be more aware of the 'big picture'. Considering the forecasting problem from multiple perspectives and drawing on multiple sources of information, foxes tend to outperform hedgehogs in forecasting competitions [35,36]. Kahneman [37] extensively discusses biases in human decision making, including confirmation bias and overconfidence in one's abilities, that apply to the human forecasting domain. Chang *et al.* [38] have shown that these biases are not insurmountable, and that forecasting accuracy can be improved by short trainings (see also [39]). These trainings address human flaws in understanding probability [40–44], but also incorporate procedures to explicitly search for information that counter personally held beliefs, and engage in

counterfactual thinking (e.g. [45]). To overcome or mitigate the influence of availability bias in the selection of analogies for forecasting purposes, Green & Armstrong [46] devise a five-step approach. This structured use of analogies, which requires experts to rate the similarity of the chosen analogies with the described target situation, was shown to increase accuracy from 32% to 46% when predicting decisions in eight conflict situations [46].

Sanders & Ritzman [47] have shown that training judges in gathering and analysing contextual data had a bigger impact on forecasting accuracy compared with only training them in technical/statistical aspects. Assessing the impact on forecasting accuracy when training humans to use algorithms remains an important and, relatively, open field of future research.

Armstrong & Green [4] emphasize that accuracy is greatly improved if the human forecasters are provided with structured checklists. Instead of trainings, which might require forecasters to remember concepts for a longer period of time, these checklists are supposed to ensure that the expert adheres to forecasting principles and that the influence of biases is mitigated. The purpose of these lists is to ensure adherence to the Golden Rule of Forecasting and the choice of simple over complex models [4]. Stewart and Lusk [48] identified seven components impacting the performance of judgemental forecasts, which include environmental predictability, fidelity of the information system, match between environment and forecaster, reliability of information acquisition, reliability of information processing, conditional bias and unconditional bias. Several of these components are addressed by the checklists of Armstrong & Green [4].

### 3.1.4. Scoring rules

Scoring rules provide summary measures of probabilistic forecasts by assigning numerical scores based on the predictive distribution and on the event or value that materializes [26,49]. In this context, such measures can be used to assign differential weights during the combination stage. Furthermore, scoring rules can be used to incentivize the assessors into expending effort by tying compensation to their scores [50].

Assuming an expected utility maximizing forecaster, a proper scoring rule ensures full revelation of the forecaster's subjective belief. More formally, if $r$ is the vector of reported beliefs and $p$ is the vector of subjectively held beliefs, then for every $r$

$$E(S(p)) > E(S(r)),$$

where $E$ denotes the expectation and $S(\cdot)$ is the scoring rule. This means that an individual expert maximizes expected utility only if he or she reports his or her own belief truthfully. Any reported belief that deviates from his or her own results in a decrease of expected utility and is therefore unfavourable.

The most widely used scoring rules in forecasting are the quadratic/Brier, logarithmic and spherical scoring rules [51,52]. Harte & Vere-Jones [53] proposed an entropy score in forecasting geological events, and Diebold & Mariano [54] elaborated on case-specific loss functions. Scoring rules take on different functional forms depending on the forecast object, i.e. whether it is a categorical quantity, intervals or continuous distributions [26]. To evaluate point forecasts, error measures such as the absolute percentage error, are also being used [55].

The main criticisms of these scoring rules are their disregard of task difficulty and the lack of sensitivity to distance [56]. To address this problem, Winkler [57] developed asymmetric scoring rules and discussed their advantages using a weather forecasting case study. Scoring rules that are sensitive to distance punish distributions with heavy or 'fat' tails. For example, the quadratic scoring rule assigns the same score to the forecasts $p_1 = (0.3, 0.4, 0.3, 0, 0)$ and $p_2 = (0.2, 0.4, 0.2, 0.1, 0.1)$, if the second outcome materializes. A scoring rule sensitive to distance would assign a higher score to the forecast with lower variance. Distance-sensitive scoring rules were developed by Epstein [58] and Staël von Holstein [59] for categorical forecasts and were expanded upon by Matheson & Winkler [51]. One of the most frequently used scoring rules sensitive to distance is the ranked probability score, which has been expanded by Boero et al. [60]. The ranked probability score was shown to be superior to the quadratic and logarithmic scores in forecasting economic aspects, such as inflation. Another family of scoring rules that are sensitive to distance are so-called beta scoring rules [61,62].

Merkle & Steyvers [62] found that when choosing a rule, it is not sufficient to choose scoring rules that are proper, but one should consider the specific way forecasters are rewarded and penalized. In their opinion, the scoring rules belonging to the beta family offer a decision maker the opportunity to tailor the scoring rule to his or her needs.

## 3.2. Group judgement

To reduce the impact of bias of one individual forecaster and to increase forecasting accuracy by canceling random error, multiple humans can be employed. Group forecasting relies on qualitative or contextual data provided by multiple human forecasters. There exists a multitude of qualitative forecasting approaches, which include but are not limited to Delphi [63,64], market research [65], panel consensus, visionary forecast and historical analogy [46,66,67], group discussion [68], decision conferencing [69,70], nominal group technique [71] and focus group. In the following subsections, we discuss the most well-known techniques: focus group, nominal group technique and Delphi method [72]. We focus on these three techniques because they were found to be the most common ones when reviewing the literature. We refer the reader to Armstrong & Green [4] for more details.

### 3.2.1. Delphi method

The Delphi methodology was developed during the 1950s by Olaf Helmer, Norman Dalkey and Ted Gordon at the RAND Corporation [73]. It was not designed to replace quantitative approaches or models, but to offer a structured approach if no quantitative data were available [74]. It has been applied in various fields including healthcare [75], marketing [76], education [77], information systems [78], transportation and engineering [79], and finance [80].

The key features of the Delphi method are anonymity, iteration, controlled feedback and statistical combination of the group response [64]. Anonymity is ensured by giving forecasters a questionnaire containing the forecasting problem, whose responses the other judges cannot discern. This is supposed to prevent social pressures from changing a forecaster's judgement. The anonymous responses are then statistically analysed, and the mean and variance are supplied to all the forecasters to update their prior belief. If someone's update is an outlier, the forecaster usually has to provide a reason. The process is then repeated for several rounds. To combine the individual judgements, the Delphi method often employs a linear opinion pool [81]. There exist several variations of this technique. For example, the first round can be unstructured to not constrain the forecaster [82], or structured to make the procedure simpler for the monitoring team [64].

Studies comparing forecasts produced by the Delphi method with individual human forecasts have shown an improvement in accuracy and reduction in variance, favouring the former approach [81,83]. Despite anonymity in eliciting judgements, a main criticism of the Delphi technique is the inherent pressure to conform to group opinion after the first round of iteration. This pressure could lead to valid minority judgements being disregarded, neglecting tail probabilities [84]. Psychological studies have found that the forecasting accuracy of the Delphi method benefits from emphasizing reasoning, if judges have to provide detailed explanations for their judgement. The provided reasons could then be used in the feedback process, making it more convincing to other judges who tend to be biased toward their own assessments [85]. As with any other qualitative method relying on several forecasters, the quality and accuracy of the generated forecast depends on the study design, as well as how it addresses human biases such as anchoring, framing and desirability bias [86].

### 3.2.2. Focus groups

Unlike the Delphi technique, focus groups rely on face-to-face discussions between human forecasters on a predefined forecasting topic under the supervision of a moderator [87–89]. The advantages of this method are the simplicity of setting up the group, fast and easy sharing of information, and supposedly high acceptance of the group opinion by individual forecasters [72]. The method suffers from several downsides, including susceptibility to groupthink [90], which might be exacerbated in comparison with Delphi by its reliance on face-to-face discussions, a desire to be accepted [91], and incongruences due to the social status of group members [92]. The method does not define how individual judgements are to be combined and the choice of the combination rule depends on the moderator and the social dynamics of the group. Armstrong [93] argues that the use of focus groups to derive forecasts is not valid because the method violates forecasting principles. One of these principles prescribes the independent generation of a forecast by each group member, which is not followed by the procedural design of focus groups [93].

### 3.2.3. Nominal group technique

The nominal group technique is a structured method to elicit the judgement of forecasters in a physical group. It was developed to compensate for some of the shortcoming of the Delphi technique,

emphasizing creativity through group dynamics [72]. The process can be divided into five steps: first, the moderator poses the forecasting question. Then each forecaster individually produces a forecast, which is then explained to other members of the group to generate debate. These forecasts are subsequently anonymously assessed and ranked by each individual, before being combined by the moderator, commonly using a linear opinion pool [94]. This technique has several advantages over the previously discussed ones. In contrast to a focus group, the nominal group technique follows a clear structure and is not as prone to groupthink and social pressure. It is better than Delphi when it comes to stimulating creativity and tends to be less time consuming because it does not involve multiple iterations [94]. Nevertheless, the method also suffers from drawbacks, such as a limit on the number of forecasters for it to be effective, risk of groupthink (compared with Delphi), and requiring the forecasters to be in the same physical space at forecasting time [72]. Several studies suggest that the nominal group technique is less accurate and reliable than Delphi (Hutchings *et al.* [95], Rowe & Wright [81]).

# 4. Quantitative forecasting methods

In contrast to human judgement, quantitative forecasting methods mostly rely on data to derive a prediction model. The choice of a quantitative method depends on factors such as the context of the forecast, the relevance and availability of historical data, the degree of accuracy desirable, and the time period of the forecast [67]. In our review, we first distinguish quantitative methods along the variables used to form a prediction. Univariate methods use past data, either time domain or frequency domain, of the variable to be forecast to make a prediction. Explanatory methods use data both on the variable to be forecast and extraneous variables to derive a forecasting model. Models that can be used for univariate and explanatory purposes are grouped together into overarching methods. We are aware that there exists a wide variety of quantitative forecasting methods, and we focus only on the most popular ones found during the literature search process. Also, we omit specific applications of the presented methods for the sake of brevity.

## 4.1. Univariate methods

This section focuses on methods that use historical data on the same variable that the prediction is formed on. We distinguish these methods further into time and frequency domain methods.

### 4.1.1. Time domain methods

Time domain methods use historical data to identify patterns and pattern changes. They are commonly applied in diverse fields such as finance [96], electricity markets, retail and optical transport networks [97]. Although there exist multiple models for time-series forecasting, we limit this review to the most commonly used ones and exclude more application-specific methods such as 'robust-trend' [98] and 'theta' [99].

The accuracy of the different time-series models presented in this section and the impacting factors have been discussed in depth by Makridakis *et al.* [100]. They found that the model has to be chosen in accordance to the available data to maximize accuracy. For example, if available data are on a quarter-year basis instead of on a yearly basis, it is advisable to use a time-series model that is capable of incorporating seasonality. Chatfield [101] distinguishes between univariate, multivariate and judgemental time-series methods. Whereas univariate models draw on past data of one particular variable to forecast the future, multivariate time series rely, at least partly, on values of more other related series. Judgemental time-series models concern humans extrapolating time series into the future and adjusting the series for contextual data [102]. Chatfield [101] concluded that there is no 'best' forecasting method. The choice of model rather depends on the objective in producing the forecast, the type of time series and its statistical properties such as trend or seasonality, the number of past observations available, the length of the forecasting horizon, the number of series to be forecasted and the cost allowed per series, the skill, experience and interests of the analyst, and the computer programs available. He derived several general principles related to time-series models. First, post-sample forecast errors might not be minimized by fitting the 'best' model to historical data, as the 'best' model might be overfitting, i.e. incorporating too much noise of the historical data in its forecast. Univariate models are most suitable for short-term (up to six months

into the future) forecasting and the combination of forecasts from different models generally outperforms any individual method. This can be explained by the fact that each method makes different assumptions about trends, seasonality, etc. and combining different models reduces the effect of bias compared with when using only one. Another observation is that forecasting accuracy benefits from a combination of time-series forecasts derived by different methods. Iosevich *et al.* [103] for example, outlined a dynamic modelling approach that combines multiple time-series forecasts and illustrated its high accuracy.

Several forecasting competitions covering a range of real-world time series have shown that more sophisticated models do not necessarily outperform simpler ones [2,3,6,100,104,105].

The measures used in these competitions to assess the accuracy of time-series models were the symmetric mean absolute percentage error (sMAPE), average ranking, percentage better, median symmetric absolute percentage error (median symmetric APE), and median relative absolute error (median RAE) [100]. Armstrong & Collopy [106] evaluated these measures for making comparisons of errors across time series. They were judged on their reliability, construct validity, sensitivity to small changes, protection against outliers, and their relationship to decision making. Their recommendation was to use the geometric mean of the relative absolute error when the task involves model calibration for a set of time series. To select the most accurate time-series model, they recommend using the median relative absolute error in the case of few series being available. If more time-series can be accessed, then the median absolute percentage error is recommended as selection criterion. The commonly used root mean square error was deemed unreliable, and it is not recommended when comparing accuracy across series [106].

### 4.1.1.1. Moving average

This method calculates a series of averages of different subsets of the dataset, with different variations, such as simple, cumulative or weighted forms. Where the simple moving average allocates the same weight to each data point, the cumulative version uses the cumulative average, and the weighted moving average weights are usually determined by using the data point's date. For more details, please refer to Chou [107] and Hadley [108].

### 4.1.1.2. Exponential smoothing

In contrast to simple moving average, exponential smoothing assigns exponentially decreasing weights over time. It is commonly applied to smooth data, acting as low-pass filters to remove high-frequency noise. There are several exponential smoothing approaches available in the statistical literature. Single exponential smoothing, first suggested by Brown [109] and expanded by Holt [110], applies a smoothing factor on past data, and derives forecasts by calculating their weighted average. As single exponential smoothing does not perform well if there are trends or seasonality in the data, the approach has been expanded to second- and third-order exponential smoothing. If there is only one perceivable trend in the data, second-order exponential smoothing is appropriate, which has to be extended to a three-parameter smoothing if seasonality is detected [111]. For an extensive overview of exponential smoothing methods and their applicability please refer to Gardner [112,113]. He discusses the formulations for different types of trend and seasonality and shows that the exponential smoothing methods correspond to one or more stochastic models, including regression and ARIMA [113]. One particular method of exponential smoothing that was found to be highly accurate is exponential smoothing with damped trends [114]. Using empirical data, Gardner & McKenzie [114] showed that both methods of exponential smoothing with damped trend, i.e. multiplicative and additive seasonality, have a lower mean absolute percentage error than other exponential smoothing methods. The authors attributed the improved accuracy of the proposed method to the avoidance of overshooting the data and of the excessive amplification of seasonal patterns that is inherent to exponential smoothing with linear trends.

During several forecasting competitions, Makridakis *et al.* [105] found that exponential smoothing performed well in time-series forecasting. When investigating strategies to select models to forecast time series, Fildes [115] found that exponential smoothing with damped trends was superior, especially when choosing a model that is applied to all series in an aggregate manner. Dantas & Oliviera [116] propose improving exponential smoothing for time-series forecasting by incorporating the statistical learning techniques bootstrapping and clustering. Using the M3-competition dataset, they managed to reduce forecasting error when compared with conventional exponential smoothing.

### 4.1.1.3. Autoregressive-integrated-moving average

This method, also known as the Box–Jenkins method, was developed in the early 1900s but became popular only in the 1970s [117,118]. There are three basic steps in this approach. First, a tentative model is identified. This usually happens under the assumption that the pattern of the time series can be explained by one of the three model components. In the second step, the autoregressive component relates the current value to its own previous values and the moving average component relates the current value to the previous errors. Thirdly, the integration step combines the two previous components, and is mainly determined by the autocorrelation of lag variables [118].

The most important aspect of using this model is to achieve stationarity. If one can determine a trend in the modelled data, stationarity has not been achieved yet, and more differencing is necessary. Once stationarity is achieved, model parameter coefficients can be estimated, and the forecast is generated.

The ARIMA method is very popular because of its wide applicability. It is able to incorporate seasonality and trends, deemed highly accurate, and extendable to exogenous variables [118]. Incorporating exogenous variables gives the so-called ARIMAX-model, which can be considered a hybrid between time-series and correlational models.

Drawbacks of this approach are its complexity, its requirement of large datasets, and its need for updating once new data are collected [118]. For further details on ARIMA and its extensions, refer to Makridakis *et al.* [119] and Pankratz [120]. One commonly used extension is the X-11-ARIMA model that decomposes time series into seasonals, trend cycles and irregular elements. For details on how to set the filters for seasonal adjustment, refer to Dagum [121].

### 4.1.1.4. Unobserved components model

This model, which can be considered an alternative to ARIMA, was first introduced by Harvey [122]. The UCM can be described as multiple regressions models with time varying coefficients. It additively decomposes a time series into trend, seasonal, cyclical and irregular components, and allocates different weights to events, depending on when they occur in the series. An example of UCM forecasting demand in telephone networks is discussed by Tych *et al.* [123]. For further details on UCMs, refer to Young [124].

### 4.1.1.5. Naive approach

The naive approach forecasts at any point *t* the last observed value (at *t*-1), implying that the forecaster does not possess any further knowledge than what has been observed last [125]. This simplicity makes naive methods the most cost-effective models according to Hyndman & Athanasopoulos [125]. Several studies with empirical studies have shown that these models perform well in environments where patterns are hard to forecast [125]. Because of its ease of implementation and freedom of any other assumptions, the naive approach is often used as the baseline against which other forecasting methods are compared.

Note that the term naive must not be confused with simple. Green & Armstrong [6] make the distinction of simple and complex forecasting. According to the authors, forecasting is simple if the forecasting process is understandable with respect to methods, representation of prior knowledge in models, the relationships between model elements, and relationships among models, forecasts and decisions. Using the definition by Green & Armstrong [6], the naive approach is simple, but not every simple forecasting method or process is naive.

### 4.1.2. Frequency domain methods

Frequency domain methods are approaches that account for variation in time-series data by deriving its cyclical components, such as sines and cosines. The combination of these components is then propagated into the future to form a prediction. According to Chambers *et al.* [67], frequency domain models tend to be elaborate and time-intensive to construct.

### 4.1.3. Fourier time-series decomposition

This method aims to explain the time series entirely as a composition of sinusoidal functions. By mapping time-series data to the frequency domain, seasonality and historic trends can be incorporated by including additional frequencies from the decomposed time series. Once the number of sinusoidal functions is determined, their combination is propagated to the future. Taking the inverse Fourier

transform of the combination then yields the forecast in the units of the original time series. Increasing the number of sinusoidal functions results in a more accurate approximation of the original time series but also increases chances of overfitting [126].

## 4.2. Explanatory methods

Methods belonging to this category use data on the variable to be forecast as well as on explanatory variables to train models. The trained models then make a prediction from other instantiations of the explanatory variables. Note that the methods in this section are not exhaustive. We focus on the methods most commonly encountered during our survey of the literature, including regression models, support vector machines and ensemble methods. Because they are most often used in the context of ensemble methods, we also omit a separate discussion of tree models.

### 4.2.1. Regression methods

These models use a dataset containing values for input and response variables. Linear regression applies a linear function to each observation and minimizing the residual sum of squares, one can derive the coefficients of each input variable. The assumption of a linear relationship between response variable and predictors can have negative implications for the accuracy of the model. For example, if the real relationship between response and predictor is nonlinear, the linear model's accuracy can, in some cases, be very low. Furthermore, the correlation of error terms, non-constant variance of error terms, outliers, and collinearity all affect the accuracy of linear regression models negatively [127].

In logistic regression models, the response variable is binary, which means it can be used for classification purposes. It computes the probability that an object is of a certain type, or belongs to a certain class, is used often in artificial intelligence and machine learning applications. For example, Korkmaz et al. [128] used logistic regression to generate probabilistic forecasts of civil unrest using social network data from Twitter, Facebook, etc. Conditional random fields (CRFs) is a common tool in artificial intelligence that uses regression efficiently to model effects of interactions of objects in large datasets [129]. An exemplary application of CRF to forecast loads in an electricity network is given by Guo [130].

Closely associated with regression models are prediction intervals. The prediction produced by a linear regression model is the mean of the distribution of possible values. Depending on the assumptions made about the distribution around the mean, one can then compute the probability with which the real value will fall within a specified interval. Patel [131] provides a comprehensive explanation of prediction intervals and how they are calculated using different probability distributions.

For details on simple and multiple regression, refer to Clelland et al. [132] or Makridakis et al. [119].

### 4.2.2. Support vector machines

Support vector machines are a classification approach that has been applied to many different forecasting or prediction cases, including stock market forecasting [133], financial time-series forecasting [134], and predicting the energy generated by renewable sources [135]. Comparing the accuracy of support vector machine with multilinear regression and neural networks when predicting electricity consumption, Kayetz et al. [136] found that support vector machines outperformed the benchmark.

The basic idea of support vector machines is to find a separating hyperplane that best divides the observations in a dataset. Support vector classifiers, which are an integral part of support vector machines, only produce a linear decision boundary. This restriction is relaxed in support vector machines, which combine a nonlinear kernel specifying the similarity of two observations and a support vector classifier [127]. For more details on support vector machines please refer to Hastie et al. [137].

## 4.3. Overarching quantitative models

For some quantitative models, the distinction between time and frequency domain models is not adequate, as they can either be considered a combination of the two or can be used for both purposes. Although there are more modelling approaches in this category, we focus on neural networks, Bayesian networks, ensemble methods and simulations.

### 4.3.1. Neural networks

Neural networks were proposed in the 1940s by McCulloch & Pitts [138]. They consist of neurons that contain the activation at each time step, a possible threshold that can be changed by a learning function, the activation function which calculates the activation for each time step, and a respective output function. These neurons are connected by synapses, each with their own weights that are adjusted through subsequent learning and backpropagation [139]. The success of these models was initially impeded by the lack of computing power, data and flaws inherent to their architecture [139,140]. Some of these shortcomings of single-layer perceptrons were addressed by introducing more layers (deep models) and backpropagation [141]. As a result, deep neural networks have emerged as a powerful modelling framework for various supervised and unsupervised machine learning problems. One can broadly distinguish between two types of neural networks, differing in their suitability for the classification or prediction problem at hand [142]. While convolutional networks (CNNs) are mostly applicable to processing data from topologies, such as pictures, residual neural network (RNNs) are capable of considering the influence of time [143]. In comparison with other network time-series models, for example hidden Markov models (Rabiner [144], Hassan & Nath [145], Raghavan *et al*. [146], Date *et al*. [147]), RNNs remain efficient when long-range dependencies have to be modelled, the state space is huge, and the dynamics of the time series are non-Markovian [148]. The specific architectures of RNNs considered responsible for these recent improvements of accuracy are bidirectional neural networks (BNNs) and long short-term memory systems (LSTMs) [143]. LSTMs were developed by Hochreiter & Schmidhuber [149] to address the deficit of conventional RNNs to handle long-term dependencies.

Because of their improved performance, RNNs have been used in a variety of fields, including demand forecasting in the aviation industry using a combination of networks [150], electric load forecasting [151,152], electricity price forecasting [153] and finance [154]. LSTMs in particular have also been used for clinical time-series modelling [155,156], traffic forecasting [157] and predicting stock market returns [158].

However, there exist multiple papers discussing the limitations of neural networks in forecasting. Critics point out that these models often require large datasets, which might not be available for very specific forecasting tasks [137]. Furthermore, a neural network is often viewed as a 'black box', making the forecasting process opaque to the user [4]. When comparing real-world time-series forecasts of statistical methods and neural networks, the former seems to outperform the more advanced machine learning methods on the basis of sMAPE and mean absolute scaled error (MASE) [2,3]. In a more recent edition of the M-competitions, one of the best performing models was a hybrid model of exponential smoothing and a recurrent neural network [159], which is described in detail in Smyl [160]. Hewamalage *et al*. [161] argue that although the performance of the model presented by Smyl [160] is impressive, it might not be adopted by non-expert judges when making forecasts. In their opinion, exponential smoothing and ARIMA are superior to RNNs when it comes to user-friendliness, efficiency and the wide availability of standard software. They compare different 'off-the-shelf' RNNs with different configurations with the accuracy of exponential smoothing methods and ARIMA. Using a variety of time-series datasets, they find that RNNs are capable of modelling seasonality directly as long as the datasets possess homogeneous patterns, otherwise the model requires a preliminary deseasonalization. Furthermore, they show that a stacked architecture with long short-term memory cells with peephole connections perform best compared with exponential smoothing and ARIMA. They conclude that RNNs do not constitute a silver bullet for forecasting time series in comparison with more established methods, but remain confident these methods will become part of standard forecasting software [161].

### 4.3.2. Bayesian networks

Bayesian networks (BNs) are representations of probabilistic dependence between a given set of random variables and an acyclic graph [162]. They provide an easy and fast way of updating prior beliefs, as well as eliciting dependence information [163]. The network's structure can either be derived by human experts or by employing historic data [162]. Using human experts, Stiber *et al*. [164] constructed Bayesian networks for each of their forecasters and combined the forecasters' posteriors using a linear opinion pool. The weights of the linear pool were determined by posterior probability weights, i.e. by the accuracy of the individual assessments. The assumption in this research was that the forecasters previously agreed upon the general structure of the BN. Etiminani *et al*. [165] claim that the use of

BNs in forecast combination can be subdivided into the problem of combining the structure and the problem of combining the parameters when employing multiple experts.

Etiminani *et al.* [165] address the issue of combining parameters. The combination of structure, meaning the creation of one BN out of many, is discussed by del Sagrado & Moral [166], who distinguish between topological fusion and graphical representation of consensus. Whereas topological fusion first obtains a consensus structure and then combines the model parameters, graphical representation of consensus approaches the combination problem in the opposite order [166].

Despite their advantages, Bayesian networks feature downsides according to French [163]. Depending on the process followed to derive the network, a significant amount of interaction with human forecasters is required to determine the consensus structure. If one wants to pursue an algorithmic combination of networks, there is a risk of inconsistencies if the various opinions on the structure diverge too much [163]. While Henrion [167] and Nadkarni & Shenoy [168] have explored ways to derive the qualitative dependence structure, eliciting the quantitative dependence structure has been identified as the main issue by Druzdzel & Van der Gaag [169] and Renooji [170]. Determining causal effect was discussed by several authors, such as Peña [171], but increasing the number of variables makes it difficult to generate and update the network. Approaches to lessen the assessment efforts include non-parametric BNs [172,173], piecewise-linear interpolation [174] and noisy-OR gates [175]. This implies that while there exist algorithms to derive the structure of the network using past data, there are challenges if one wants to combine Bayesian networks with different underlying structures.

### 4.3.3. Ensemble methods

Because ensemble methods can employ different methods, including trees, support vector machines, or neural networks, they can be considered overarching models. They generate a set of alternative models, with the premise that the combination of diverse forecasts increases overall forecasting accuracy. Common model types used in ensemble methods are trees models, as well as neural networks [176]. However, other predictive models, such as support vector machines, can be used in ensemble methods [177]. Similar to the combination of human judgement and forecasts produced by quantitative methods, the accuracy of ensemble methods partially depends on how the predictions by each method are combined. The most common approaches are naive Bayes classifiers, bootstrap aggregating (bagging), boosting, Bayesian model averaging, or Bayesian model combination [178].

The mostly used ensemble methods are bagging and boosting. For each model, bagging [179] draws random samples from the data (with replacement) and trains a model on it. The individual models are then combined by assigning each prediction the same weight. Boosting [180,181] builds an ensemble by training each new model instance to emphasize the particular instances in the training data that the previous models misclassified.

In a study comparing the accuracy of bagging and boosting of decision trees and neural networks, Opitz & Maclin [176] found that while bagging is almost always more accurate than a single classifier, it is sometimes less accurate than boosting. At the same time, boosting can create ensembles that are less accurate than a single classifier, especially when using neural networks. Opitz & Maclin [176] ascribe their findings to the characteristics of the used dataset. In the case of noisy datasets, boosting overfits, thus decreasing overall accuracy. They also found a diminishing return of adding ensembles, implying that most of the gain of an ensemble comes from the first few classifiers.

Applications of ensemble methods can be found in diverse fields, including weather forecasting [182], bioinformatics [183], medicine [184] or finance [185]. Oliveira & Torgo [186] explored ensemble methods in time-series forecasting, showing that they can result in a higher accuracy than ARIMA models.

## 4.4. Simulation models in forecasting

Borchev [187] and Sterman [188] classify simulation models as agent-based, system dynamics and compartmental models. They are common in modelling the spread of diseases and assessing the cost-effectiveness of interventions, such as in Crooks & Hailegiorgis [189], Sadilek *et al.* [190], or Bendor *et al.* [191]. In the field of policy analysis, simulations are applied to assess the effects of policies on the economy, as discussed in Homer & Hirsch [192], Tesfatsion [193,194] and Barlas [195]. Although simulation models are widespread, they can be problematic. They depend on qualitative data for calibration and forecasters to validate the modelling assumptions. Depending on the forecasting problem this process can be very time consuming and elaborate [187]. Other popular time-domain

simulation models are random walks. These random walk models, such as standard or geometric Brownian motion, assume changes to be lognormal distributed and use past data to determine annual drift and volatility. One can either apply the closed-form solution or Monte Carlo simulation to derive the probability density function of the future event [196,197].

# 5. Forecast combination

After discussing human and quantitative forecasting separately, we explore issues and challenges arising when combining the two in this section. We first discuss benefits and issues arising from combining human judgement with quantitative methods, specifically human belief updating and algorithm aversion. Then, we present an overview of the most common combination methods, which are subdivided into Bayesian and non-Bayesian approaches.

## 5.1. Benefits and issues of forecast combination

In this subsection, we discuss research motivating the combination of human judgement with forecasts provided by quantitative models. Furthermore, we provide an overview of research into human advice taking and belief updating, as well as of algorithm aversion.

### 5.1.1. Motivation for combining human judgement with quantitative models

Previous research contrasting the accuracy and performance of human forecasters and quantitative models yields mixed results. Highhouse [198], Dawes [199], Schweitzer & Cachon [200], Grove et al. [9], Kuncel et al. [10] and Ægisdóttir et al. [11] have shown that algorithms usually outperform human judges on forecasting tasks, although a real-world example of Nike®, given by Worthen [201], warns against relying exclusively on computer models without any human supervision and input. A survey of 240 US corporations found that only 11% used forecasting software, and of those 60% routinely adjusted the generated forecasts based on individual judgement [202]. Fildes & Petropoulos [203] found similar results. Since 2003 the use of pure judgemental forecasting has decreased, while the use of combined algorithmic/judgemental methods has increased [203].

Lawrence et al. [7] propose a rough forecasting procedure that draws on the advantages of quantitative models and human judgement contingent on the availability of historic data. It assumes that contextual information is used by human forecasters, and quantitative data are analysed by quantitative models to inform the human. If there is no quantitative data available, the forecast is developed by the human without machine assistance. Armstrong [204] found that when contextual or domain knowledge is available, human forecasters tend to outperform statistical methods. Brown [205] also concluded that advice seekers should place a stronger emphasis on human judgements, a conclusion that was also supported by applied research conducted by Chatfield et al. [206] in the electric utility industry. Nevertheless, the complementary strengths of human judgement and quantitative models suggest that the combination of these methods might yield superior forecasting results [207]. A recent study into forensic facial recognition has shown that the combination of neural networks and human forecasters has the potential to stabilize classification performance, decreasing variability and increasing performance of average forecasters [208]. Yaniv & Hogarth [209] have shown experimentally that when contextual information is scarce, statistical forecasts usually outperform humans. In their study, forecasters achieved the highest accuracy with the combination of a statistical base-rate model and human judgement of contextual data. Miyoshi & Matsubara [210] found that simply averaging over the forecast produced by a recurrent neural network and a set of human forecasters outperformed the stand-alone quantitative forecast and human forecasters. They also developed a flexible algorithm that can determine the optimal number of human forecasters as a function of the expected error of the quantitative model's forecast.

Because human forecasters interact with quantitative models to derive a joint forecast, one can hypothesize that the quality of the derived forecast depends at least in parts on the quality of the interaction. Research has studied when humans trust quantitative models and what factors impact the amount of updating that occurs after the judge has been provided with their results. We address these questions in the following sections. For more general reviews of the psychological aspects of human forecasting and advice taking, please refer to Lawrence et al. [7] and Bonaccio & Dalal [211].

### 5.1.2. Human advice taking and belief updating

In human/machine forecasting, a quantitative model might be used to inform the human expert's judgement or multiple experts provide their judgements to a decision maker. One intuitive research question that arises is how humans take the provided advice and use it to update their belief. On a human-to-human basis, Önkal et al. [212] and Ayton & Önkal [213] investigated empirically the driving factors in using recommendations and advice. The authors assess whether experienced or presumed credibility has more impact on judges' readiness to use advice. Authors such as Fogg [214], Wathern & Burknell [215] and Harvey & Fischer [216] have argued that the former, i.e. a good track record of making right forecasts, has the biggest impact on whether users apply the forecasters' recommendations.

Other researchers, such as Armstrong [217] and Kahneman [37], in turn claimed that presumed credibility, i.e. the credibility purveyed through the status of the advice giver, plays the biggest role in judgement adoption. Önkal et al. [212] showed, experimentally, that advice from a forecaster with high experienced credibility received a higher weight, and a lower level of credibility did not affect the weighting negatively. High presumed credibility in turn did not result in an allocation of more weight to the model, although low presumed credibility resulted in a decrease. Investigating the interaction between the two kinds of credibility, the authors also found that the weighting depends on the expertise of advice-seekers. Among non-experts, experienced credibility eclipsed presumed credibility, while both kinds of credibility were influential in determining the weight allocated to the forecaster judgement if the advice-seekers were professionals in the same industry. Extending this topic to technology and its incorporation into forecasting, research by Agarwal & Prasad [218] implies that the amount of updating depends on whether the advice receiver was an active or passive user of technology. Furthermore, Önkal et al. [219] have shown that, although seeking outside advice generally improves forecasting accuracy, the amount of updating depends on the source. They found that in the process of belief updating, information from a statistical procedure was discounted more than when the source was another human forecaster. If the two sources were either human or statistical procedures, this effect vanished, indicating that human advice is preferred over statistical procedures when both types of sources are available [219].

Several other authors investigated how humans use advice that is provided by human forecasters. Yaniv & Kleinberger [220] found that advice is discounted relative to one's prior, meaning that humans tend to assign higher weights to advice that is consistent with, or confirms, their beliefs and lower weights to other diverging views. Soll & Larrick [221] found that the two most common strategies are choosing one source and averaging them. Despite the latter proving to be more accurate in most circumstances, humans tend to prefer the former. Combining Yaniv & Kleinberger [220] and Soll & Larrick [221] could suggest that humans tend to choose the forecast that is consistent with, or confirms, their beliefs, which makes the forecast prone to bias.

### 5.1.3. Algorithm aversion

Algorithm aversion can be described as the aversion of humans to take advice if it was generated by a machine algorithm or quantitative model. Dietvorst et al. [222,223] investigate how humans use machine-generated forecasts contingent on the quality of outputs and on how much human forecasters can alter the provided forecasts. Carbone et al. [224] and Armstrong [225] suggested that allowing for human adjustment of the quantitative model might harm accuracy. Supporting this claim, Eggleton [226] and O'Connor et al. [227] found that deteriorating forecasting accuracy due to the incorporation of human judgement is mostly due to human judges reading systematic patterns into the noise of time series. Fildes et al. [228] found empirical evidence contesting this interpretation. The authors showed that these impairments could be explained by an optimism bias and the imposition of small adjustments to impose ownership of the forecasts [228].

Although previous research has shown that machine algorithms can be more reliable and forecast future events better (see [9,12]), human beings often put higher trust in human advice, as shown by Eastwood et al. [229], Diab et al. [230] and Kaufmann & Budescu [231]. In his research on trust in algorithmic decision aids, Sheridan [232] identified reliability, robustness, validity, transparency, understandability, usefulness and utility as the main drivers of trust. Whereas reliability has been confirmed extensively by Lee & Morray [233] and Muir [234], Seong & Bisantz [235] also addressed transparency, understandability and validity, and found that humans put more trust in algorithms that can be understood by users and which perform consistently well. These factors are also considered important determinants of forecasting accuracy by Armstrong & Green [4].

Dietvorst *et al.* [222,223] derived two main insights regarding humans' readiness to use machine algorithms. First, people lose trust and confidence in machine algorithms faster than in human advisors once they see model forecasting errors. This phenomenon has also been discussed by Alvarado-Valencia & Barrero [236], who have found disuse of computer models in forecasting with high task complexity and lower system performance. The authors suggested explaining the computer models and showing past performance to users, although they conceded that the delivery of this information is still controversial. A more recent paper found the opposite, suggesting that humans prefer algorithmic over human advice. Logg *et al.* [237] found that when tasked with providing numeric estimates about a visual stimulus, the popularity of songs, and about romantic attraction, most judges preferred algorithmic advice over human expert advice. This appreciation of recommendations generated by quantitative methods decreased when the judges had to choose between their own judgement and the method's, and if the judge was knowledgeable in forecasting [237].

Assessing the ability of human forecasters to select the best model, Petropolous *et al.* [238] and De Baets & Harvey [239] found that it depends largely on the quality differential between models. Using behavioural experiments, the authors found that judges distinguished between good and bad models, but less so when the choice was between models of good and intermediate quality [238,239]. The perceived quality of the model also influenced by how much judges adjusted their initial forecast. Judges were found to consider the recommendations of methods as long as they had a good track record of accurate predictions [239].

Dietvorst *et al.* [223] have also shown that offering human users the possibility to adjust or modify the algorithm makes them more likely to use the machine output. When confronted with the choice of using the machine output most judges declined to use it after seeing the algorithm make a wrong prediction, but when given the opportunity to slightly modify the output, this aversion decreased [223]. This suggests a potential trade-off between how much users are being allowed to tweak the algorithm and forecasting accuracy [228]. Fildes *et al.* [228] propose four different strategies concerning the tweaking and combination of quantitative prediction and human judgement: '50% model + 50% manager' [207], error bootstrapping [240], avoidance of small adjustments (less than 20%), and the avoidance of wrong-sided adjustments. Using their empirical data, these strategies were found to increase forecasting accuracy significantly [228]. Ahlburg [241] discusses how forecasts can be improved using Theil's decomposition. Using examples, he shows how decomposition suggest a linear correction procedure, which may improve accuracy [241].

## 5.2. Combination methods

Decision makers tend to ask multiple experts for judgements and estimates to negate bias and to obtain more accurate forecasts. A key research question is how to combine these judgements into one. As pointed out in the section on human judgement, the rules governing the combination step can either be qualitative, for example a group discussion to reach consensus, or quantitative, i.e. using a mathematical method. The quantitative approaches amount to a 'pseudo group' decision process, where the members do not interact with each other and, in some cases, may not even be aware of the existence or the identity of the other members. In recent years some of this work has  been popularized under the label of 'wisdom of crowds' [24].

Clemen & Winkler [8] discuss mathematical approaches for combining the judgements of multiple human experts and distinguish between axiomatic and Bayesian approaches. One of the main obstacles in combining multiple judgements is the potential dependence between the forecasters [242,243]. Judgements and advice originating from highly correlated sources are unlikely to improve forecasting accuracy, implying that advice from independent sources is particularly beneficial [244,245].

The next subsections distinguish between combination rules that are consistent with Bayesian statistics and those which are not. We expand on previous surveys by including more recent combination approaches, such as maximum entropy aggregation, and algorithmic procedures such as democratic opinion pools and contribution-weighted models. We draw on recent empirical studies to compare the effectiveness and suitability of different mathematical rules.

### 5.2.1. Bayesian combination methods

Bayesian approaches view the various forecasts as information that is used to update the decision maker's prior using a likelihood function over the possible forecasts [31]. One can broadly distinguish between methods that combine point probabilities, i.e. the probability of a discrete event, and

approaches combining continuous probability distributions. The following subsection discusses the most common combination consistent with Bayesian statistics.

### 5.2.1.1. Combination of point probabilities

There exist several methods to combine point probabilities that are consistent with Bayesian statistics. One method assumes independence between forecasters [8], whereas the method proposed by Genest & Schervish [246] allows for miscalibration. Another model put forth by Winkler [247] and Morris [248] assumes that each forecaster's information represents a sample from a Bernoulli process. Morris [242,248] presents a set of assumptions that need to hold for this 'Bernoulli' combination method to work. The first assumption is invariance to scale, meaning that the variance of the forecasters' priors alone provides no information to the decision maker about the uncertain quantity. The second assumption, invariance to shift, the decision maker's assessment of how surprised the forecaster is likely to be when the true value of the uncertain variable is revealed, is not conditional on the true value. This assumption implies that, if the revealed value is shifted by some amount, the assessment of the location of the forecaster's prior must shift by that amount [248]. Furthermore, the method assumes normality of the forecaster's priors.

In the single forecaster case, these assumptions are sufficient to determine the posterior as the normalized product of the forecaster's prior and the decision maker's own prior, given that the forecaster is calibrated [248]. In the case of multiple forecasters, the composite prior is the normalized product of the individual forecasters, which requires independence alongside calibration of their assessments. Therefore, the joint calibration function does not only reflect each forecaster's probability assessment ability, but also incorporates the degree of dependence among them [248]. Unfortunately, determining the joint calibration function in the case of dependent forecasters is not discussed, although the author hints at the potential difficulties [242].

Morris [248] elaborates on the axioms underlying the different approaches to judgement aggregation of point estimates or probability functions. These axioms characterize desirable properties of the processing rule that operates on the forecaster's and decision maker's priors to determine the posterior consensus probability function. The author states that the answer should not depend on who observes a given piece of data as long as there is agreement on the likelihood function, and that a uniform prior of a calibrated forecaster is non-informative. If both the forecaster and the decision maker have uniform priors, the updated distribution should also be uniform. If only the decision maker has a uniform prior, he or she should adopt the forecaster's prior. Morris [248] also discusses the meaning and implications of forecaster calibration and introduces several calibration levels that need to be considered when faced with different estimation problems, i.e. point estimate or probability density function.

A fourth model to combine point forecasts that incorporates the inter-judge dependence via a common covariance matrix was proposed by [249].

### 5.2.1.2. Probability copulas

In contrast to combination rules concerning point forecasts, copulas use continuous probability forecasts as inputs. Probability copulas use a generating function with the marginals as input arguments to derive the joint probability distribution [250]. Jouini & Clemen [251] adopted a copula approach, using the judgements of multiple experts as marginals, to derive the joint judgement. Once the appropriate copula is defined, one can determine the posterior via Bayes rule using the likelihood function and a given prior (for more details on copulas see Nelsen [252] and Durante & Sempi [253]). The approach consists of the following steps:

1. Elicit the forecasters' priors on the unknown quantity to determine the marginal distributions.
2. Determine the concordance probability, i.e. the probability that the probabilities assigned by the forecasters 'move' in the same direction / are positively correlated. The concordance probability is used as the measure of dependence between forecasters.
3. Based on the concordance probability, determine the appropriate copula structure, which is necessary to construct the joint distribution.

This approach is computationally easy as soon as one has determined the appropriate copula. The challenge lies in determining the concordance probability [251]. Eliciting the dependence parameter and the type of copula are complex tasks, which are necessary because different copula families

exhibit very different behaviour even for the same rank correlation [243]. Arbenz & Canestraro [254] propose an elicitation technique that specifically focuses on the tail behaviour of the joint distribution in order to determine the adequate copula. Another approach to identify the copula structure is the use of a minimally informative copula with given rank correlation [255]. This approach takes the copula that is minimally informative with respect to the uniform copula subject to the constraints provided by the forecasters. This research has been expanded further by Bedford *et al.* [256] and Kotz & Van Dorp [257].

### 5.2.1.3. Normal posterior

Another approach to combine continuous probability forecasts has been put forth by Winkler [56]. It assumes that the consensus distribution, i.e. the joint posterior in the case of a flat prior density, is the density function of the estimated errors. If the decision maker has a non-diffuse prior density, then the posterior distribution of interest is the product of the decision maker's prior and the density of estimated errors, which is comparable to Morris [242].

One downside of this approach is its restriction on the shape of the posterior distribution. Assuming normality of each forecaster's judgement and its errors, implies that the posterior distribution is going to be normal as well, which might not be realistic for many forecasting problems. Furthermore, the covariance between forecasters needs to be estimated using past data [258].

### 5.2.1.4. Maximum entropy combination

Maximum entropy and minimum cross-entropy methods have had a large share of literature coverage and particularly in the assignment of prior probabilities in decision analysis using partial information (e.g. [17,18,259–264]). The entropy measure attains its maximum when all outcomes are equally likely. When more information is available, the entropy decreases and reaches zero when full knowledge is achieved. The minimum cross-entropy approach finds a distribution that satisfies some given constraints and is closest to a target distribution according to the Kullback–Leibler divergence.

The idea behind this approach is to incorporate only available information and not making any assumptions about unknowns [260]. The joint probability is constructed from a set of known constraints, such as expected forecasts, expected forecasting performance based on past data, and expected correlations between forecasters. Agmon *et al.* [265] developed an algorithm that finds the joint probability function with maximum entropy under given constraints, in almost every case. Once the likelihood function is found, the posterior probability function can be determined by Bayes rule. The advantages of this approach are that no additional assumptions are imposed, and that dependence and past forecasting performance can be incorporated in the constraint set [259]. Its main drawback is computational tractability [265]. In some cases, the likelihood function might not possess a closed-form solution and can only be derived numerically.

### 5.2.2. Non-Bayesian combination methods

The most widely used methods for aggregating human belief are linear and log-linear opinion pools. Although Bayesian approaches provide a normative framework for aggregating forecaster judgements, they are less preferred to non-Bayesian methods because of the inherent difficulty of determining the likelihood function. This section provides an expanded overview of opinion pools and research that has been done to improve their forecasting accuracy. Studies also investigated the appropriate choice of pooling methods (e.g. [266]) and interpretations of the pooling methods from a Kullback–Leibler divergence perspective (e.g. [263]).

### 5.2.2.1. Linear opinion pools

Named by Stone [267], this method combines human judgement by calculating the arithmetic mean of assigned probabilities. Davis-Stober *et al.* [268] have shown that a linear combination of a group is usually more accurate than the judgement of one, randomly selected, member. In its most simple case, all forecasters are assigned equal weight, which makes the model susceptible to malicious or uninformed forecasters [268]. Reputation or past performance of the individual forecasters can be considered by assigning different weights. Several authors proposed performance-based linear opinion pools. Winkler & Clemen [269] have shown that by only considering high-performing forecasters and taking their average, the overall forecasting error can be reduced significantly. Budescu & Chen [270] and

Chen et al. [39] have proposed a so-called 'contribution-weighted model' or CWM, which determines the individual forecaster's weight according to how much he or she has contributed to accuracy of previous forecasting problems. This model does not only filter out badly performing forecasters, it can also assign higher weights depending on a performance measure. Empirically, the CWM approach has been proven to be more accurate and robust than simple averaging [39]. Another approach to account for the past performance of and dependence between individual forecasters, was put forth by Karvetski et al. [271]. Assigning weights according to how well the forecaster's judgement conforms to axioms of probability calculus, forecasting accuracy was improved by 30% coherence compared over linear pools with equal weights [271,272].

A linear opinion pool that incorporates a dependence measure to select and aggregate forecasters has been proposed by Morales-Nápoles and Worm [273]. The dependence calibration score uses a Hellinger distance to assess the proximity between a calibration and forecaster distribution, which was elaborated by Abou-Moustafa et al. [274], and offered a closer examination of distance measures for Gaussian distributions. In a follow-up study, Morales-Nápoles et al. [275] used a Hellinger distance to compare a Gumbel copula with a copula generated from forecasters' assessments of tail dependence. Aggregating the forecasters based on this scoring rule, by allocating higher weights to better calibrated forecasters, outperformed individual forecasters. Turner et al. [276] studied recalibration to reduce the impact of systematic biases during judgement and elicitation and found that recalibrating the individual judgements and then averaging them in log-odds produced a significant improvement in Brier score.

Jose et al. [277] introduced so-called trimmed opinion pools in order to address the calibration and overconfidence of forecasters. The trimming of forecaster judgement has yielded improvements in forecasting accuracy as well. Davis-Stober et al. [278] showed how to derive the individual weights that should be assigned to each human forecaster, depending on the particular source of individual forecasting (in)accuracy, diversity of individual forecasts, and overall group size. In particular, they showed that for large forecasting groups there exists a trade-off between diversity of individual forecasts and forecaster accuracy when one aims to determine the optimal composition of the group, i.e. the weight of every individual forecaster [278]. Kaplan [279] advocates determining the weight of every forecaster based on his or her amount of available information, and not on the ability to encode belief into a forecast. Problems from the linear combination of forecasts, such as inconsistent evaluations or abstaining forecasters have been discussed in Predd et al. [280].

### 5.2.2.2. Log-linear opinion pools

The idea behind log-linear opinion pools is similar to the linear version, with the difference being multiplicative, instead of additive, averaging [175]. Similar to the linear opinion pool, a problem with this combination method is its implicit assumption of independence between forecasters, which might not hold as forecasters might draw from the same source of information. Furthermore, as Abbas [263] and Etiminani et al. [165] have pointed out, the joint probability might result in a value of zero, if one forecaster assigns a zero probability to an event. To avoid this problem, forecasts of 0 may be replaced by an arbitrarily small value $\varepsilon$.

### 5.2.2.3. Democratic opinion pools

The approach developed by Etiminani et al. [165] was used to combine the parameters of Bayesian networks. The algorithm forms clusters of the forecasters' judgements and determines the cluster containing the most forecasters. Once the largest group is identified, a linear combination of their judgements is applied. Because of the reduction in the number of forecasters contributing to the joint judgement, the authors claim that the algorithm is superior when it comes to speed [165]. Their second claim, that the accuracy of the resulting forecast is higher, could be questionable as the algorithm might not only eliminate malicious and poor forecasters, but also independent diverging judgements that would balance some errors and biases.

### 5.2.3. Discussion of accuracy of different mathematical combination methods

Several papers have shown that combining model forecasts improves forecast accuracy relative to a forecast provided by one forecaster [104,281,282]. Clemen & Winkler [8] and Newbold & Granger [283] suggest that comparatively simple averaging methods that ignore correlations in their estimation can improve the accuracy of forecasts. Similarly, Hendry & Clements [281] claim that

simple averaging often outperforms sophisticated models. In contrast to the widely held belief that (log-) linear pools are adequate methods to combine individual forecasts, Wilson [284] found empirical evidence that Bayesian approaches are more accurate if there is dependence among forecasters. Morris [242] and Werner *et al*. [243] both highlight the necessity to incorporate dependence in elicitation and combination of judgement. Bunn [285], Goodwin [286] and Davis-Stober *et al*. [268] found that the combination of judgement using a mathematical approach is most effective when the forecasts are negatively correlated.

Clearly, there seems to be a trade-off between forecasting accuracy and the complexity of the algorithm used to combine individual forecasts. Whereas methods using linear combinations of individual forecasts are computationally easy, the marginal contribution of each additional forecaster decreases if they are correlated [287]. On the other hand, Bayesian approaches offer a normative framework and are deemed more accurate while being difficult to implement. Procedures that combine forecasts by applying the maximum entropy principle have not been studied extensively but deserve more attention because of their freedom from imposed assumptions.

# 6. Summary and conclusion

The purpose of this survey was to provide a high-level overview of topics and methods relevant to the ever-growing domain of forecasting. It was also the intention to improve mutual understanding and to foster dialogue between academic disciplines engaged in this field.

Taking the number of publications as an indicator, it is clear that interest in forecasting techniques and applications has increased by a factor of 10 over the previous 40 years. With the increase of quantitative forecasting models came an interest in the performance difference between human judgement and quantitative forecasting methods, their limits, and benefits of combining them. The promise of combining human judgement and quantitative methods lies in the mutual balancing of their strengths and limitations. Human forecasters are able to use contextual data to inform their judgement, something that purely quantitative models are not capable of if the context is not represented in the dataset. Also, humans tend to outperform the accuracy of quantitatively generated forecasts if data are sparse. Contrastingly, quantitative techniques can survey and learn from vast datasets that would overwhelm human cognitive abilities. In medicine for example, the combination of human judgement and a classification derived by a neural network resulted in a 99.5% accuracy for detecting cancerous growth, a 85% decrease of human error rate [288].

We can summarize the results of our review by several stylized points.

*Choice of appropriate quantitative model:* We reviewed the most common quantitative models that are being applied in the forecasting domain, differing in their area of applicability and construction difficulty. In this survey, we distinguished quantitative models into univariate, explanatory and overarching methods. Univariate methods both comprise time-series and frequency-domain methods and make a prediction solely on the basis of data on the quantity to be forecast. The explanatory methods discussed in this review comprise regression models and support vector machines. To make a prediction, these methods draw on data about the quantity to be forecast and other predictor variables. Overarching methods, including neural networks, Bayesian networks, ensemble methods and simulation, are approaches that can be used both for univariate and explanatory purposes. Therefore, a forecaster or decision maker needs to consider the forecasting problem and available resources when deciding on an appropriate quantitative model. The choice of model is also dependent on the availability and quality of data. We found that what constitutes an appropriate quantitative model differs across academic disciplines. Researchers in the computer science and operations research domains point to the successes of advanced machine learning models, such as neural networks, when promoting their use in forecasting. Critics of these models emphasize that the underlying mechanisms use statistical methods to discover patterns in data without being grounded in theory and prior knowledge on cause and effect, thereby rendering them arbitrary and invalid for the use in forecasting. Given the practical evidence of their successful application, we abstain from labelling more advanced quantitative methods as invalid. Instead, we emphasize the need for dialogue between the fields partaking in the domain of forecasting. Such a dialogue could create the basis for mutual understanding and common terminology, something that appeared to be lacking from the publications surveyed in this review. The creation of advanced forecasting algorithms for example might benefit from the axiomatic foundations derived by decision theorists and the experimental data generated by behavioural scientists.

*Interaction between quantitative models and human forecasters:* There has been significant research on the interaction between quantitative models and humans, such as the effect of past performance on humans' readiness to use the provided information. Behavioural studies have shown that human forecasters exhibit a higher readiness to use recommendations provided by models if they understand how they work and if the output can be adjusted. These insights form the basis for two disparate sets of beliefs regarding the use of advanced quantitative models. One argues that because these methods, including neural networks, violate the axiom of using simple and easy-to-understand models, they should not be used at all. Proponents of this view emphasize theoretical findings that highlight the low performance of these complex methods and the reluctance of human forecasters to use them. Other researchers subscribe to the belief that the aversion towards these models can be overcome by adequate training. They reason that explaining the methods will improve the trust people put in them. To support this perception, they cite studies showing the improved classification and forecasting accuracy when human judgement is combined with advanced quantitative models.

*Training and incentivization of human forecasters:* Given that human forecasters' readiness to consider recommendations generated by quantitative models depends on their understanding, training them to better comprehend these models appears logical. Surveying the literature, we did not find extensive work on how to best train human forecasters for the use of quantitative models. When looking at the effects of training on forecasting accuracy of human forecasters in isolation, the results were mixed. Some studies pointed to the benefits of training human forecasters in probability theory and de-biasing techniques, while others did not find a significant influence of these measures on forecasting accuracy. A similar observation applies to the use of incentives.

*Combination of judgements and forecasts:* There is no consensus in the scientific community about which method is the most accurate or efficient. On the one hand, non-Bayesian approaches are very appealing to users, as they are computationally easy and have been used in many research publications. A multitude of non-Bayesian combination rules, such as contribution weights or democratic opinion pools, were devised to improve accuracy since the last comprehensive survey on the combination of judgements. On the other hand, Bayesian approaches are normative, but are usually bugged down by high computational effort, and in the case of the normal and copula combination method require several assumptions to be functional. This review also added the combination of forecasts employing maximum entropy.

We conclude by identifying several directions that are ripe for future research. Most areas identified as important to human/machine-forecasting have been researched in depth, but the interactions between them require more attention. For example, there has to be an assessment of quantitative models in a holistic context, i.e. which model is most suitable given a certain training of human forecasters under a certain incentive scheme and a specific combination method. Furthermore, the question of how much weight or significance should be assigned to quantitative models and how much to the human forecasters has to be discussed and addressed. Answering this question has clear implications for forecasting accuracy, and it also carries ethical significance. Methods to combine forecasts and judgements should be discussed in conjunction with different scoring rules, especially because some approaches determine the weighting of forecasters by their individual scores. In the domain of judgemental forecasting, methods need to be developed and further researched that support judgemental forecasters, including decomposition, guidance, and the identification and use of analogies. Some researchers have already begun determining an axiomatic approach to choosing and developing forecasting approaches. We consider this undertaking as very important and believe it could benefit from a dialogue between the different domains investigating the area of forecasting.

Data accessibility. This article has no additional data.

Authors' contributions. M.Z. carried out the web scraping and literature search, reviewed the identified publications for relevance, aggregated the sources, structured and drafted the manuscript. He also implemented the edits and suggestions made by his fellow authors and reviewers. A.E.A., PhD, conceived the survey paper. He provided guidance regarding the structure of the manuscript, made edits to it, and highlighted relevant work that was not identified during the initial stage of the literature search, especially in the field of forecast combination. D.V.B., PhD, edited and provided guidance regarding the overall structure of the manuscript. He also pointed to relevant information pertaining to psychological and behavioural factors influencing forecasting. A.G., PhD, edited the manuscript, contributed to the structuring of the manuscript, and provided guidance and information pertaining to quantitative forecasting methods, in particular neural networks. All authors gave final approval for publication.

Competing interests. We declare we have no competing interests

Funding. The authors acknowledge the support of the National Science Foundation award INFEWS 17–39551 and IARPA-BAA-16–02.

This research is based upon work supported in part by the Office of the Director of National Intelligence (ODNI), Intelligence Advanced Research Projects Activity (IARPA), via IARPA Contract no. 2017-17071900005. The views and conclusions contained herein are those of the authors and should not be interpreted as necessarily representing the official policies, either expressed or implied, of ODNI, IARPA, or the US Government. The US Government is authorized to reproduce and distribute reprints for governmental purposes notwithstanding any copyright annotation therein.

Acknowledgement. We acknowledge the support and feedback of five anonymous reviewers/referees. Their effort improved the quality of the paper substantially.

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
