## [Peer Review File · Royal Society Open Science]

Review History

RSOS-192145.R0 (Original submission)

Review form: Reviewer 1

Is the manuscript scientifically sound in its present form?

No

Are the interpretations and conclusions justified by the results?

No

Is the language acceptable?

Yes

Do you have any ethical concerns with this paper?

No

Have you any concerns about statistical analyses in this paper?

Yes

Recommendation?

Major revision is needed (please make suggestions in comments)

Comments to the Author(s)

Detailed comments are in the attached file (Appendix A).

The survey confuses confuses atheoretical machine learning methods with validated quantitative methods that use theory and evidence from experiments to specify models. The confusion leads to misleading conclusions.

The survey misses key papers providing evidence from comparative studies on which methods do and do not reduce forecast errors. In particular the recent comprehensive review by Armstrong & Green (2018) "Forecasting methods and principles: Evidence-based checklists" would be a good place to start in improving the survey (using snow-balling perhaps) and for identifying the terms used in the forecasting literature for searches for papers on applications that use evidence-based methods.

Review form: Reviewer 2

Is the manuscript scientifically sound in its present form?

No

Are the interpretations and conclusions justified by the results?

Yes

Is the language acceptable?

Yes

Do you have any ethical concerns with this paper?

No

Have you any concerns about statistical analyses in this paper?

No

Recommendation?

Reject

Comments to the Author(s)

The forecasting field is vast, in terms of both its range of applications and its techniques. Producing a survey that covers this huge area is therefore challenging and involves difficult judgments on which topics to include and which to emphasize. It also requires a clarity of purpose and careful structuring, enabling key findings to be synthesized and new arguments to emerge. While this paper provides an excellent documentation of how relevant papers were identified and selected for the review, I have a number of concerns about its contribution to the literature. These relate to its structure, its clarity, its coverage, and the extent to which some of its conclusions are novel.

First, I found the paper difficult to navigate given the way it is structured. It might be better to delineate forecasting tasks such as point forecasting based on time series data, point forecasting when contextual data is available, prediction interval formation, density forecasting, event forecasting using probabilities and, for each of these tasks in turn, to identify the strengths and limitations of human, machine and combination forecasting and how each approach is best applied where it is appropriate. For example, this could include methods for improving

judgmental forecasts (including possibly decomposition and feedback) in relation to each task. A task-orientated structure is especially appropriate for human judgment because its effectiveness is known to be highly sensitive to the nature of the task. Currently, each section mixes tasks, strengths and weaknesses and improvement strategies so it is difficult to see which strategy might be appropriate for a given task. For example, on page 12 the paper refers to Sanders and Ritzman's finding that training forecasters in gathering and handling contextual data was more beneficial than training them in technical and statistical aspects of forecasting -but their paper only related to point forecasting based on time series data. Would this finding also apply to probability forecasters? Indeed, a task-orientated structure would make it easier to contrast human, machine and combination forecasting in relation to each specific task. In addition, some discussion appears to be in the wrong section. For example, in section 3 on individual human forecasting there is reference at the start of in section 3.3.1 to aggregating opinions. Section 4 on machine forecasting, refers to judgmental time series forecasting -citing Goodwin and Wright (1993). On page 23 SMAPE and MASE are introduced, but shouldn't they, and similar measures, relate to the earlier discussion of incentive schemes? On page 28 much of the discussion of the findings of Onkal et al. 2009 surely belongs to the subsequent section on algorithm aversion.

Second much of the paper reads simply like a catalogue of forecasting methods, and the discussion is sometimes too brief for an uninitiated reader to understand what a method involves. I appreciate that brevity is essential in a review with a scope as wide as this one, but it is difficult to discern what the paper's purpose is. Is it intended to introduce non-specialists to forecasting (e.g. the discussion of moving averages and exponential smoothing would suggest this) or to update specialists -who would already know about moving averages etc. -on the latest findings? The current treatment falls between these two stools.

There are also several areas where the discussion is unclear or contradictory. What is strategic behaviour in relation to probability forecasting? And why would an advice seeker benefit from an incentive scheme that rewards strategic behaviour rather than truthful reporting (page 11 - unless this is a typo). On page 28, line 44 the sentences relating to Onkal et al. (2009) are confusing. On page 31 we are told that "Opinions and advice originating from highly correlated sources are unlikely to improve forecasting accuracy", but then "as long as correlation between forecasters is not perfectly positive, adding more forecasters increases forecasting accuracy". On page 31, you state that you have excluded behavioral aggregation from your survey, but surely this has already been included in section 3.2?

It is of course easy when reviewing a paper that is as ambitious in its scope as this paper to argue that other works and topics should be included. However, given the importance of weather forecasting, I think that ensemble forecasting should be discussed. On the integration of machine and human forecasting you could consider correction methods (e.g. Theil's method -see Ahlburg, 1984) where a machine forecasts the errors of a human forecaster and then corrects their forecasts accordingly. I would also like to see some discussion of whether it is better to allow a human to adjust a machine forecast or simply to aggregate human and machine forecasts mechanically (e.g. by taking a simple average). Recent work on judgmental selection of machine methods (Petropoulos et al. 2018, De Baets and Harvey 2020) might also be worth including, though I appreciate the dates of your literature search precluded the selection of these papers.

Finally, I have doubts about the novelty of the paper's findings that (1) neither human or machine forecasting is universally superior, and (2) the better method varies as a function of factors such as availability, quality, extent, and format of data, suggesting that (3) the two approaches can complement each other to yield more accurate and resilient models. (1) and (2) are self-evident and (3) was highlighted as early as 1990 in the paper by Blattberg and Hoch.

Minor points

Do figures 3 and 4 include overlapping categories?

Is there a journal called Judgmental Forecasting as Table II suggests or is this perhaps the edited book by Wright and Ayton?

Section 4.2.1 Are regression models always very simple?

Page 39, line 43 Are machine models always incapable of using contextual data?

There are lots of typos in the manuscript. Please proof read it

References

Ahlburg, D. A. (1984). Forecasting evaluation and improvement using Theil's decomposition. *Journal of Forecasting*, 3, 345- 351.

De Baets, S., & Harvey, N. (2020). Using judgment to select and adjust forecasts from statistical models. *European Journal of Operational Research*.

Petropoulos, F., Kourentzes, N., Nikolopoulos, K., & Siemsen, E. (2018). Judgmental selection of forecasting models. *Journal of Operations Management*, 60, 34-46.

Review form: Reviewer 3

Is the manuscript scientifically sound in its present form?

Yes

Are the interpretations and conclusions justified by the results?

Yes

Is the language acceptable?

Yes

Do you have any ethical concerns with this paper?

No

Have you any concerns about statistical analyses in this paper?

No

Recommendation?

Accept with minor revision (please list in comments)

Comments to the Author(s)

Please see attached file (Appendix B).

Review form: Reviewer 4

Is the manuscript scientifically sound in its present form?

Yes

Are the interpretations and conclusions justified by the results?

Yes

Is the language acceptable?

Yes

Do you have any ethical concerns with this paper?

No

Have you any concerns about statistical analyses in this paper?

No

Recommendation?

Accept with minor revision (please list in comments)

Comments to the Author(s)

This paper presents a critical review of extant work on human, machine, and hybrid forecasting, as well as forecast aggregation. It fits with the journal's scope on reviews of multidisciplinary topics as it provides an extensive summary of the state-of-the-art in forecasting, explores developments in this field and points to promising research avenues/thrusts.

Overall, I believe that the paper provides a detailed review of an important multidisciplinary area with strong implications across a wide variety of domains. It is a very long review, as it aims to be all-encompassing in its scope. Although the length of the paper may not be problematic from the journal's perspective, it may be distracting from the reader's perspective, which may be worth considering. Moving certain tangential subsections into Appendices or use of footnotes could be alternatives, but there could be others.

My comments are as follows:

1. As this is a review of forecasting, I would urge the authors to only cite the work that actually requires making forecasts, instead of involving estimation tasks. As has been shown in repeated studies, people's responses to general knowledge tasks, for example, cannot be generalized into the forecasting domain. This is relevant for probability elicitation section in the paper, as well as the sections on Delphi and advice taking, among potentially others.
2. What would set this paper apart would be the final section. In order for this not to feel like a literature review for a dissertation work, it would be fundamental to include (i) a critical overarching evaluation across the findings, and (ii) further venues for promising research. The paper does this to a limited extent and it would be highly valuable to expand on this final section (e.g., to include practical implications across sectors).

Review form: Reviewer 5

Is the manuscript scientifically sound in its present form?

Yes

Are the interpretations and conclusions justified by the results?

Yes

Is the language acceptable?

Yes

Do you have any ethical concerns with this paper?

No

Have you any concerns about statistical analyses in this paper?

No

Recommendation?

Accept with minor revision (please list in comments)

Comments to the Author(s)

This paper reviews forecasting methods, including both human and machine based methods. The relatively wide scope of the review differentiates it from other reviews that consider only human methods or machine methods. Overall, I find the review to be well done. I particularly appreciate the charts that illustrate the growing importance of this field of study and the attention paid to Bayesian methods. I recommend the paper be accepted with minor revision.

Although the paper is generally well written, there are some instances in which past tense is used inappropriately. For example, in the abstract, "The survey started with..." should be revised to, "The survey starts with..." I would suggest the authors take a final look at the manuscript and make minor grammatical and compositional improvements as needed.

Decision letter (RSOS-192145.R0)

02-Mar-2020

Dear Mr Zellner:

Manuscript ID RSOS-192145 entitled "A Survey of Human and Machine Forecasting Methods" which you submitted to Royal Society Open Science, has been reviewed. The comments from reviewers are included at the bottom of this letter.

In view of the criticisms of the reviewers, the manuscript has been rejected in its current form. However, a new manuscript may be submitted which takes into consideration these comments.

Please note that resubmitting your manuscript does not guarantee eventual acceptance, and that your resubmission will be subject to peer review before a decision is made.

Your resubmitted manuscript should be submitted by 30-Aug-2020. If you are unable to submit by this date please contact the Editorial Office.

on behalf of R. Kerry Rowe (Subject Editor)
openscience@royalsociety.org

Associate Editor Comments to Author:

Comments to the Author:

Thank you for the contribution. We've received a larger than usual number of reviewer reports on your paper, and though several of the reviewers nominally recommend acceptance with minor revision, given the concerns, comments and queries from the first three reviewers, we feel it would be better to give you the opportunity to address these points in a thorough revision. As our 'major revision' decision would only allow 3 weeks to revise, the 'reject and resubmit' decision will allow you several months to consider and make the changes recommended by the reviewers. Bear in mind that you will need to provide a marked up version of the manuscript highlighting the changes made, and also a point-by-point response when you resubmit. It will also remain at the discretion of the Editors whether to return your manuscript to any of the existing reviewers. Thanks again for your support and good luck.

Reviewers' Comments to Author:

Reviewer: 1

Comments to the Author(s)

Detailed comments are in the attached file... (RSOS-192145_Proof_hi-Comments.pdf)

The survey confuses confuses atheoretical machine learning methods with validated quantitative methods that use theory and evidence from experiments to specify models. The confusion leads to misleading conclusions.

The survey misses key papers providing evidence from comparative studies on which methods do and do not reduce forecast errors. In particular the recent comprehensive review by Armstrong & Green (2018) "Forecasting methods and principles: Evidence-based checklists" would be a good place to start in improving the survey (using snow-balling perhaps) and for identifying the terms used in the forecasting literature for searches for papers on applications that use evidence-based methods.

Reviewer: 2

Comments to the Author(s)

The forecasting field is vast, in terms of both its range of applications and its techniques. Producing a survey that covers this huge area is therefore challenging and involves difficult judgments on which topics to include and which to emphasize. It also requires a clarity of purpose and careful structuring, enabling key findings to be synthesized and new arguments to emerge. While this paper provides an excellent documentation of how relevant papers were identified and selected for the review, I have a number of concerns about its contribution to the literature. These relate to its structure, its clarity, its coverage, and the extent to which some of its conclusions are novel.

First, I found the paper difficult to navigate given the way it is structured. It might be better to delineate forecasting tasks such as point forecasting based on time series data, point forecasting when contextual data is available, prediction interval formation, density forecasting, event forecasting using probabilities and, for each of these tasks in turn, to identify the strengths and limitations of human, machine and combination forecasting and how each approach is best applied where it is appropriate. For example, this could include methods for improving judgmental forecasts (including possibly decomposition and feedback) in relation to each task. A task-orientated structure is especially appropriate for human judgment because its effectiveness is known to be highly sensitive to the nature of the task. Currently, each section mixes tasks, strengths and weaknesses and improvement strategies so it is difficult to see which strategy might be appropriate for a given task. For example, on page 12 the paper refers to Sanders and Ritzman's finding that training forecasters in gathering and handling contextual data was more beneficial than training them in technical and statistical aspects of forecasting -but their paper only related to point forecasting based on time series data. Would this finding also apply to probability forecasters? Indeed, a task-orientated structure would make it easier to contrast human, machine and combination forecasting in relation to each specific task. In addition, some discussion appears to be in the wrong section. For example, in section 3 on individual human forecasting there is reference at the start of in section 3.3.1 to aggregating opinions. Section 4 on machine forecasting, refers to judgmental time series forecasting -citing Goodwin and Wright (1993). On page 23 SMAPE and MASE are introduced, but shouldn't they, and similar measures, relate to the earlier discussion of incentive schemes? On page 28 much of the discussion of the findings of Onkal et al. 2009 surely belongs to the subsequent section on algorithm aversion.

Second much of the paper reads simply like a catalogue of forecasting methods, and the discussion is sometimes too brief for an uninitiated reader to understand what a method involves. I appreciate that brevity is essential in a review with a scope as wide as this one, but it is difficult to discern what the paper's purpose is. Is it intended to introduce non-specialists to forecasting (e.g. the discussion of moving averages and exponential smoothing would suggest this) or to update specialists -who would already know about moving averages etc. -on the latest findings? The current treatment falls between these two stools.

There are also several areas where the discussion is unclear or contradictory. What is strategic behaviour in relation to probability forecasting? And why would an advice seeker benefit from an incentive scheme that rewards strategic behaviour rather than truthful reporting (page 11 - unless this is a typo). On page 28, line 44 the sentences relating to Onkal et al. (2009) are confusing. On page 31 we are told that "Opinions and advice originating from highly correlated sources are unlikely to improve forecasting accuracy", but then "as long as correlation between forecasters is not perfectly positive, adding more forecasters increases forecasting accuracy". On page 31, you state that you have excluded behavioral aggregation from your survey, but surely this has already been included in section 3.2?

It is of course easy when reviewing a paper that is as ambitious in its scope as this paper to argue that other works and topics should be included. However, given the importance of weather forecasting, I think that ensemble forecasting should be discussed. On the integration of machine and human forecasting you could consider correction methods (e.g. Theil's method -see Ahlburg, 1984) where a machine forecasts the errors of a human forecaster and then corrects their forecasts accordingly. I would also like to see some discussion of whether it is better to allow a human to adjust a machine forecast or simply to aggregate human and machine forecasts mechanically (e.g. by taking a simple average). Recent work on judgmental selection of machine methods (Petropoulos et al. 2018, De Baets and Harvey 2020) might also be worth including, though I appreciate the dates of your literature search precluded the selection of these papers.

Finally, I have doubts about the novelty of the paper's findings that (1) neither human or machine forecasting is universally superior, and (2) the better method varies as a function of factors such as availability, quality, extent, and format of data, suggesting that (3) the two approaches can complement each other to yield more accurate and resilient models. (1) and (2) are self-evident and (3) was highlighted as early as 1990 in the paper by Blattberg and Hoch.

Minor points

Do figures 3 and 4 include overlapping categories?

Is there a journal called Judgmental Forecasting as Table II suggests or is this perhaps the edited book by Wright and Ayton?

Section 4.2.1 Are regression models always very simple?

Page 39, line 43 Are machine models always incapable of using contextual data?

There are lots of typos in the manuscript. Please proof read it

References

Ahlburg, D. A. (1984). Forecasting evaluation and improvement using Theil's decomposition. *Journal of Forecasting*, 3, 345- 351.

De Baets, S., & Harvey, N. (2020). Using judgment to select and adjust forecasts from statistical models. *European Journal of Operational Research*.

Petropoulos, F., Kourentzes, N., Nikolopoulos, K., & Siemsen, E. (2018). Judgmental selection of forecasting models. *Journal of Operations Management*, 60, 34-46.

Reviewer: 3

Comments to the Author(s)

Please see attached file (Review Abbas et al 2020.pdf)

Reviewer: 4

Comments to the Author(s)

This paper presents a critical review of extant work on human, machine, and hybrid forecasting, as well as forecast aggregation. It fits with the journal's scope on reviews of multidisciplinary topics as it provides an extensive summary of the state-of-the-art in forecasting, explores developments in this field and points to promising research avenues/thrusts.

Overall, I believe that the paper provides a detailed review of an important multidisciplinary area with strong implications across a wide variety of domains. It is a very long review, as it aims to be all-encompassing in its scope. Although the length of the paper may not be problematic from the journal's perspective, it may be distracting from the reader's perspective, which may be worth considering. Moving certain tangential subsections into Appendices or use of footnotes could be alternatives, but there could be others.

My comments are as follows:

1. As this is a review of forecasting, I would urge the authors to only cite the work that

actually requires making forecasts, instead of involving estimation tasks. As has been shown in repeated studies, people's responses to general knowledge tasks, for example, cannot be generalized into the forecasting domain. This is relevant for probability elicitation section in the paper, as well as the sections on Delphi and advice taking, among potentially others.

2. What would set this paper apart would be the final section. In order for this not to feel like a literature review for a dissertation work, it would be fundamental to include (i) a critical overarching evaluation across the findings, and (ii) further venues for promising research. The paper does this to a limited extent and it would be highly valuable to expand on this final section (e.g., to include practical implications across sectors).

Reviewer: 5

Comments to the Author(s)

This paper reviews forecasting methods, including both human and machine based methods. The relatively wide scope of the review differentiates it from other reviews that consider only human methods or machine methods. Overall, I find the review to be well done. I particularly appreciate the charts that illustrate the growing importance of this field of study and the attention paid to Bayesian methods. I recommend the paper be accepted with minor revision.

Although the paper is generally well written, there are some instances in which past tense is used inappropriately. For example, in the abstract, "The survey started with..." should be revised to, "The survey starts with..." I would suggest the authors take a final look at the manuscript and make minor grammatical and compositional improvements as needed.

Author's Response to Decision Letter for (RSOS-192145.R0)

See Appendices C & D.

RSOS-201187.R0

Review form: Reviewer 1

Is the manuscript scientifically sound in its present form?

No

Are the interpretations and conclusions justified by the results?

No

Is the language acceptable?

No

Do you have any ethical concerns with this paper?

No

Have you any concerns about statistical analyses in this paper?

No

Recommendation?

Accept with minor revision (please list in comments)

Comments to the Author(s)

Might the paper be better titled something along the lines of "A survey of trends in publishing on forecasting methods, by application and method name "

Review form: Reviewer 2**Is the manuscript scientifically sound in its present form?**

No

Are the interpretations and conclusions justified by the results?

No

Is the language acceptable?

Yes

Do you have any ethical concerns with this paper?

No

Have you any concerns about statistical analyses in this paper?

No

Recommendation?

Reject

Comments to the Author(s)

This revised paper is improved in some respects -and again I welcome the details of the research methodology. The emphasis on combining forecasts is also appropriate. However, I still have serious concerns about some aspects of the paper's structure and the relative emphasis it places on several forecasting topics, in addition to other issues.

A major structural concern is the placement of subsection 4.1.1 'Regression models' within the section on Frequency Domain models. Frequency domain models are generally used to account for variation in time series through cyclic components at different frequencies. Hence the input variables are typically cosines and sines, but there is no reference to this in the paper. Although least squares estimation can be used to obtain these models, the reader might be led to believe that all regression models are frequency domain, which is very far from the case. A distinction between univariate forecasting methods and explanatory methods -which draw on information from independent variables -would be more appropriate and illuminating than that of frequency versus time domain. I strongly disagree with the statement in the Conclusions (page 42, line 18) that the most common classification distinguishes between time and frequency domain models, and a combination of the two. I certainly cannot understand why logistic regression models appears in the frequency domain section. Again, reading the paper the reader may infer that these models are based on least squares when they are obtained through maximum likelihood estimation. Section 4 would certainly benefit from an introductory statement of what the frequency domain is rather than the vague statement that models based on it "use highly refined and specific information about relationships between system elements..."

Another structural oddity is the inclusion of the following sentence in section 4 'Quantitative Forecasting Methods': "Judgmental time-series models concern humans extrapolating time-series into the future and adjusting the series for contextual data (page 19, line 20). Manifestly, judgmental forecasts are not derived via quantitative methods so why is this sentence placed here? Quantitative models can be applied to judgmental forecasts, for example by using (psychological) bootstrap models, but oddly these are not mentioned at all in the paper. Incidentally, people don't usually adjust the series, they adjust forecasts.

I am also not clear why naïve forecasts are included under simulation models (page 27).

In terms of emphasis - there is no mention of prediction intervals -a major way of expressing forecasts and linear regression models -a widely used forecasting method (even discounting econometric applications) merits less than four lines. Yet there is a whole sub-section on focus groups. Focus groups are not a forecasting method -they are designed 'to explore the dimensions of a topic and the range of conceivable responses rather than achieving a consensus' (see Ord et al, 2017, page 393).

There still tends to be a merging of the discussion of different forecasting tasks which gives the paper a shapeless feel. Several parts of the paper would benefit if the forecasting task that was being referred to was made clearer. On page 34 there is a discussion of forecasts of 'point probabilities'. By point probabilities do you mean probabilities for discrete quantities or events (e.g. the probability that it will rain tomorrow) as opposed to estimates of continuous probability distributions?

Overall, I think the paper would benefit from a tree diagram early on which maps out the different forecasting methods and provides the paper with a clear structure. A clear definition of the task that is being discussed at the start of each section would also be helpful. I found the conclusions to be unexciting and they ignore a major aspect of recent research -the need to develop methods to support judgmental forecasters, such as decomposition, guidance and the identification and use of analogies.

Other points

Please tell the reader that the golden rule of forecasting is on page 1

Page 15, bottom. What do you mean by: "Because surveying the multitude of group judgment is not the focus of this review paper.."

Page 28, line 40. Spelling is: O'Connor.

Page 33, line 7. You surely don't mean the opposite result to that found by Ahlburg?

Page 34, line 33. forecasters (i.e. plural)

Reference

Ord K., Fildes, R. and Kourentzes, N. (2017) Principles of Forecasting 2e, New York: Wessex.

Review form: Reviewer 3

Is the manuscript scientifically sound in its present form?

Yes

Are the interpretations and conclusions justified by the results?

Yes

Is the language acceptable?

Yes

Do you have any ethical concerns with this paper?

No

Have you any concerns about statistical analyses in this paper?

No

Recommendation?

Accept with minor revision (please list in comments)

Comments to the Author(s)

The authors' responses have dealt with the points made by the five reviewers. In my view, there are just a few minor issues that remain to be addressed.

1) Additional panels have been added to Figures 2 and 4. I think that this is because the topics have been divided into time-domain and frequency domain searches. But this is not clear. Two things need to be done. First, add to the figure captions to explain what the upper and lower panels of the figures represent. Second, ensure that the axes in the two panels look the same. For instance, in Figure 2, the numbers on the vertical axis are in different sized typefaces, the labels on those axes are bold in one case and not the other, and the axes have a different range of values in the two cases. In Figure 4, the divisions on the vertical axes are different: 50,000 in the upper panel and 100,000 in the lower one.

2) In the last paragraph on page 14, point forecasts and pdf forecasts are mentioned. It might be worth adding that interval forecasts (without point forecasts) are also not uncommon, especially in economics.

3) At the bottom of page 14, we are told that studies have found no "clear evidence supporting representing data visually instead of in table format representation when eliciting point forecasts". This statement is misleading. For example, Harvey & Bolger (1996) did find clear evidence that graphical presentation is superior when data contain trends (most data sets). (There is also mounting evidence that the type of graphical format matters - e.g., Okan et al, QJEP, 2018 - but that does not need to be mentioned here.)

4) Page 31, line 6: intractable -> opaque

5) Page 31, line 12: impressive but it -> impressive, it

Decision letter (RSOS-201187.R0)

Dear Mr Zellner

The Editors assigned to your paper RSOS-201187 "A Survey of Human Judgment and Quantitative Forecasting Methods" have now received comments from reviewers and would like you to revise the paper in accordance with the reviewer comments and any comments from the Editors. Please note this decision does not guarantee eventual acceptance.

Please submit your revised manuscript and required files (see below) no later than 21 days from today's (ie 29-Sep-2020) date. Note: the ScholarOne system will 'lock' if submission of the revision is attempted 21 or more days after the deadline. If you do not think you will be able to meet this deadline please contact the editorial office immediately.

on behalf of the Associate Editor and Professor R. Kerry Rowe (Subject Editor)
openscience@royalsociety.org

Associate Editor Comments to Author:

Your paper presents the editors with something of a challenge. While it seems the paper is improved on the initial submission, one of the reviewers continues to have substantial concerns with it. We're going to give you the benefit of the doubt and allow a further round of review to allow you the opportunity to satisfy the most critical of the reviewers that your paper should be considered ready for acceptance. Please carefully and clearly respond to the remaining concerns of the reviewers - as the journal does not permit multiple rounds of revision, this will be considered your final opportunity to revise the paper.

Reviewer comments to Author:

Reviewer: 1

Comments to the Author(s)

Might the paper be better titled something along the lines of "A survey of trends in publishing on forecasting methods, by application and method name "

Reviewer: 2

Comments to the Author(s)

This revised paper is improved in some respects -and again I welcome the details of the research methodology. The emphasis on combining forecasts is also appropriate. However, I still have serious concerns about some aspects of the paper's structure and the relative emphasis it places on several forecasting topics, in addition to other issues.

A major structural concern is the placement of subsection 4.1.1 'Regression models' within the section on Frequency Domain models. Frequency domain models are generally used to account for variation in time series through cyclic components at different frequencies. Hence the input variables are typically cosines and sines, but there is no reference to this in the paper. Although least squares estimation can be used to obtain these models, the reader might be led to believe that all regression models are frequency domain, which is very far from the case. A distinction between univariate forecasting methods and explanatory methods -which draw on information from independent variables -would be more appropriate and illuminating than that of frequency versus time domain. I strongly disagree with the statement in the Conclusions (page 42, line 18) that the most common classification distinguishes between time and frequency domain models, and a combination of the two. I certainly cannot understand why logistic regression models appears in the frequency domain section. Again, reading the paper the reader may infer that these models are based on least squares when they are obtained through maximum likelihood estimation. Section 4 would certainly benefit from an introductory statement of what the frequency domain is rather than the vague statement that models based on it "use highly refined and specific information about relationships between system elements..."

Another structural oddity is the inclusion of the following sentence in section 4 'Quantitative Forecasting Methods': "Judgmental time-series models concern humans extrapolating time-series into the future and adjusting the series for contextual data (page 19, line 20). Manifestly, judgmental forecasts are not derived via quantitative methods so why is this sentence placed here? Quantitative models can be applied to judgmental forecasts, for example by using (psychological) bootstrap models, but oddly these are not mentioned at all in the paper. Incidentally, people don't usually adjust the series, they adjust forecasts.

I am also not clear why naïve forecasts are included under simulation models (page 27).

In terms of emphasis - there is no mention of prediction intervals -a major way of expressing forecasts and linear regression models -a widely used forecasting method (even discounting econometric applications) merits less than four lines. Yet there is a whole sub-section on focus groups. Focus groups are not a forecasting method -they are designed 'to explore the dimensions of a topic and the range of conceivable responses rather than achieving a consensus' (see Ord et al, 2017, page 393).

There still tends to be a merging of the discussion of different forecasting tasks which gives the paper a shapeless feel. Several parts of the paper would benefit if the forecasting task that was being referred to was made clearer. On page 34 there is a discussion of forecasts of 'point probabilities'. By point probabilities do you mean probabilities for discrete quantities or events

(e.g. the probability that it will rain tomorrow) as opposed to estimates of continuous probability distributions?

Overall, I think the paper would benefit from a tree diagram early on which maps out the different forecasting methods and provides the paper with a clear structure. A clear definition of the task that is being discussed at the start of each section would also be helpful. I found the conclusions to be unexciting and they ignore a major aspect of recent research -the need to develop methods to support judgmental forecasters, such as decomposition, guidance and the identification and use of analogies.

Other points

Please tell the reader that the golden rule of forecasting is on page 1

Page 15, bottom. What do you mean by: "Because surveying the multitude of group judgment is not the focus of this review paper.."

Page 28, line 40. Spelling is: O'Connor.

Page 33, line 7. You surely don't mean the opposite result to that found by Ahlburg?

Page 34, line 33. forecasters (i.e. plural)

Reference

Ord K., Fildes, R. and Kourentzes, N. (2017) *Principles of Forecasting 2e*, New York: Wessex.

Reviewer: 3

Comments to the Author(s)

The authors' responses have dealt with the points made by the five reviewers. In my view, there are just a few minor issues that remain to be addressed.

1) Additional panels have been added to Figures 2 and 4. I think that this is because the topics have been divided into time-domain and frequency domain searches. But this is not clear. Two things need to be done. First, add to the figure captions to explain what the upper and lower panels of the figures represent. Second, ensure that the axes in the two panels look the same. For instance, in Figure 2, the numbers on the vertical axis are in different sized typefaces, the labels on those axes are bold in one case and not the other, and the axes have a different range of values in the two cases. In Figure 4, the divisions on the vertical axes are different: 50,000 in the upper panel and 100,000 in the lower one.

2) In the last paragraph on page 14, point forecasts and pdf forecasts are mentioned. It might be worth adding that interval forecasts (without point forecasts) are also not uncommon, especially in economics.

3) At the bottom of page 14, we are told that studies have found no "clear evidence supporting representing data visually instead of in table format representation when eliciting point forecasts". This statement is misleading. For example, Harvey & Bolger (1996) did find clear evidence that graphical presentation is superior when data contain trends (most data sets). (There is also mounting evidence that the type of graphical format matters - e.g., Okan et al, QJEP, 2018 - but that does not need to be mentioned here.)

4) Page 31, line 6: intractable -> opaque

5) Page 31, line 12: impressive but it -> impressive, it

===PREPARING YOUR MANUSCRIPT===

===PREPARING YOUR REVISION IN SCHOLARONE===

<https://royalsociety.org/journals/authors/author-guidelines/#supplementary-material> to include a suitable title and informative caption. An example of appropriate titling and captioning may be found at https://figshare.com/articles/Table_S2_from_Is_there_a_trade-off_between_peak_performance_and_performance_breadth_across_temperatures_for_aerobic_sc_ope_in_teleost_fishes_/3843624.

Author's Response to Decision Letter for (RSOS-201187.R0)

See Appendix E.

RSOS-201187.R2 (Revision)

Review form: Reviewer 1

Is the manuscript scientifically sound in its present form?

Yes

Are the interpretations and conclusions justified by the results?

Yes

Is the language acceptable?

Yes

Do you have any ethical concerns with this paper?

No

Have you any concerns about statistical analyses in this paper?

No

Recommendation?

Accept with minor revision (please list in comments)

Comments to the Author(s)

p.2: Suggest revising along the lines of... "Critics of this view point out that the use of machine learning or "big data" methods – such as stepwise regression and neural nets – that use statistical procedures to discover apparent patterns in data without recourse to theory and prior knowledge are akin to alchemy (see, e.g., Einhorn, 1972)."

Roger Penrose is also sceptical on the possibility of "AI" (Shadows of the Mind).

p.25: The relevant section should mention Gardner's conclusions re the improvements in accuracy provided by "damped trend" exponential smoothing models.

p.27: There is no single "naïve approach". See Green & Armstrong re the evidence on simple (often could be characterized as "naïve") vs complex methods.

p. 52: "not grounded in statistical theory". Rather than "statistical theory" should be something along the lines of "theory and prior knowledge on cause and effect".

p. 53: You mention analogies in the context of "ripe for future research", but this is not mentioned in the body text and the Green and Armstrong paper on "structured analogies" in the references is not cited in the text.

Review form: Reviewer 3

Is the manuscript scientifically sound in its present form?

Yes

Are the interpretations and conclusions justified by the results?

Yes

Is the language acceptable?

Yes

Do you have any ethical concerns with this paper?

No

Have you any concerns about statistical analyses in this paper?

No

Recommendation?

Accept with minor revision (please list in comments)

Comments to the Author(s)

The authors have successfully addressed the issues that I raised (as Referee 3) in my review of their first revision. I do think that they have also gone some way to dealing with the points raised by Referee 2 but I was not convinced that their responses to will fully satisfy that referee (e.g., on focus groups). However, it is up to him/her to make that decision.

One point that could be dealt with later in the publication process (but might be better to address now) is that the new list of references excludes some papers that were in the original reference list and are still cited in the text.

Decision letter (RSOS-201187.R1)

Dear Mr Zellner

On behalf of the Editors, we are pleased to inform you that your Manuscript RSOS-201187.R1 "A Survey of Human Judgment and Quantitative Forecasting Methods" has been accepted for publication in Royal Society Open Science subject to minor revision in accordance with the referees' reports. Please find the referees' comments along with any feedback from the Editors below my signature.

Please submit your revised manuscript and required files (see below) no later than 7 days from today's (ie 07-Jan-2021) date. Note: the ScholarOne system will 'lock' if submission of the revision is attempted 7 or more days after the deadline. If you do not think you will be able to meet this deadline please contact the editorial office immediately.

on behalf of Prof R. Kerry Rowe (Subject Editor)
 openscience@royalsociety.org

Associate Editor Comments to Author:

Thank you for engaging with the concerns of the reviewers. A few tweaks remain - and you need to ensure these are addressed in a final revision - to get the paper to a point the editors would be comfortable accepting the paper. Please carefully check the final comments from the reviewers and respond to them.

Reviewer comments to Author:

Reviewer: 3

Comments to the Author(s)

The authors have successfully addressed the issues that I raised (as Referee 3) in my review of their first revision. I do think that they have also gone some way to dealing with the points raised by Referee 2 but I was not convinced that their responses to will fully satisfy that referee (e.g., on focus groups). However, it is up to him/her to make that decision.

One point that could be dealt with later in the publication process (but might be better to address now) is that the new list of references excludes some papers that were in the original reference list and are still cited in the text.

Reviewer: 1

Comments to the Author(s)

p.2: Suggest revising along the lines of... "Critics of this view point out that the use of machine learning or "big data" methods – such as stepwise regression and neural nets – that use statistical procedures to discover apparent patterns in data without recourse to theory and prior knowledge are akin to alchemy (see, e.g., Einhorn, 1972)."

Roger Penrose is also sceptical on the possibility of "AI" (Shadows of the Mind).

p.25: The relevant section should mention Gardner's conclusions re the improvements in accuracy provided by "damped trend" exponential smoothing models.

p.27: There is no single "naïve approach". See Green & Armstrong re the evidence on simple (often could be characterized as "naïve") vs complex methods.

p. 52: "not grounded in statistical theory". Rather than "statistical theory" should be something along the lines of "theory and prior knowledge on cause and effect".

p. 53: You mention analogies in the context of "ripe for future research", but this is not mentioned in the body text and the Green and Armstrong paper on "structured analogies" in the references is not cited in the text.

===PREPARING YOUR MANUSCRIPT===

===PREPARING YOUR REVISION IN SCHOLARONE===

- An editable file of each table (.doc, .docx, .xls, .xlsx, or .csv).
- An editable file of all figure and table captions.

- Any electronic supplementary material (ESM).
- If you are requesting a discretionary waiver for the article processing charge, the waiver form must be included at this step.
- If you are providing image files for potential cover images, please upload these at this step, and inform the editorial office you have done so. You must hold the copyright to any image provided.
- A copy of your point-by-point response to referees and Editors. This will expedite the preparation of your proof.

- Ensure that your data access statement meets the requirements at <https://royalsociety.org/journals/authors/author-guidelines/#data>. You should ensure that you cite the dataset in your reference list. If you have deposited data etc in the Dryad repository, please only include the 'For publication' link at this stage. You should remove the 'For review' link.
- If you are requesting an article processing charge waiver, you must select the relevant waiver option (if requesting a discretionary waiver, the form should have been uploaded at Step 3 'File upload' above).
- If you have uploaded ESM files, please ensure you follow the guidance at <https://royalsociety.org/journals/authors/author-guidelines/#supplementary-material> to include a suitable title and informative caption. An example of appropriate titling and captioning may be found at https://figshare.com/articles/Table_S2_from_Is_there_a_trade-off_between_peak_performance_and_performance_breadth_across_temperatures_for_aerobic_scope_in_teleost_fishes_/3843624.

Author's Response to Decision Letter for (RSOS-201187.R1)

See Appendix F.

Decision letter (RSOS-201187.R2)

Dear Mr Zellner,

It is a pleasure to accept your manuscript entitled "A Survey of Human Judgment and Quantitative Forecasting Methods" in its current form for publication in Royal Society Open Science. The comments of the reviewer(s) who reviewed your manuscript are included at the foot of this letter.

on behalf of R. Kerry Rowe (Subject Editor)
openscience@royalsociety.org

Appendix A**ROYAL SOCIETY
OPEN SCIENCE****A Survey of Human and Machine Forecasting Methods**

Journal:	Royal Society Open Science
Manuscript ID	RSOS-192145
Article Type:	Review
Date Submitted by the Author:	07-Dec-2019
Complete List of Authors:	Abbas, Ali E.; University of Southern California, Daniel J. Epstein Department of Industrial and Systems Engineering Budescu, David; Fordham University Galstyan, Aram; University of Southern California, Information Sciences Institute Zellner, Maximilian; University of Southern California, Daniel J. Epstein Department of Industrial and Systems Engineering
Subject:	Human-computer interaction < COMPUTER SCIENCE, Computer modelling and simulation < COMPUTER SCIENCE, Artificial intelligence < COMPUTER SCIENCE
Keywords:	Forecasting, Human machine interaction, Human machine forecasting, Aggregation, Belief updating
Subject Category:	Engineering

**Author-supplied statements**

Relevant information will appear here if provided.

***Ethics***

*Does your article include research that required ethical approval or permits?:*

This article does not present research with ethical considerations

*Statement (if applicable):*

CUST_IF_YES_ETHICS :No data available.

***Data***

*It is a condition of publication that data, code and materials supporting your paper are made publicly*
*available. Does your paper present new data?:*

My paper has no data

*Statement (if applicable):*

CUST_IF_YES_DATA :No data available.

***Conflict of interest***

I/We declare we have no competing interests

*Statement (if applicable):*

CUST_STATE_CONFLICT :No data available.

***Authors' contributions***

This paper has multiple authors and our individual contributions were as below

*Statement (if applicable):*

All authors contributed to this work in equal amounts.

A Survey of Human and Machine Forecasting Methods

Ali E. Abbas (aliabbas@usc.edu)
University of Southern California

David V. Budescu (budescu@fordham.edu)
Fordham University

Aram Galstyan (galstyan@isi.edu)
University of Southern California

Maximilian Zellner (mzellner@usc.edu)
University of Southern California

Abstract

This paper surveys the literature on human and machine forecasting as well as hybrid forecasting methods that involve both humans and machines. The survey started with key search terms that identified more than 230 publications in the fields of computer science, operations research, risk analysis, decision science, and psychology. The survey results show an almost tenfold increase in the application-focused forecasting literature between the 1990's and the current decade, with a clear rise of machine forecasting models. Comparative studies of machine and human forecasting show that (1) neither method is universally superior, and (2) the better method varies as a function of factors such as availability, quality, extent, and format of data, suggesting that (3) the two approaches can complement each other to yield more accurate and resilient models. From this review, we also identify four research thrusts in the human/machine-forecasting literature: (i) the choice of the appropriate machine model, (ii) the nature of the interaction between machine models and human forecasters, (iii) the training and proper incentivization of human forecasters, and (iv) the aggregation of opinions (both machine and humans) into one judgement. This review surveys current research in all four areas and argues that future research in the field of human/machine forecasting needs to consider all of them when investigating predictive performance. We also address some of the ethical dilemmas might arise due to the combination of machine and human models.

1. Introduction

People (and organizations) usually employ human experts and / or apply algorithmic procedures to forecast an uncertain quantity or to determine its distribution. With the wide availability of data and advances in computing technology, algorithmic forecasts offer the opportunity to support human forecasters by mining large datasets and learning patterns and trends from data.

Several survey papers have previously reviewed the literature associated with human forecasting alone. For example, Lawrence, Goodwin, O'Conner, & Önkal (2006) offers a comprehensive view of judgmental forecasting. Clemen and Winkler (1999) also review a variety of human aggregation methods. Other literature focuses on comparing human and machine forecasts. For example, Grove, Zald, Lebow, Snitz, & Nelson (2000); Kuncel, Klieger, Connelly, & Ones (2013); Ægisdóttir, et.al. (2006); and Meehl (1954) review findings comparing human and machine predictions in a clinical setting. Within the context of mental health practitioners, they found that machine prediction methods outperform humans, although prediction accuracy varied by several factors including the type of prediction, how and where predictor data were gathered, which statistical procedure was used, and how much information was available.

These studies delineate cases in which either human or machine forecasting methods proved superior. Combining the two fields could mean that their inherent shortcomings balance each other out, thereby increasing forecasting accuracy and reliability. For this reason, this literature review surveys both human and machine-forecasting methods as well as hybrid (human and machine) forecasting. We evaluate research on human forecasting, such as incentivization, scoring, calibration, and group-forecasting, but also discuss qualitative methods. The machine methods in this review include common approaches such as regression models and smoothing of time-series, and more advanced methods such as neural and Bayesian networks, ARIMA, and simulation. On the intersection of human and machine forecasting, we discuss research issues such as algorithm aversion, belief updating, and human trust in machine forecasts. Aggregation of forecasts, which applies to human and machine methods and which has been discussed in previous survey papers, is updated and expanded with more recent methods.

The objective of this review is twofold: (i) first, we survey the machine forecasting and the hybrid
forecasting literature together with human forecasting, and identify current and future trends. (ii) In the
process, we revisit and update the previous literature reviews to include new literature in the field of
aggregation and forecasting. The research methodology, which is discussed in more detail in the
following section, involved defining appropriate key terms for searching platforms such as Google
Scholar, Mendeley, and EBSCOhost. Searching for these terms, we observe an almost tenfold increase of
publications concerning forecasting applications. The majority of these focus on time-series models and
other machine forecasting methods, thereby reinforcing the need for a comprehensive review of these
methods and their interactions with human forecasters.

The remainder of this paper is structured as follows. Section 2 offers an overview of the research
methodology used to identify the relevant citations used in this review. Section 3 reviews the literature on
human forecasters. Section 4 reviews machine models for forecasting, while distinguishing between time-
series, correlational, and overarching models that can be used for both purposes. Section 5 reviews
human/machine-forecasting methods. Section 6 discusses the mathematical aggregation of forecasters'
opinion, which can be divided into non-Bayesian and Bayesian approaches. Section 7 summarizes the
results and the main conclusions.

**2. Research Methodology**

The approach used to select the literature citations for this review is shown in Figure 1. First, we
identified the main goal of the study, which is to review the literature on human forecasting, machine
forecasting, and hybrid forecasting methods. The key terms to determine relevant literature are given by
Table I.

We then performed a broad search of the main forecasting methods that have been proposed and the
main application areas. The application areas were then used to conduct more in-depth searches, which
helped us identify the most relevant journals and publications in the field. These, and referenced

materials, were then searched and their findings condensed to derive main trends. Figure 1 shows the process of the literature review and references the sections of this paper that contain the results.

Table 1: Key search terms

 • Forecasting • Human forecasting • Forecasting using experts • Causal and time series forecasting 	 • Machine forecasting • Artificial intelligence for prediction • Human computer interaction in forecasting • Aggregation of expert opinion
---	---

Figure 1: Methodology used for literature review

2.1. FORECASTING FIELDS AND METHODS

Using the search term “forecasting” to determine common forecasting application fields and methods, Google Scholar yielded approximately 2.7 million results when a search was conducted in June 2018. The forecasting fields were identified by conducting a search using the key term “forecasting” and then using the results given by the “related search” feature. In a second step, we took these results to conduct a search to determine the number of publications on each topic. Figure 2 shows that forecasting demand and weather are the fields most frequently covered by scientific publications, followed by electricity forecasting.

In the past 38 years, the number of publications in these areas increased more than tenfold, from 111.400 publications in the period from 1980-1989 to 1.280.200 publications in the period 2010-2018. From our point of view, this increase in publications could be attributed, at least in part, to the increasing

use of renewable energies. The amount of energy that renewable sources can produce usually depends on the weather, so utility companies have an interest in forecasting energy demand and the weather.

Figure 2: Distribution of most frequently searched forecasting topics according to Google Scholar

Using the search term “forecasting methods” and the “related search” feature of Google Scholar, we identified the most prominent forecasting methods. Figure 3 shows the trend in publications concerning the most searched forecasting techniques. The figure shows that time series forecasting has been the most prominent forecasting approach, followed by neural networks, and the specific time-series forecasting method ARIMA (short for Autoregressive Integrated Moving Average). Affective forecasting, which mainly covers forecasting based on behavioral decision making, was also a prominent search term.

Figure 3: Most frequently searched forecasting methods according to Google Scholar

Figure 4 shows the share of publications in several subareas of forecasting. We classified machine forecasting methods into the more common types such as time-series, causal and artificial intelligence. Artificial intelligence applications increased their share in publications, mostly at the expense of more traditional machine forecasting methods. Over the four periods from 1980-1989 to 2010-2018, the total number of publications across the areas surveyed increased from 43.390 to 442.800.

Figure 4: Distribution of publications in various forecasting fields using Google Scholar

2.2. DETERMINATION OF RELEVANT JOURNALS

We restricted the original search to the most cited peer reviewed journals in each of the subareas of human/machine-forecasting. Table 2 lists the journals that appeared most frequently when searching for the specific search key terms of Table I on Google Scholar. Additionally, we determined the most prestigious conferences in the field of machine learning and artificial intelligence using the automatic H5-index ranking provided by Google Scholar. In the field of machine forecasting, we also used the transactions of the IEEE Computer Society, which bundles the proceedings of relevant conferences, such as the “IEEE Transactions on Pattern Analysis and Machine Intelligence”, the “IEEE International Conference on Big Data”, the “IEEE Conference on Data Mining”, and the “IEEE Transactions on Knowledge and Data Engineering”. These journals and conference proceedings provided an initial starting point. Publications from other sources were included if they were cross-referenced in the original set of journals and were considered relevant to this review.

Human Forecasting	Machine Forecasting
 • International Journal of Forecasting • Journal of Forecasting • Journal of Behavioral Decision Making • Psychometrika • Psychological Assessment • Judgmental Forecasting 	 • International Journal of Forecasting • Journal of Forecasting • Management Sciences • Neurocomputing • Computers & Operations Research • Association for Uncertainty in Artificial Intelligence • Journal of Machine Learning Research • Proceeding of the AAAI Conference on Artificial Intelligence • Transactions of the IEEE Computer Society • ACM SIGKDD Conference on Knowledge Discovery and Data Mining • Conference on Neural Information Processing Systems
Human-Machine Interaction	Forecaster opinion aggregation
 • Computers in Human Behavior • Journal of Behavioral Decision Making • Ergonomics • International Journal of Forecasting • International Journal of Industrial Ergonomics 	 • Risk Analysis • International Journal of Forecasting • European Journal of Operational Research • Management Science • Operations Research

Table II: Relevant journals for each field of hybrid forecasting

After determining the relevance of the pre-filtered literature by reading the abstract and conclusion, we identified 234 sources to be relevant to this specific literature review. Figure 5 shows the distribution and type of sources over the years. The figure shows a clear “recency” pattern: most sources quoted have been published after the year 2000 (46.6%), and 23.3% after the year 2010.

Figure 5: Type of publication over years

Figure 6 depicts the different categories and fields in which the assessed literature was published. This was achieved by assigning a journal to a field if the name of the journal or the title carried certain terms. Whereas Computer Science and Operations Research are wide fields with multiple subfields, the category Forecasting exclusively deals with different aspects of forecasting.

Figure 6: Distribution of sources along categories

All the journal publications and most of the books were downloaded to be analyzed using the library of the University of Southern California. The databases and websites used for this review were the

following: Google Scholar, Library of the University of Southern California, Mendeley, ScienceDirect,
EBSCO, IEEEExplore, Emerald, WISO, and INFORMS.

**3. Human Forecasting**

Human forecasting employs one or multiple human forecasters to provide an opinion. In this
section we distinguish between aspects arising in individual human forecasting and forecasting using
groups. The first section discusses issues concerning using individuals when generating a forecast, while
the second section elaborates on group forecasting techniques.

**3.1. INDIVIDUAL HUMAN FORECASTING**

This section reviews individual human forecasting methods, and related topics such as probability
elicitation, the impact of incentive schemes, forecaster calibration, training, as well as scoring rules.

**3.1.1. PROBABILITY ELICITATION**

Using human opinion for forecasting purposes, a decision maker who requires a forecast on a
specific problem is faced with eliciting probability forecasts from an expert and aggregating multiple
opinions (Clemen & Winkler, 1999). Concerning individual human forecasters, research focuses on
elicitation and evaluation methods (e.g. Spetzler and von Holstein (1975), Wallsten and Budescu (1983)),
and multiple ways to construct continuous probability distributions from the elicited point estimates (e.g.
Moder and Rodgers (1968), Smith (1993), Abbas (2003), Abbas (2006)). One has to distinguish between
eliciting a point forecast and a probability distribution over the space of all outcomes, which requires
methods to construct the continuous distribution. The elicitation process applies to both point and
distribution forecast and aggregation approaches are discussed in a separate section because they apply to
both human and machine forecasting.

There exist three main methods of eliciting expert opinion. The first method provides a fixed
probability and asks the expert for the corresponding variable value. The second approach provides a
fixed value and elicits the corresponding probability, and the third method is a combination of the two

(Spetzler & von Holstein, 1975). Abbas et. al (2006) assess the two methods along several dimensions
including monotonicity, accuracy, and precision of estimated fractiles in a behavioral experiment, and
found a slight superiority of the fixed variable value approach. They also found that participants preferred
this approach, alleging that fixed value estimates were more familiar to them from their everyday-life
(Abbas, Budescu, Yu, & Haggerty, 2008). Despite the results of the elicitation methods being similar, the
insights suggest that how forecasting questions are presented to experts impacts the speeds and accuracy
of the resulting forecast.

The immediately following subsections investigate factors that impact the quality of the elicited
forecast, as well as methods to control for biases and past forecasting performance.

17 18 19 20 21 22 23 **3.1.2. INCENTIVE SYSTEMS**

Incentive schemes in forecasting are designed to foster truthful reporting by the forecaster
(Surowiecki (2001), Brier (1950), Gneiting and Raftery (2007), and Winkler (1996)). Using game
theoretic terminology, truthful reporting refers to an agent revealing his, or her, true opinion or
assessment to the less-informed principal. Ottaviani and Sørensen (2006) and Lichtendahl and Winkler
(2007) have shown that forecasters who compete to be the most accurate have an incentive to not report
their true opinion. In their paper Lichtendahl, Grushka-Cockayne, and Pfeifer (2013) have shown that an
advice seeker benefits from setting up an incentive scheme that rewards strategic behavior instead of
truthful reporting. Their claim suggests that such a system incentivizes forecasters to place more emphasis
on their private instead of on their public signal, where the term signal refers to private or publicly
available information.

This insight addresses one of the biggest issues of forecast aggregation. According to Morris (1974),
dependence between forecasters due to common sources of information, educational background, etc. is
the biggest challenge to the aggregation of opinion. Designing an incentive scheme in such a way that
strategic behavior is rewarded, forecasters tend to use more information that is only available to them to

increase their chances of “winning”. By doing this, the forecasts provided become more independent from
each other resulting in a higher forecasting accuracy.

Using this game theoretic insight, researcher in the field of artificial intelligence have developed an
algorithm to prevent untruthful reporting and to determine the human forecaster who gives the most
truthful assessment (Witkowski, Freeman, Wortman Vaughan, Pennock, & Krause, 2018).

14 15 **3.1.3.FORECASTER CALIBRATION AND TRAINING**

Tetlock (2006) introduced the distinction between hedgehogs and foxes. The term hedgehog is used
for domain specific or niche forecasters, who deliberately do not consider other domains when making a
forecast, while foxes are trying to be more aware of the “big picture”. Considering the forecasting problem
from multiple perspectives and drawing on multiple sources of information, foxes tend to outperform
hedgehogs in forecasting competitions (Tetlock (2006), Tetlock and Gardner (2016)). Kahneman (2011)
extensively discusses biases in human decision making, such as confirmation bias and overconfidence in
one’s abilities, that apply to the human forecasting domain. Chang, Chen, Mellers, and Tetlock (2016)
have shown that these biases are not insurmountable, and that forecasting accuracy can be improved by
short trainings (see also Chen, Budescu, Lakshmikanth, Mellers, and Tetlock (2016). The training
addresses human flaws in understanding probability (Bar-Hillel (1980), Kahneman and Tversky (1973),
Kahneman and Tversky (1984), Lichtenstein, Slovic, Fischhoff, Layman, and Combs (1978), and Slovic
and Fischhoff (1977)), but also incorporated procedures to explicitly search for information that go against
personally held beliefs and counterfactual thinking.

Sanders and Ritzman (1992) have shown that training subjects in gathering and analyzing contextual
data had a bigger impact on forecasting accuracy compared to only training them in technical/ statistical
aspects. Assessing the impact on forecasting accuracy when training humans to use algorithms remains an
important and, relatively, open field of future research.

3.1.4. SCORING RULES

Scoring rules provide summary measures of probabilistic forecasts by assigning numerical scores based on the predictive distribution and on the event or value that materializes (Gneiting and Raftery (2007), Kotz, Read, Balakrishnan, Vidakovic, and Johnson (2004)). In the context of human/machine-forecasting, such measures can be used to assign differential weights during the aggregation stage. Scoring rules are used to incentivize the assessor into expending effort by tying compensation to his or her score (Garthwaite, Kadane, & O'Hagan, 2005).

Assuming an expected utility maximizing forecaster, a proper scoring rule ensures full revelation of the forecaster's subjective belief. More formally, if r is the vector of reported beliefs and p is the vector of subjectively-held beliefs, then for every r

$$E(S(p)) > E(S(r)),$$

where E denotes the expectation and $S(\cdot)$ is the scoring rule

The most widely-used scoring rules in forecasting are the quadratic/ Brier, logarithmic, and spherical scoring rules (Matheson and Winkler (1976), Murphy and Winkler (1970)). Harte and Vere-Jones (2005) proposed an entropy score in forecasting geological events, and Diebold and Mariano (1995) elaborated on case-specific loss functions. Scoring rules take on different functional forms depending on the forecast object, i.e. whether it is a categorical quantity, intervals, or continuous distributions (Gneiting & Raftery, 2007). To evaluate point forecasts, error measures such as the absolute percentage error, are also being used (Gneiting T. , 2011). The main points of criticism of these scoring rules are their disregard of task difficulty and the lack of sensitivity to distance (Winkler R. L., 1981). To address this problem, Winkler (1994) developed asymmetric scoring rules and discussed their advantages using a weather forecasting case study. Scoring rules that are sensitive to distance punish distributions with heavy or "fat" tails. For example, the quadratic scoring rule assigns the same score to the forecasts $p_1 = (0.3, 0.4, 0.3, 0, 0)$ and $p_2 = (0.2, 0.4, 0.2, 0.1, 0.1)$, if the second outcome materializes. A scoring rule sensitive to distance would assign a higher score to the forecast with lower variance. Distance-sensitive

scoring rules were developed by Epstein (1969) and Staël von Holstein (1970) for categorical forecasts
and were expanded upon by Matheson and Winkler (1976). One of the most frequently used scoring rule
sensitive to distance is the ranked probability score, which has been expanded by Boero, Smith, and
Wallis (2011). The ranked probability score was shown to be superior to the quadratic and logarithmic
scores in forecasting economic aspects, such as inflation. Another family of scoring rules that are
sensitive to distance are so-called beta scoring rules (Buja, Stuetzle, & Shen, 2005) (Merkle & Steyvers,
2013).

Regarding the criteria to choose a scoring rule, Merkle and Steyvers (2013) found that it is not
sufficient to choose scoring rules that are proper, but one should consider the specific way forecasters are
rewarded and penalized. In their opinion, the scoring rules belonging to the beta family offer a decision
maker the opportunity to tailor the scoring rule to his or her needs.

17 18 19 20 21 22 23 24 25 26 27 28 **3.2. GROUP FORECASTING**

To reduce the impact of bias of one individual forecaster and to increase forecasting accuracy,
multiple humans can be employed. Group forecasting rely on qualitative or contextual data provided by
multiple human forecasters. There exists a multitude of qualitative forecasting approaches, which include
but are not limited to Delphi (Dalkey (1969), Linstone and Turoff (1975)), Market Research (Bass, King,
& Pessemier, 1968), Panel Consensus, Visionary Forecast, and Historical Analogy (Spencer (1961),
Chambers, Mullick, and Smith (1971)), Group Discussion (Aumann, 1976), Decision Conferencing
(Phillips L. D., 1984), (Phillips L. D., 1987), Nominal Group Technique (Delbecq, Van de Ven, &
Gustafson, 1975), and Focus Group. In the following subsections we discuss the most well-known
techniques Focus Group, Nominal Group Technique and Delphi method (Landeta, Barrutia, & Lertxundi,
2011).

3.2.1. DELPHI METHOD

The Delphi methodology was developed during the 1950s by Olaf Helmer, Norman Dalkey and Ted Gordon at the RAND Corporation (Linstone & Turoff, 2011). It was not designed to replace quantitative approaches or models, but to offer a structured approach if no quantitative data were available (Wright, Lawrence, & Collopy, 1996). It has been applied in various fields including healthcare (Hudak, Brooke, Finstuen, & Riley, 1993), marketing (Lunsford & Fussell, 1993), education (Olshfski & Joseph, 1991), information systems (Neiderman, Brancheau, & Wetherbe, 1991), transportation and engineering (Saito & Sinha, 1991), and finance (Kauko & Palmroos, 2014).

The key features of the Delphi method are anonymity, iteration, controlled feedback, and statistical aggregation of the group response (Linstone & Turoff, 1975). Anonymity is ensured by giving forecasters a questionnaire containing the forecasting problem, whose responses the other subjects cannot discern. The aim is to prevent social pressures from changing a forecaster's opinion. The anonymous responses are then statistically analyzed, and the mean and variance are supplied to all the forecasters to update their prior belief. If someone's update is an outlier, the forecaster usually has to provide a reason. The process is then repeated for several rounds, before all individual opinions are aggregated. Most commonly a linear opinion pool with equal weights for each forecaster is being used for the aggregation (Rowe & Wright, 1999). There exist several variations of this technique. For example, the first round can be unstructured to not constrain the forecaster (Martino, 1992), or structured to make the procedure simpler for the monitoring team (Linstone & Turoff, 1975).

Studies comparing forecasts produced using the Delphi method with individual human forecasts have shown an improvement in accuracy and reduction in variance, favoring the former approach (Rowe and Wright (1999), North and Pyke (1969)). Despite anonymity in eliciting opinions, a main criticism of the Delphi technique is the inherent pressure to conform to group opinion after the first round of iteration. This pressure could lead to valid minority opinions being disregarded, neglecting tail probabilities (Sackman, 1975). Psychological studies have found that the forecasting accuracy of the Delphi method benefits from emphasizing reasoning, such that subjects have to provide detailed reasons for their opinion.

This reasoning could then be used in the feedback process, making it more convincing to other subjects
who tend to be biased toward their own assessments (Bolger & Wright, 2011). As with any other
qualitative method relying on several forecasters, the quality and accuracy of the generated forecast
depends on the study design, as well as how it addresses human biases such as anchoring, framing, and
desirability bias (Winkler & Moser, 2016).

14 15 **3.2.2. FOCUS GROUPS**

Unlike the Delphi technique, focus groups rely on face-to-face discussions between human
forecasters on a predefined forecasting topic under the supervision of a moderator (Blackburn and Stockes
(2000), Krueger and Casey (2014), Robinson (1999)). The advantages of this method are the simplicity of
setting up the group, fast and easy sharing of information, and high acceptance of the group opinion by
individual forecasters (Landeta, Barrutia, & Lertxundi, 2011). The method also suffers from several
downsides, including susceptibility to group-think (McNees, 1987), which might be exacerbated in
comparison to Delphi by its reliance on face-to-face discussions, a desire to be accepted (Janis, 1982), and
incongruences due to social status of group member (Collins & Guetzkow, 1964). The method does not
predefine how individual opinions are to be aggregated. The choice of the aggregation rule depends on
the moderator and the social dynamics of the. Depending on these, and other, factors the aggregate could
be a linear pool, the majority opinion, or even the opinion of the group member with the highest social
status.

**3.2.3. NOMINAL GROUP TECHNIQUE**

The nominal group technique is a structured method to elicit the opinion of forecasters in a physical
group. It was developed to compensate for some of the shortcoming of the Delphi technique, emphasizing
creativity through group dynamics (Landeta, Barrutia, & Lertxundi, 2011). The process can be divided
into five steps: First, the moderator poses the forecasting question. Then each forecaster individually
produces a forecast, which is then explained to other members of the group to generate debate. These
forecasts are subsequently anonymously assessed and ranked by each individual, before being aggregated

by the moderator, commonly using a linear opinion pool (Moore, 1987). This technique has several
advantages over the previously discussed ones. In contrast to a focus group, the nominal group technique
follows a clear structure and is not as prone to groupthink and social pressure. It is better than Delphi
when it comes to stimulating creativity and tends to be less time consuming because it does not involve
multiple iterations (Moore, 1987). Nevertheless, the method also suffers from drawbacks, such as a limit
on the number of forecasters for it to be effective, risk of groupthink (compared to Delphi), and requiring
the forecasters to be in the same physical space at forecasting time (Landeta, Barrutia, & Lertxundi,
2011). Several studies suggest that the nominal group technique is less accurate and reliable than Delphi
(Hutchings, Rosalind, Colin, and Nick (2006), Rowe and Wright (1999)).

**4. MACHINE FORECASTING**

In contrast to human forecasting techniques, machine forecasting relies heavily on quantitative data to
derive a forecast because it uses past observations to build the prediction model. The choice of a machine
forecasting method depends on factors such as the context of the forecast, the relevance and availability
of historical data, the degree of accuracy desirable, and the time period of the forecast (Chambers,
Mullick, & Smith, 1971). In our review we focus on quantitative models, which can be subdivided into
time series and similarity-based/correlational models. We include a third category of models, which is
able to address both time series and similarity-based forecasting.

**4.1. TIME SERIES METHODS**

Time series methods use historical data to identify patterns and pattern changes. They are
commonly applied in diverse fields such as finance (Leuthold, MacCormick, Schmitz, & Watts, 1970),
electricity markets, retail, and optical transport networks (Cavalcante, Celestino, & Patel, 2017). Although
there exist a multitude of time-series forecasting methods, we limit this review to the most commonly
used ones and exclude more application-specific methods such as “Robust-Trend” (Grambsch and Stahel
1990) and “Theta” (Assimakopoulos and Nikolopoulos 2000).

The accuracy of the different time series methods presented in this section and the impacting factors
have been discussed in depth by Makridakis, et. al., (1982). They found that the method has to be chosen
in accordance to the available data to maximize accuracy. For example, if available data is on a quarter
6 year basis instead of on a yearly basis, it is advisable to use a time series method that is capable of
7 incorporating seasonality. Chatfield (1988) distinguishes between univariate, multivariate, and judgmental
time-series methods. Whereas univariate models draw on past data of one particular variable to forecast
the future, multivariate time series depend, at least partly, on values of more other series. Judgmental
time-series methods concern humans extrapolating time-series into the future and adjusting the series for
contextual data (Goodwin & Wright, 1993). Chatfield (1988) concluded that there is no 'best' forecasting
method. The choice of method rather depends on the objective in producing the forecast, the type of time
series and its statistical properties such as trend or seasonality, the number of past observations available,
the length of the forecasting horizon, the number of series to be forecasted and the cost allowed per series,
the skill, experience and interests of the analyst, and the computer programs available. He derived several
observations when dealing with time-series methods. First, post-sample forecast errors might not be
minimized by fitting the 'best' model to historical data, as the 'best' model might be overfitting, i.e.
incorporating too much noise of the historical data in its forecast. Univariate models are most suitable for
short-term (up to 6 months into the future) forecasting and the combination of forecasts from different
methods generally outperforms any individual method. This can be explained by the fact that each method
makes different assumptions about trends, seasonality, etc., and combining different methods reduces the
effect of bias compared to when using only one. Another observation was that forecasting accuracy
benefits from a higher level of aggregation of series. Iosevich, Arutyunyan, and Hou (2015) for
example, outlined a dynamic modeling approach that aggregates multiple time series and showed its high
accuracy using an example.

Several forecasting competitions covering a range of real-world time-series have shown that more
sophisticated models do not necessarily outperform simpler methods (Makridakis, et al., (1982),
Makridakis, et al., (1993), Makridakis and Hibon (2000)).¹ The measures used in the so-called M-

Competitions to assess the accuracy of time series models were the symmetric mean absolute percentage
error (sMAPE), average ranking, percentage better, median symmetric absolute percentage error (median
symmetric APE), and median relative absolute error (Median RAE) (Makridakis, et al., 1982).

Armstrong and Collopy (1992) evaluated these measures for making comparisons of errors across
time series. They were judged on their reliability, construct validity, sensitivity to small changes,
protection against outliers, and their relationship to decision making. Their final recommendation was to
use the geometric mean of the relative absolute error when the task involves model calibration for a set of
time series. To select the most accurate time series method, the findings recommend using the Median
Relative Absolute Error in case of few series being available. If more time-series can be accessed, then
the Median Absolute Percentage Error is recommended as selection criterion. The commonly used Root
Mean Square Error was deemed unreliable, and it was recommended not to use it when comparing
accuracy across series (Armstrong & Collopy, 1992).

**4.1.1. MOVING AVERAGE**

This forecasting method calculates a series of averages of different subsets of the data set, with
different variations, such as simple, cumulative, or weighted forms. Where the simple moving average
allocates the same weight to each data point, the cumulative version uses the cumulative average, and the
weighted moving average weights are usually determined by using the data point's date. For more details,
please refer to Chou (1975) and Hadley (1968).

**4.1.2. EXPONENTIAL SMOOTHING**

In contrast to simple moving average, exponential smoothing assigns exponentially decreasing
weights over time. It is commonly applied to smooth data, acting as low-pass filters to remove high
frequency noise. There are several exponential smoothing approaches available in the statistical literature.
Single exponential smoothing, first suggested by Brown (1956) and expanded by Holt (2004), applies a
smoothing factor on past data, and derives forecast by calculating the weighted average. As single
exponential smoothing does not perform well if there are trends or seasonality in the data, the approach

has been expanded to second and third order exponential smoothing. If there is only one perceivable
trend in the data, second order exponential smoothing is appropriate, which has to be extended to a three-
parameter smoothing if seasonality is detected (Winters, 1960). During several forecasting competitions,
Makridakis, et al., (1993) found that exponential smoothing performed well in time-series forecasting.
Dantas and Oliviera (2018) propose improving exponential smoothing for time-series forecasting by
incorporating the statistical learning techniques bootstrapping and clustering. Using the M3-competition
data set, they managed to reduce forecasting error when compared to conventional exponential
smoothing.

**4.1.3. FOURIER TIME SERIES DECOMPOSITION**

This method aims to explain the time series entirely as a composition of sinusoidal functions, thus
being able to incorporate trends as well as seasonality (Pollock, Green, & Nguyen, 1999).

**4.1.4. AUTOREGRESSIVE-INTEGRATED-MOVING AVERAGE (ARIMA)**

This method was developed in the early 1900s but became popular only in the 1970s (Chase (2013),
Box, Jenkins, Reinsel, and Ljung (2015)). There are three basic steps in this approach, also referred to
Box-Jenkins method. First, a tentative model is identified. This usually happens under the assumption that
the pattern of the time series can be explained by one of the three model components: The autoregressive
component relates the current value to its own previous values. The moving average process relates the
current value to the previous errors. The integration refers to the combination of the two previous
categories, which is mainly determined by the autocorrelation of lag variables (Box, Jenkins, Reinsel, &
Ljung, 2015).

The most important aspect using this model is to achieve stationarity. In case one can determine a
trend in the modelled data, stationarity has not been achieved yet, and more differencing is necessary.
Once stationarity is achieved, model parameter coefficients can be estimated and the forecast is generated.

The ARIMA method is very popular because of its wide applicability. It is able to incorporate
seasonality and trends, deemed highly accurate, and extendable to exogenous variables (Box, Jenkins,

Reinsel, & Ljung, 2015). Incorporating exogenous variables gives the so-called ARIMAX-model, which
can be considered a hybrid between time-series and correlational models.

Drawbacks of this approach are its complexity, its requirement of large data sets, and its need for
updating once new data is collected (Box, Jenkins, Reinsel, & Ljung, 2015). For further details on
ARIMA and its extensions, refer to Makridakis, Wheelwright, and Hyndman (1998) and Pankratz (1991).
One commonly used extension is the X-11-ARIMA model that decomposes time series into seasonals,
trend cycles, and irregular elements. For details on how to set the filters for seasonal adjustment, refer to
Dagum (1983).

**4.2. Similarity-Based/ Correlational models**

Correlational, or similarity-based, models use highly refined and specific information about
relationships between system elements, and are able to formally incorporate special events. According to
Chambers, Mullick, and Smith (1971), correlational or causal models tend to be elaborate and time-
intensive to construct.

**4.2.1. REGRESSION MODELS**

These very simple models use a dataset containing values for input and response variables.
Applying a linear function to each observation and then minimizing the residual sum of squares, one can
derive the coefficients of each input variable. For details on simple and multiple regression, refer to
Clelland, DeCani, and Brown (1973) or Makridakis, Wheelwright, and Hyndman (1998). Logistic
regression, which computes the probability that an object is of a certain type, is often found in artificial
intelligence and machine learning applications. For example, Korkmaz, et al., (2015) used logistic
regression to make a probability forecast on civil unrest using social network data from Twitter,
Facebook, etc. Conditional Random Fields (CRFs) is a common tool in artificial intelligence that uses
regression efficiently to model effects of interactions of objects in large datasets (Ristovski,
Radosavljevic, Vucetic, & Obradovic, 2013). An exemplary application of CRF to forecast loads in an
electricity network is given by Guo (2015).

4.2.2. UNOBSERVED COMPONENTS MODEL (UCM)

This model was first introduced by Harvey (1989) to the field. The UCM can be described as a model with multiple regressions with time varying coefficients. It additively decomposes a time series into trend, seasonal, cyclical, and irregular components and allocates different weights to events, depending on when they occur in the series. A good example of UCM forecasting demand in telephone networks is discussed by Tych, Pedregal, Young, and Davies (2002). For further details on UCM's, refer to Young (2011).

4.3. Overarching quantitative models

For some quantitative models, the distinction between time-series and correlational models is not adequate, as they can either be considered a combination of the two or can be used both for time-series and correlational purposes. Although there are more modeling approaches that could fall into this category, we focus on neural networks, Bayesian networks, and simulation.

4.3.1. NEURAL NETWORKS

Neural networks (Hastie, Tibshirani, & Friedman, 2017) are used in a variety of fields such as medical diagnosis and image recognition (Cross, Harrison, & Kennedy, 1995), demand forecasting in the aviation industry using a combination of networks (Sineglazov, Chumachenko, & Gorbatyuk, 2014), electric load forecasting (Yuan & Fine, 1992), and finance (Kaastra & Boyd, 1996). Although neural networks have been proposed in the 1940s by McCulloch and Pitts (1943), the lack of computing power and the problem of exclusive-or handling impeded the success of the approach (Ke-Lin & Swamy, 2014).

In general, neural networks are made up of neurons that contain the activation at each time step, a possible threshold that can be changed by a learning function, the activation function which calculates the activation for each time step, and a respective output function. These neurons are connected by synapses, each with their own weights that are adjusted through subsequent learning and backpropagation. Neural networks contain a propagation function that computes the input to the neuron from the outputs of its predecessors and a learning rule (Ke-Lin & Swamy, 2014). Depending on the learning task, we can

distinguish between three different learning paradigms. In supervised learning, the neural network draws
on a set of data with pairs of input and output variables and aims to find the function that matches the
example (Hastie, Tibshirani, & Friedman, 2017). After this training step, the function is applied to inputs
without the observed output (Ojha, Abraham, & Snasel, 2017). These models are also described as
classification and function approximation models, thus being a combination of the two generic
quantitative models. Unsupervised learning uses some given data and aims to minimize the cost function
given by the network output. In reinforcement learning, data is usually not given, but generated by an
agent's interaction with the surrounding environment, which is modeled as a Markov decision process in
many circumstances. Usually, the neural network in reinforcement learning is part of an overall algorithm
(Bertsekas & Tsitsiklis, 1996). Neuro-dynamic programming, which combines a neural network and
stochastic optimization, has been applied in multiple cases, such as vehicle routing (Secomandi, 2000),
resource management (de Rigo, Rizzoli, Soncini-Sessa, Weber, and Zenesi (2001), Damas, et al., (2000)),
medicine (Deng & Ferris, 2008), and retail (Nunnari & Nunnari, 2017).

Although neural networks have become very popular in the field of artificial intelligence, several
papers revealed some of their limitations. For example, they often require large datasets to train the
network as well as considerable computing power (Hastie, Tibshirani, & Friedman, 2017). Furthermore, a
neural network is often viewed as a "black box", making the forecasting process intractable to the user.
When comparing real-world time-series forecasts of statistical methods and neural networks, the former
seems to outperform the more advanced machine learning methods on the basis of sMAPE and MASE
(Makridakis, Spiliotis, & Assimakopoulos, 2018). The authors compared the forecasting accuracy of
machine learning methods, such as Multi-Layer Perceptron, Bayesian Neural Networks, Generalized
Regression Neural Networks, and CART Regression Trees, to the accuracy of random walks, different
methods using exponential smoothing, Theta model, and ARIMA. They saw the main reasons for the
worse performance of machine learning models primarily in over-fitting the model and in the
computational complexity. Nevertheless, they point out that statistical methods improved in accuracy over

time and are confident that the same will apply to machine learning methods (Makridakis, Spiliotis, &
Assimakopoulos, 2018).

8 **4.3.2. BAYESIAN NETWORKS (BN)** 9

Bayesian networks are representations of probabilistic dependence between a given set of random
variables and a acyclic graph (Nagarajan, Scutari, and Lèbre 2013). They provide an easy and fast way of
updating prior beliefs, as well as eliciting dependence information (French, 2011). The network's
structure can either be derived by human experts or by employing historic data (Nagarajan, Scutari, and
Lèbre 2013). Using human experts, Stiber, Small, and Pantazidou (2004) constructed Bayesian Networks
for each of their forecasters and aggregated the forecasters' posteriors using a linear opinion pool. The
weights of the linear pool were determined by posterior probability weights, i.e. by the accuracy of the
individual assessments. The assumption in this research was that the forecasters previously agreed upon
the general structure of the BN. Etiminani, Naghibzadeh, and Pena (2013) claim that the use of BNs in
forecast aggregation and forecasting can be subdivided into the problem of aggregating the structure and
the problem of aggregating the parameters when employing multiple experts.

Etiminani, Naghibzadeh, and Pena (2013) address the issue of aggregating parameters. The
aggregation of structure, meaning the aggregation of several BNs into one BN, is discussed by del
Sagrado and Moral (2003), who distinguish between topological fusion and graphical representation of
consensus. Whereas topological fusion first obtains a consensus structure and then aggregates the model
parameters, graphical representation of consensus approaches the aggregation problem in the opposite
order (del Sagrado & Moral, 2003). The authors propose new combination methods when the initial
models differ on some variables.

Despite their advantages, Bayesian networks feature downsides according to French (2011).
Depending on which process is followed, a significant amount of interaction with human forecasters is
required to determine the consensus structure. If one wants to pursue an algorithmic aggregation of
networks, there is a risk of inconsistencies if the various opinions on the structure diverge too much

(French, 2011). While Henrion (1989) and Nadkarni and Shenoy (2004) have explored ways to derive the
qualitative dependence structure, eliciting the quantitative dependence structure has been identified as the
main issue by Druzdzel and Van der Gaag (1995) and Renooji (2001). Determining causal effect was
discussed by several authors, such as Peña (2017), but increasing the number of variables make it difficult
to generate and update the network. Approaches to lessen the assessment efforts include non-parametric
BN's (Hanea, Morales Nápoles, and Abadei (2015), Morales Nápoles, Kurowicka, and Roelen (2008)),
piecewise-linear interpolation (Wisse, van Gosliga, van Elst, & Barros, 2008), and noisy-OR gates
(Zagorecki & Druzdzel, 2004). This implies that while there exist algorithms to derive the structure of the
network using past data, there exist challenges if one wants to aggregate Bayesian Networks with
different underlying structures.

**4.4. Simulation Models in Forecasting**

Borchev (2013) and Sterman (2000) classify simulation models as agent-based, system dynamics,
and compartmental models. They are common in modeling the spread of diseases and assessing the cost-
effectiveness of interventions, such as in Crooks and Hailegiorgis (2014), Sadilek, Kautz, and Silenzio
(2012), or Bendor, Metcalf, Fontenot, Sangunett, and Hannon (2006). In the field of policy analysis
simulations are applied to assess the effects of policies on the economy, as discussed in Homer and Hirsch
(2006), Tesfatsion (2002), Tesfatsion (2003), and Barlas (2009). Although simulation models are
widespread, they are problematic. They depend on qualitative data for calibration and forecasters to
validate the modeling assumptions. Depending on the forecasting problem this process can be very time
consuming and elaborate (Borchev, 2013).

One particular simulation model for forecasting are naïve forecasts and random walks. According
to Hyndman and Athanasopoulos (2013) naïve forecasts are the most cost-effective models. Using the
naïve approach, forecasts are equivalent to the last observed value, implying that the forecaster does not
possess any further knowledge than what has been observed last. Usually these models perform well in
environments where patterns are hard to forecast (Hyndman & Athanasopoulos, 2013). The naïve model

can be extended by incorporating seasonality and drift to yield random walks. Random walks, such as
standard or geometric Brownian motion, assume changes to be log-normal distributed and use past data to
determine annual drift and volatility. One can either apply the closed form solution or Monte Carlo
simulation to derive the probability density function of the future event (Hull (2015), Ross (2014)).

11 12 13 **5. Human/Machine-Forecasting**

After discussing human and machine forecasting separately, in this section we explore issues and
challenges arising when combining the two. The first sub-section compares the two forecasting strands
and provides a motivation for the combination of human/machine-forecasting. Next, we discuss human
advice taking and belief updating. To conclude this section, we discuss human aversion towards machine
predictions and how allowing humans to change the machine model slightly can increase trust.

27 **5.1. Comparison of forecasting methods and motivation for human/machine-forecasting**

Previous research contrasting the accuracy and performance of human forecasters and machine
models yields mixed results. Highhouse (2008), Dawes (1971), Schweitzer and Cachon (2000), Grove,
Zald, Lebow, Snitz, and Nelson (2000), Kuncel, Klieger, Connelly, and Ones (2013), and Ægisdóttir,
et.al., (2006) have shown that algorithms usually outperform human subjects on forecasting tasks,
although a real-world example of Nike® , given by Worthen (2003), warns against relying exclusively on
computer models without any human supervision and input. A survey of 240 US corporations found that
only 11% used forecasting software, and of those 60% routinely adjusted the generated forecasts based on
individual judgement (Sanders & Manrodt, 2003). Lawrence, Goodwin, O’Conner, and Önköl (2006)
propose a rough forecasting procedure that draws on the advantages of machine models and human
judgment contingent on the availability of historic data. It assumes that contextual information is used by
human forecasters, and quantitative data is analyzed by machine models to inform the human. If there is
no quantitative data available, the forecast is developed by the human without machine assistance.
Armstrong (1983) found that when contextual or domain-knowledge is available, human forecasters tend
to outperform statistical methods. Brown (1996) also concluded that advice seekers should place a

stronger emphasis on human judgments, a conclusion that was also supported by applied research
conducted by Chatfield, Hein, and Moyer (1990) in the electric utility industry. Nevertheless, the
complementary strengths of human judgement and machine models suggest that the combination of these
methods might yield superior forecasting results (Blattberg & Hoch, 1990). A recent study into forensic
facial recognition has shown that combining neural networks and human forecasters has the potential for
stabilizing classification performance, decreasing variability, and increasing performance of average
forecasters (Phillips, et al., 2018). Yaniv and Hogarth (1993) have shown experimentally that when
contextual information is scarce, statistical forecasts usually outperform humans. In their study,
forecasters achieved the highest accuracy with a combination of a statistical base-rate model and human
judgment of contextual data. Miyoshi and Matsubara (2018) found that simply averaging over the forecast
produced by a recurrent neural network and a set of human forecasters outperformed the stand-alone
machine forecast and human forecasters on the basis of the root mean squared error. They also developed
a flexible algorithm that can determine the optimal number of human forecasters as a function of the
expected error of the machine forecast.
Because humans interact with machine in Human/Machine-Forecasting, one could assume that the
quality of these hybrids forecasts depends, at least in part, on the interaction between model and
forecaster. Research has been conducted into questions of when humans trust machine forecasts and what
factors impact the amount of updating that occurs after the human subject has been provided with the
forecasts. We address these questions in the following sections. For more general reviews of the
psychological aspects of human forecasting and advice taking, please refer to Lawrence, Goodwin,
O'Connor, and Önköl (2006) and Bonaccio and Dalal (2006).

**5.2. Human advice taking and belief updating**

In human/machine-forecasting, a machine model might be used to inform the human expert's
opinion or multiple experts provide their opinion to a decision maker. One intuitive research question that
arises is how humans take the provided advice and use it to update their belief. On a human-to-human

basis, Önkal, Gönül, Goodwin, Thomson, and Öz (2017) and Ayton and Önkal (1996) investigated
empirically the driving factors in using recommendations and advice. The authors assess whether
experienced or presumed credibility has more impact on human subjects' readiness to use advice. Authors
such as Fogg (1999), Wathern and Burkneil (2002), and Harvey and Fischer (1997) have argued that the
former, i.e. a good track record of making right forecasts, has the biggest impact on whether users apply
the forecasters' recommendations.

Other researchers, such as Armstrong (1980) and Kahneman (2011), in turn claimed that presumed
credibility, i.e. the credibility perceived through the status of the advice giver, plays the biggest role in
opinion adoption. Önkal, Gönül, Goodwin, Thomson, and Öz (2017) showed, experimentally, that advice
from a forecaster with high experienced credibility received a higher weight and a low level did not affect
the weighting negatively. High presumed credibility in turn did not result in an allocation of more weight
to the model, although low presumed credibility resulted in a decrease. Investigating the interaction
between the two kinds of credibility, the authors also found that the weighting depends on the expertise of
advice-seekers. With non-experts, experienced credibility eclipsed presumed credibility, while both kinds
of credibility were influential in determining the weight allocated to the forecaster opinion if the advice-
seekers were professionals in the same industry. Extending this topic to technology and its incorporation
into forecasting, research by Agarwal and Prasad (1997) implies that the amount of updating depends on
whether the advice receiver was/ an active or passive user of technology. Furthermore, Önkal, Goodwin,
Thomson, Gönül, and Pollock (2009) have shown that, although seeking outside advice generally
improves forecasting accuracy, the amount of updating depends on the source. They found that in the
process of belief updating information from a statistical procedure was discounted more than when the
source was another human forecaster. If the two sources were either human or statistical procedures, this
effect vanished, indicating that human advice is preferred over statistical procedures when both types of
sources are available (Önkal, Goodwin, Thomson, Gönül, & Pollock, 2009).

Several other authors investigated how humans use advice that is provided by human forecasters.
Yaniv and Kleinberger (2000) found that advice is discounted relative to one's prior, meaning that

humans tend to assign higher weights to advice that is consistent with, or confirms, their beliefs and lower
weights to other diverging views. Soll and Larrick (2009) found that the two most common strategies are
choosing one source and averaging them. Despite the latter proving to be more accurate in most
circumstances, humans tend to prefer the former. Combining Yaniv and Kleinberger (2000) and Soll and
Larrick (2009) could suggest that humans tend to choose the forecast that is consistent with, or confirms,
their beliefs, which makes the forecast prone to bias.

**5.3. Algorithm aversion**

Algorithm aversion can be described as the aversion of humans to take advice if it was generated by
a machine or algorithm.. Dietvorst, Simmons, and Massey (2015, (2016) investigate how humans use
machine forecasts contingent on the quality of outputs and on how much human forecasters can alter the
provided forecasts. Carbone, Andersen, Corriveau, and Corrsen (1983) and Armstrong (1985) suggested
that allowing for human adjustment of the machine model might harm accuracy. Supporting this claim,
Eggleton (1982) and O'Connor, Remus, and Griggs (1993) found that deteriorating forecasting accuracy
due to the incorporation of human judgment is mostly due to human subjects reading systematic patterns
into the noise of time series. Although previous research has shown that machine algorithms are more
reliable and forecast better future events (see Meehl (1954) and Grove, Zald, Lebow, Snitz, and Nelson
(2000)), human beings often put higher trust in human advice, as shown by Eastwood, Snook, and Luther
(2012) and Diab, Pui, Yankelovich, and Highhouse (2011). In his research on trust in algorithmic decision
aids, Sheridan (1988) identified reliability, robustness, validity, transparency, understandability,
usefulness, and utility as the main drivers of trust. Whereas reliability has been confirmed extensively by
Lee and Morray (1992) and Muir (1994), Seong and Bisantz (2008) also addressed transparency,
understandability, and validity, and found that humans put more trust in algorithms that can be understood
by users and which perform consistently well.

Dietvorst, Simmons, and Massey (2016) and Dietvorst, Simmons, and Massey (2015) derived two
main insights regarding humans' readiness to use machine algorithms. First, people lose trust and

confidence in machine algorithms faster than in human advisors, once they see model forecasting errors.
This phenomenon has also been discussed by Alvarado-Valencua and Barrero (2014), who have found
disuse of computer models in forecasting with high task complexity and lower system performance. The
authors suggested explaining the computer models and showing past performance to users, although they
conceded that the delivery of this information is still controversial. Low trust in machine models implies
that although human forecasters have been found to be inferior to machine algorithms, they are more
likely to be trusted by advice-seekers if the algorithm does not have an impeccable record of forecasting
performance. This insight, in combination with the common forecasting superiority of machine
algorithms, can be used to make a case against sharing the past forecasting performance of the machine
models used.

The second insight concerns overcoming algorithm aversion. Dietvorst, Simmons, and Massey (2016)
have shown that offering the human user the possibility to adjust or modify the algorithm makes them
more likely to use the machine output. When confronted with the choice of using the machine output most
subjects declined to use it after seeing the algorithm err. Having been given the opportunity to slightly
modify the output, this aversion decreased (Dietvorst, Simmons, & Massey, 2016). This suggests a
potential tradeoff between how much users are being allowed to tweak the algorithm and forecasting
accuracy. In domains where machine models are more accurate than human forecasters, one could
sacrifice some accuracy in order to make human forecasters use machine recommendations.

A research project investigating how humans respond to algorithmic recommendations found the
opposite result. Logg, Minson, and Moore (2019) found that when tasked with providing numeric
estimates about a visual stimulus, the popularity of songs, and about romantic attraction, human subjects
preferred algorithmic advice over human expert advice. This appreciation of machine generated
recommendations decreased when the subjects had to choose between their own judgment and the
machine's, and if the subject was knowledgeable in forecasting (Logg, Minson, & Moore, 2019).

6. AGGREGATION METHODS

Decision makers tend to ask multiple experts for an opinion to negate bias and to obtain a more objective opinion. A key research question in this setting is how to aggregate the experts' and the decision maker's opinion into one. As pointed out in the section on human forecasting, most of the techniques comprise elicitation and aggregation procedures. The aggregation rules can either be qualitative, for example a group discussion to reach consensus, or quantitative, i.e. using a mathematical method. We focus on quantitative aggregation models because they tend to be more tractable. Furthermore, machine model forecasts can be aggregated easier using quantitative rules, while making sure that the overall forecast is replicable. Therefore, we exclude behavior aggregation from this survey and assume that the successive rules both apply to human and machine forecasts.

Clemen and Winkler (1999) discuss mathematical approaches to aggregate the opinions of multiple human experts and distinguish between axiomatic and Bayesian approaches. One of the main obstacles in aggregating multiple opinions is the potential dependence between the forecasters (Morris (1977), Werner, Bedford, Cooke, Hanea, and Morales-Nápoles (2017)). Opinions and advice originating from highly correlated sources are unlikely to improve forecasting accuracy, implying that advice from independent sources is particularly beneficial (Yaniv I. , 2004). Nevertheless, as long as correlation between forecasters is not perfectly positive, adding more forecasters increases forecasting accuracy (Broomell & Budescu, 2009).

[revised manuscript text omitted]

normal as well, which might not be realistic for each forecasting problem. Furthermore, the covariance
between forecasters needs to be estimated using past data (Clemen & Winkler, 1990).

**6.1.4. MAXIMUM ENTROPY AGGREGATION**

Maximum Entropy and Minimum Cross-Entropy methods have had a large share of literature
coverage and particularly in the assignment of prior probabilities in decision analysis using partial
information (see for example Jaynes (1968), Levy and Deliç (1994), Myung, Ramamoorti, and Bailey
(1996) and Gzylyter Horst, and Molina (2016), Abbas (2002), Abbas (2005), Abbas (2006), Abbas (2009),
Abbas, Cadenbach, and Salimi (2017)). The entropy measure attains its maximum when all outcomes are
equally likely. Once more information is provided, the entropy decreases and reaches zero when full
knowledge is achieved. The minimum cross-entropy approach finds a distribution that satisfies some
given constraints and is closest to a target distribution according to the Kullback Leibler divergence.

[revised manuscript text omitted]

6.2.2. LOG-LINEAR OPINION POOLS

The idea behind log-linear opinion pools is similar to the linear version, with the difference being multiplicative. Instead of additive, averaging (Zagorecki & Druzdzel, 2004). Similar to the linear opinion pool, the problem with this form of aggregation is its assumption of independence between forecasters, which might not hold as forecasters might draw from the same source of information. Furthermore, as Abbas (2009) and Etiminani, Naghibzadeh, and Pena (2013) have pointed out, the aggregate probability might result in a value of zero, if one forecaster assigns a zero probability to an event. To avoid this problem forecasts of 0 are replaced by an arbitrarily small value, ϵ .

6.2.3. DEMOCRATIC OPINION POOLS

The approach developed by Etiminani, Naghibzadeh, and Pena (2013) was used for the aggregation of parameters in Bayesian networks. The algorithm forms clusters of the forecasters' opinions and determines the cluster containing the most forecasters. Once the largest group is identified, a linear aggregation of their opinions is applied. Because of the reduction in the number of forecasters contributing to the aggregate, the authors claim that the algorithm is superior when it comes to speed (Etiminani, Naghibzadeh, & Pena, 2013). Their second claim, that the accuracy of the resulting forecast is higher, could be questionable as the algorithm might not only eliminate malicious and poor forecasters, but also independent diverging opinions that would balance some errors and biases.

6.2.4. DISCUSSION OF ACCURACY OF DIFFERENT MATHEMATICAL AGGREGATION METHODS

Several papers have shown that combining model forecasts improves forecast accuracy relative to a forecast provided by one forecaster (Hendry and Clements (2004), Makridakis and Hibon (2000), Timmermann (2013)). Clemen and Winkler (1999) and Newbold and Granger (1974) suggest that comparatively simple averaging methods that ignore correlations in their estimation can improve the accuracy of forecasts. Similarly, Hendry and Clements (2004) claim that simple averaging often outperforms sophisticated models. In contrast to the widely-held belief that (log-) linear pools are adequate aggregation methods, Wilson (2017) found empirical evidence that Bayesian approaches are

more accurate if there is dependence among forecasters. Morris (1977) and Werner, Bedford, Cooke,
Hanea, and Morales-Nápoles (2017) both highlight the necessity to incorporate dependence in elicitation
and aggregation of opinion. Bunn (1987), Goodwin (2000), and Davis-Stober, Budescu, Dana, and
Broomell (2014) found that the aggregation of opinion using a mathematical approach is most effective
when the forecasts are negatively correlated.

Clearly, there seems to be a trade-off between forecasting accuracy and the difficulty of aggregation
algorithm. Whereas linear aggregation methods are computationally easy, in presence of dependence the
marginal contribution of each additional forecaster decreases (Johnson, Budescu, & Wallsten, 2001). On
the other hand, Bayesian approaches offer a normative framework and are deemed more accurate while
being difficult to implement. Maximum entropy aggregation procedures have not been studied
extensively in the literature, but should deserve more attention because of their freedom from assumptions
that might add more uncertainty to the overall forecast.

**7. Summary and Conclusions**

Taking the number of publications as an indicator, this review found that interest in forecasting
techniques and applications has increased by a factor of ten over the previous forty years. The majority of
application focused papers concerned themselves with forecasting demand, weather, and electricity load.
With the increase of machine forecasting models came an interest in the performance difference between
human and machine forecasting, their limits, and benefits of combining them. The promise of combining
human and machine forecasting lies in the mutual balancing of their strengths and limitations. Human
forecasters are able to use contextual data to inform their opinion, something that machine models are not
capable of. Also, humans tend to outperform machine forecasting accuracy if quantitative data is sparse.
Contrastingly, machine forecasting techniques can survey and learn from vast quantitative data sets that
would overwhelm human cognitive abilities. In medicine for example, neural networks are being trained
to determine whether a patient has cancer. A physician then uses these results to inform and update her
own opinion on how to proceed. In a 2016 contest to detect metastatic breast cancer, the machine model

predicted with 92.5% and the physician with 96.6% accuracy. After combining human and machine
forecasts, accuracy was 99.5%, a 85% decrease of human error rate highlighting the potential gains from
human/machine-forecasting (Wang, Khosla, Gargeya, Irshad, & Beck, 2016).

We can summarize the results of our review of the literature on machine forecasting, human
forecasting, interactions of human and machines in the forecasting domain, by several stylized points.

*Choice of appropriate machine model:* We reviewed the most common machine models that are
being applied in the forecasting domain, differing in their area of applicability and construction difficulty.
The most common distinction can be drawn between time-series models, similarity-based/correlational
models, and a combination of these two. Therefore, a forecaster or decision maker needs to consider the
forecasting problem and available resources when deciding on an appropriate machine model. The choice
of model is also dependent on the availability and quality of data.

*Interaction between machine models and human forecasters:* There has been significant research on
the interaction between machine models and humans, such as the effect of past machine performance on
humans' readiness to use the provided information. Studies have shown that human forecasters exhibit a
higher readiness to use recommendations provided by models if they understand how the model works
and if they can tweak it to a certain amount. Further research has to be conducted in the field of algorithm
usage by human forecasters, such as how much updating occurs depending on trust in the algorithm, and
the amount of information provided.

*Training and incentivization of human forecasters:* Considering human forecasters, we have
discussed how the accuracy of forecasts benefits from incentive schemes that promote strategic behavior
of the forecasters, as well as the positive effect of training. Forecasting accuracy by human forecasters
could be improved significantly if the forecasters are trained in basic probability theory and de-biasing
techniques.

*Aggregation of opinions:* There is no consensus in the scientific community about which method is
the most accurate or efficient. On the one hand, non-Bayesian approaches are very appealing to users, as
they are computationally easy, and have been used in many research publications. A multitude of Non-

Bayesian aggregation rules, such as contribution weights or democratic opinion pools, were devised to
improve accuracy since the last comprehensive survey on opinion aggregation. On the other hand,
Bayesian approaches are normative, but are usually bugged down by high computational effort, and in the
case of the normal and copula aggregation method require several assumptions to be functional. This
review added recent methods such as Maximum Entropy Aggregation and contributed weights to the
extant surveys on aggregation rules.

We conclude with several directions for future research. Most areas identified as important to
human/machine-forecasting have been researched in depth, but the interactions between them requires
attention. For example, there has to be an assessment of machine models in a holistic context, i.e. which
machine model is most suitable given a certain training of human forecasters under a certain incentive
scheme and a specific aggregation method. Furthermore, the question of how much weight or significance
should be assigned to machine models and how much to the human forecasters has to be discussed and
addressed more directly, as well as rules to aggregate opinions when the human expert provides a point
forecast and the machine model a probability density for example. This question does not only matter in
the light of forecasting accuracy, but also carries ethical significance. Also, aggregation methods should
be discussed in conjunction to different scoring rules, especially because some approaches determine the
weighting of forecasters by their individual scores. We hope that this literature review motivates the
investigation of these interactions, such that a decision maker can choose the appropriate and holistic
forecasting design in the future.

**Acknowledgements**

The authors acknowledge the support of IARPA-BAA-16-02 and the National Science
Foundation award CMMI 16-29752.

References

- Önkal, D., Gönül, M. S., Goodwin, P., Thomson, M., & Öz, E. (2017). Evaluating expert advice in forecasting: Users' reactions to presumed vs. experienced credibility. *International Journal of Forecasting*, 33, 280-297.
- Önkal, D., Goodwin, P., Thomson, M., Gönül, S., & Pollock, A. (2009). The Relative Influence of Advice From Human Experts and Statistical Methods on Forecast Adjustments. *Journal of Behavioral Decision Making*, 22, 390-409.
- Ægisdóttir, S., White, M. J., Spengler, P. M., Maugherman, A. S., Anderson, L. A., Cook, R. S., . . . Rush, J. D. (2006). The Meta-Analysis of Clinical Judgment Project: Fifty-Six Years of Accumulated Research on Clinical Versus Statistical Prediction. *The Counseling Psychologist*, 34(3), 341-382.
- Abbas, A. E. (2003). Entropy Methods for Univariate Distributions in Decision Analysis. *AIP Conference Proceedings* 659, 339 (pp. 339-349). Moscow, ID: American Institute of Physics.
- Abbas, A. E. (2006). Entropy methods for joint distributions in decision analysis. *IEEE Transactions on Engineering Management*, 53(1), 146-159.
- Abbas, A. E. (2009). A Kullback-Leibler View of Linear and Log-Linear Pools. *Decision Analysis*, 6(1), 25-37.
- Abbas, A. E., Budescu, D. V., Yu, H.-T., & Haggerty, R. (2008). A Comparison of Two Probability Encoding Methods: Fixed Probability vs. Fixed Variable Values. *Decision Analysis*, 5(4), 190-202.
- Abbas, A. E., Cadenbach, A. H., & Salimi, E. (2017). A Kullback-Leibler View of Maximum Entropy and Maximum Log-Probability Methods. *Entropy*, 19(5), 232-246.
- Abou-Moustafa, K. T., De La Torre, F., & Ferrie, F. P. (2010). Designing a metric for the difference between Gaussian densities. In J. Angeles, B. Boulet, J. J. Clark, J. Kövecses, & K. Siddiqi (Eds.), *Brain, body and machine* (pp. 57-70). Berlin: Springer.
- Agarwal, R., & Prasad, J. (1997). The Role of Innovation Characteristics and Perceived Voluntariness in the Acceptance of Information Technologies. *Decision Sciences*, 28(3), 557-582.
- Agmon, N., Alhassid, Y., & Levine, R. (1979). An Algorithm for Finding the Distribution of Maximal Entropy. *Journal of Computational Physics*, 30, 250-258.
- Alvarado-Valencia, J. A., & Barrero, L. H. (2014). Reliance, trust and heuristics in judgmental forecasting. *Computers in Human Behavior*, 36, 102-113.
- Arbenz, P., & Canestraro, D. (2012). Estimating copulas for insurance from scarce observations, expert opinion, and prior information: A Bayesian approach. *Astin Bulletin*, 42(1), 271-290.
- Armstrong, J. S. (1980). The seer-sucker theory: the value of experts in forecasting. *Technology Review*, 82, 16-24.
- Armstrong, J. S. (1983). Relative accuracy of judgmental and extrapolative methods in forecasting annual earnings. *Journal of Forecasting*, 2, 437-447.
- Armstrong, J. S. (1985). *Long range forecasting: From crystal ball to computer* (2 ed.). New York: Wiley.
- Armstrong, J. S., & Collopy, F. (1992). Error measures for generalizing about forecasting methods: Empirical comparisons. *International Journal of Forecasting*, 8, 69-80.
- Aumann, R. J. (1976). Agreeing to Disagree. *Annals of Statistics*, 4, 1236-1239.
- Ayton, P., & Önkal, D. (1996). Effects of expertise on forecasts and confidence in forecasts. *16th International Symposium on Forecasting*. Istanbul.

Bar-Hillel, M. (1980). The base-rate fallacy in probability judgements. *Acta Psychologica*, 44(3), 211-
233.
- Barlas, Y. (2009). Systemic feedback modeling for policy analysis. In Y. Barlas (Ed.), *System Dynamics -*
*Encyclopedia of Life Support Systems* (pp. 1131-1175). Istanbul: EOLSS Publishers Co Ltd.
- Bass, F. M., King, C. W., & Pessemier, E. A. (1968). *Applications of the sciences in marketing*
*management* (1st Edition ed.). New York: Wiley.
Bedford, T., Daneshkhah, A., & Wilson, K. (2016). Approximate uncertainty modeling in risk analysis
with vine copulas. *Risk Analysis*, 36(4), 792-815.
- Bendor, T. K., Metcalf, S. S., Fontenot, L. E., Sangunett, B., & Hannon, B. (2006). Modeling the spread
of the Emerald Ash Borer. *Ecological Modelling*, 197(1), 221-236.
- Bertsekas, D. P., & Tsitsiklis, J. N. (1996). *Neuro-dynamic programming*. Cambridge: Athena Scientific.
- Bickel, J. E. (2007). Some Comparisons among Quadratic, Spherical, and Logarithmic Scoring Rules.
*Decision Analysis*, 4(2), 49-65.
- Blackburn, R., & Stokes, D. (2000). Breaking Down the Barriers: Using Focus Groups to Research Small
and Medium-sized Enterprises. *International Small Business Journal*, 19(1), 44-67.
- Blattberg, R. C., & Hoch, S. J. (1990). Database models and managerial intuition: 50% model + 50%
manager. *Management Science*, 36, 887-899.
- Boero, G., Smith, J., & Wallis, K. F. (2011). Scoring rules and survey density forecasts. *International*
*Journal of Forecasting*, 27, 379-393.
- Bolger, F., & Wright, G. (2011). Improving the Delphi process: Lessons from psychological research.
*Technological Forecasting & Social Change*, 78, 1500-1513.
- Bonaccio, S., & Dalal, R. S. (2006). Advice taking and decision-making: An integrative literature review,
and implications for the organizational sciences. *Organizational Behavior and Human Decision*
*Processes*, 101, 127-151.
- Borchev, A. (2013). *The Big Book of Simulation Modeling - Multimethod Modeling with AnyLogic 6* (1st
ed.). New York: Anylogic.
- Box, G. E., Jenkins, G. M., Reinsel, G. C., & Ljung, G. M. (2015). *Time Series Analysis: Forecasting and*
*Control* (5th Edition ed.). Hoboken: John Wiley & Sons.
- Brier, G. W. (1950). Verification of forecasts expressed in terms of probabilities. *Monthly Weather*
*Review*, 78, 1-3.
- Broomell, S. B., & Budescu, D. V. (2009). Why are expert correlated? Decomposing correlations between
judges. *Psychometrika*, 74(3), 531-553.
- Brown, L. (1996). Analyst forecasting errors and their implications for security analysts: An alternative
perspective. *Financial Analysts Journal*, 1, 40-47.
- Brown, R. G. (1956). *Exponential Smoothing for Predicting Demand*. Cambridge, Massachusetts: Arthur
D. Little.
- Budescu, D., & Chen, E. (2015). Identifying expertise to extract the wisdom of crowds. *Management*
*Science*, 20(1), 37-46.
- Buja, A., Stuetzle, W., & Shen, Y. (2005, November 3). *University of Pennsylvania*. Retrieved July 17,
2018, from <http://www-stat.wharton.upenn.edu/~buja/PAPERS/paper-proper-scoring.pdf>
- Bunn, D. (1987). Expert user of forecasts: Bootstrapping and linear models. In G. Wright, & P. Ayton,
*Judgmental Forecasting* (pp. 229-241). Chichester: Wiley.

Carbone, R., Andersen, A., Corriveau, Y., & Corrsen, P. P. (1983). Comparing for different time series
methods the value of technical expertise, individualized analysis and judgmental adjustment.
*Management Science*, 20, 229-566.
Cavalcante, J., Celestino, J., & Patel, A. (2017). Performance Management of Optical Transport Networks
Through Time Series Forecasting. *IEEE 31st International Conference on Advanced Information*
*Networking and Applications*. Taipei.
Chambers, J. C., Mullick, S. K., & Smith, D. D. (1971, July). How to choose the right forecasting
technique. *Harvard Business Review*.
Chang, W., Chen, E., Mellers, B., & Tetlock, P. (2016). Developing expert political judgement: the
impact of training and practice on judgemental accuracy in geopolitical forecasting tournaments.
*Judgement and Decision Making*, 11(5), 509-527.
Chase, Jr., C. W. (2013). *Demand-Driven Forecasting: A Structured Approach to Forecasting*. New
York: Wiley & Sons, Inc.
Chatfield, C. (1988). What is the 'best' method of forecasting? *Journal of Applied Statistics*, 15(1), 19-38.
Chatfield, R., Hein, S., & Moyer, S. (1990). Long term earnings forecasts in the electric utility industry:
Accuracy and valuation implications. *Financial Review*, 25(3), 421-439.
Chen, E., Budescu, D. V., Lakshmikanth, S. K., Mellers, B. A., & Tetlock, P. E. (2016). Validating the
Contribution-Weighted Model: Robustness and Cost-Benefit Analysis. *Decision Analysis*, 13(2),
128-152.
Chou, Y.-I. (1975). *Statistical Analysis* (2nd Edition ed.). New York: Holt, Rinehart & Winston of
Canada.
Clelland, R. C., DeCani, J. S., & Brown, F. E. (1973). *Basic statistics with business applications* (2nd
Edition ed.). New York: Wiley.
Clemen, R. T., & Winkler, R. L. (1999). Combining Probability Distributions from Experts in Risk
Analysis. *Risk Analysis*, 19(2), 187-203.
Clemen, R., & Winkler, R. (1990). Unanimity and compromise among probability forecasters.
*Management Science*, 36, 767-779.
Collins, B. E., & Guetzkow, H. (1964). *A Social Psychology of Group Processes for Decision-Making*.
New York: John Wiley and Sons.
Crooks, A. T., & Hailegiorgis, A. B. (2014). An agent-based modeling approach applied to the spread of
cholera. *Environmental Modelling and Software*, 62, 164-177.
Cross, S. S., Harrison, R. T., & Kennedy, R. L. (1995). Introduction to neural networks. *Lancet*, 346,
1075-1079.
Dagum, E. B. (1983). Spectral Properties of the Concurrent and Forecasting Seasonal Linear Filters of the
X-11-Arima Method. *The Canadian Journal of Statistics*, 11(1), 73-90.
Dalkey, N. C. (1969). *The Delphi model: An experimental study of group opinions*. The Rand
Corporation. Santa Monica: The Rand Corporation.
Damas, M., Salmeron, M., Diaz, A., Ortega, J., Prieto, A., & Olivares, G. (2000). Genetic algorithms and
neuro-dynamic programming: application to water supply networks. *Proceeding of 2000*
*Congress on Evolutionary Computation*. La Jolla: IEEE.
Dantas, T. M., & Oliveira, F. L. (2018). Improving time series forecasting: An approach combining
bootstrap aggregation, clusters and exponential smoothing. *International Journal of Forecasting*,
34(4), 748-761.

Davis-Stober, C. P., Budescu, D. V., Broomell, S. B., & Dana, J. (2015). The Composition of Optimally
Wise Crowds. *Decision Analysis*, *12*(3), 130-143.
- Davis-Stober, C. P., Budescu, D. V., Dana, J., & Broomell, S. B. (2014). When Is a Crowd Wise?
*Decision*, *1*(2), 79-101.
Dawes, R. (1971). A case study of graduate admissions: Application to three principles of human decision
making. *American Psychologist*, *26*(2), 180-188.
- de Menezes, L. M., Bunn, D. W., & Taylor, J. W. (2000). Review of guidelines for the use of combined
forecasts. *European Journal of Operations Research*, *120*, 190-204.
- de Rigo, D., Rizzoli, A. E., Soncini-Sessa, R., Weber, E., & Zenesi, P. (2001). Neuro-dynamic
programming for the efficient management of reservoir networks. *Proceedings of MODSIM*
*2001, International Congress on Modelling and Simulation*. Canberra: Modelling and Simulation
Society of Australia and New Zealand.
- del Sagrado, J., & Moral, S. (2003). Qualitative Combination of Bayesian Networks. *International*
*Journal of Intelligent Systems*, *18*, 237-249.
- Delbecq, A. L., Van de Ven, A. H., & Gustafson, D. H. (1975). *Group Techniques for Program Planning*.
Glenview, IL: Scott Foresman.
- Deng, G., & Ferris, M. C. (2008). Neuro-dynamic programming for fractionated radiotherapy planning. In
C. J. Alves, P. M. Pardalos, & L. N. Vicente (Eds.), *Optimization in Medicine. Springer*
*Optimization and Its Applications* (Vol. 12, pp. 47-70). New York: Springer.
- Diab, D., Pui, S., Yankelovich, M., & Highhouse, S. (2011). Lay perceptions of selection decision aids in
U.S. and non-U.S. samples. *International Journal of Selection and Assessment*, *19*, 209-216.
- Diebold, F. X., & Mariano, R. S. (1995). Comparing Predictive Accuracy. *Journal of Business and*
*Economic Statistics*, *13*, 253-265.
- Dietvorst, B. J., Simmons, J. P., & Massey, C. (2015). Algorithm Aversion: People Erroneously Avoid
Algorithms After Seeing Them Err. *Journal of Experimental Psychology: General*, *144*(1), 114-
126.
- Dietvorst, B. J., Simmons, J. P., & Massey, C. (2016). Overcoming Algorithm Aversion: People Sill Use
Imperfect Algorithms If They Can (Even Slightly) Modify Them. *Management Science, Articles*
*in Advance*, 1-17.
- Druzdzel, M. J., & Van der Gaag, L. C. (1995). Elicitation of probabilities for belief networks:
Combining qualitative and quantitative information. In *Proceedings of the eleventh conference on*
*uncertainty in artificial intelligence* (pp. 141-148). Morgan Kaufmann Publishers Inc.
- Durante, F., & Sempi, C. (2015). *Principles of copula theory*. Boca Raton, FL, USA: CRC Press.
- Eastwood, J., Snook, B., & Luther, K. (2012). What people want from their professionals: Attitudes
toward decision-making strategies. *Journal of Behavioral Decision Making*, *25*, 458-468.
- Eggleton, I. R. (1982). Intuitive time series extrapolation. *Journal of Accounting Research*, *20*, 68-102.
- Epstein, E. S. (1969). A Scoring System for Probability Forecasts of Ranked Categories. *Journal of*
*Applied Meteorology*, *8*, 985-987.
- Etiminani, K., Naghibzadeh, M., & Pena, J. M. (2013). DemocraticOP: A democratic way of aggregating
Bayesian network parameters. *International Journal of Approximate Reasoning*, *54*(2013), 602-
614.
- Fan, Y., Budescu, D. V., Mandel, D. R., & Himmelstein, M. (2019). Improving Accuracy by Coherence
Weighting of Direct and Ration Probability Judgments. *Decision Analysis, forthcoming*.

Fogg, B. (1999). Persuasive technologies - Now is your chance to decide what they will persuade us to do
- and how they'll do it. *Communications of the ACM*, 42, 26-29.
French, S. (1985). Group consensus probability distributions: A critical survey. In J. Bernardo, M.
DeGroot, D. Lindley, & A. Smith, *Bayesian Statistics 2* (pp. 183-197). Amsterdam, North-
Holland.
French, S. (2011). Aggregating expert judgement. *Revista de la Real Academia de Ciencias Exactas,*
*Fisicas y Naturales. Serie A. Matematicas*, 105(1), 181-206.
Garthwaite, P. H., Kadane, J. B., & O'Hagan, A. (2005). Statistical Methods for Eliciting Probability
Distributions. *Journal of the American Statistical Association*, 100(470), 680-701.
Genest, C., & Schervish, M. (1985). Modeling expert judgements for Bayesian updating. *Annals of*
*Statistics*, 13, 1198-1212.
Gneiting, T. (2011). Making and Evaluating Point Forecasts. *Journal of the American Statistical*
*Association*, 106(494), 746-762.
Gneiting, T., & Raftery, A. (2007). Strictly proper scoring rules, prediction, and estimation. *Journal of the*
*American Statistical Association*, 102, 359-378.
Gneiting, T., & Raftery, A. E. (2007). Strictly Proper Scoring Rules, Prediction, and Estimation. *Journal*
*of the American Statistical Association*, 102(477), 359-378.
Goodwin, P. (2000). Correct or combine? Mechanically integrating judgmental forecasts with statistical
methods. *International Journal of Forecasting*, 16, 85-99.
Goodwin, P., & Wright, G. (1993). Improving judgmental time series forecasting: A review of the
guidance provided by research. *International Journal of Forecasting*, 147-161.
Grove, W. M., Zald, D. H., Lebow, B. S., Snitz, B. E., & Nelson, C. (2000). Clinical Versus Mechanical
Prediction: A Meta-Analysis. *Psychological Assessment*, 12(1), 19-30.
Guo, H. (2015). Accelerated Continuous Conditional Random Fields for Load Forecasting. *IEEE*
*Transactions on Knowledge and Data Engineering*, 27(8), 2023-2033.
Gzyl, H., ter Horst, E., & Molina, G. (2016). Inferring probability densities from expert opinion. *Applied*
*Mathematical Modelling*, 43, 306-320.
Hadley, G. (1968). *Introduction to business statistics*. San Francisco: Holden-Day.
Hanea, A., Morales Nápoles, O., & Abadei, D. (2015). Non-parametric Bayesian networks: Improving
theory and reviewing applications. *Reliability Engineering & System Safety*, 144, 265-284.
Harte, D., & Vere-Jones, D. (2005). The Entropy Score and its Uses in Earthquake Forecasting. *pure and*
*applied geophysics*, 162(6-7), 1229-1253.
Harvey, A. (1989). *Forecasting, structural time series models and the Kalman filter*. Cambridge:
Cambridge University Press.
Harvey, N., & Fischer, I. (1997). Taking advice: Accepting help, improving judgment and sharing
responsibility. *Organizational Behavior and Human Decision Processes*, 70, 117-133.
Hastie, T., Tibshirani, R., & Friedman, J. (2017). *The Elements of Statistical Learning - Data Mining,*
*Inference, and Prediction*. New York: Springer.
Hendry, D., & Clements, M. (2004). Pooling of forecasts. *Econometrics Journal*, 7, 1-31.
Henrion, M. (1989). Some practical issues in constructing belief networks. In *Proceedings of the third*
conference on uncertainty in artificial intelligence (pp. 161-173). Elsevier Science Publishing
Company.
Highhouse, S. (2008). Stubborn reliance on intuition and subjectivity in employee selection.
Organizational and Industrial Psychology, 1(3), 333-342.

Holt, C. C. (2004). Forecasting seasonals and trends by exponentially weighted moving averages.
*International Journal of Forecasting*, 20(1), 5-10.
Homer, J. B., & Hirsch, G. B. (2006). System dynamics modeling for public health: background and
opportunities. *The American Journal of Public Health*, 96(3), 452-458.
Hudak, R. P., Brooke, P. P., Finstuen, K., & Riley, P. (1993). Health care administration in the year 2000:
practitioners' views of future issues and job requirements. *Hospital & Health Services*
*Administration*, 38(2), 181-195.
Hull, J. C. (2015). *Options, Futures, and other Derivatives*. New Jersey: Pearson Education Inc.
Hutchings, A., Rosalind, R., Colin, S., & Nick, B. (2006). A Comparison of Formal Consensus Methods
Used for Developing Clinical Guidelines. *Journal of Health Services Research & Policy*, 11(4),
218-224.
Hyndman, R. L., & Athanasopoulos, G. (2013). *Forecasting: principles and practice*. O Texts.
IARPA. (2016). *Hybrid Forecasting Competition*. Retrieved August 27, 2018, from
<https://www.iarpa.gov/index.php/research-programs/hfc/hfc-baa>
Iosevich, S., Arutyunants, G., & Hou, Z. (2015). Dynamic Aggregation for Time Series Forecasting.
*IEEE International Conference on Big Data*. Santa Clara.
Janis, I. L. (1982). *Groupthink: Psychological Studies of Policy Decisions and Fiascoes*. Cengage
Learning: Boston.
Jaynes, E. T. (1968). Prior Probabilities. *IEEE Transactions on Systems Science and Cybernetics*, 4(3),
227-241.
Johnson, T. R., Budescu, D. V., & Wallsten, T. S. (2001). Averaging Probability Judgments: Monte Carlo
Analyses of Asymptotic Diagnostic Value. *Journal of Behavioral Decision Making*, 14, 123-140.
Jose, V. R., Grushka-Cockayne, Y., & Lichtendahl Jr., K. C. (2013). Trimmed Opinion Pools and the
Crowd's Calibration Problem. *Management Science*, 60(2), 463-475.
Jouini, M. N., & Clemen, R. T. (1996). Copula Models for Aggregating Expert Opinion. *Operations*
*Research*, 44(3), 444-457.
Kaastra, I., & Boyd, M. (1996). Designing a neural network for forecasting financial and economic time
series. *Neurocomputing*, 10(3), 215-236.
Kahneman, D. (2011). *Thinking fast and slow*. New York: Farrar, Straus and Giroux.
Kahneman, D., & Tversky, A. (1973). On the psychology of prediction. *Psychological Review*, 80(4),
237-251.
Kahneman, D., & Tversky, A. (1984). Choices, values, and frames. *American Psychologist*, 39(4), 341-
350.
Kaplan, S. (1990). 'Expert information' vs. 'expert opinion': Another approach to the problem of
eliciting/combining/using expert knowledge in PRA. *Journal of Reliability Engineering and*
*System Safety*, 39.
Karvetski, C. W., Olson, K. C., Mandel, D. R., & Twardy, C. R. (2013). Probabilistic Coherence
Weighting for Optimizing Expert Forecasts. *Decision Analysis*, 10(4), 305-326.
Kauko, K., & Palmroos, P. (2014). The Delphi method in forecasting financial markets-An experimental
study. *International Journal of Forecasting*, 30, 313-327.
- Ke-Lin, D., & Swamy, M. N. (2014). *Neural Networks and Statistical Learning*. New York: Springer.
- Korkmaz, G., Cadena, J., Kuhlman, C. J., Marathe, A., Vullikanti, A., & Ramakrishnan, N. (2015).
Combining Heterogeneous Data Sources for Civil Unrest Forecasting. *IEEE/ACM International*
Conference on Advances in Social Network Analysis and Mining. Paris.

Kotz, S., & Van Dorp, J. (2010). Generalized diagonal band copulas with two-sided generating densities.
*Decision Analysis*, 7(2), 196-214.
- Kotz, S., Read, C. B., Balakrishnan, N., Vidakovic, B., & Johnson, N. L. (2004). *Encyclopedia of*
*Statistical Sciences*. New York: John Wiley & Sons, Inc.
- Krueger, R. A., & Casey, M. A. (2014). *Focus Groups: A Practical Guide for Applied Research*.
Thousand Oaks: Sage Publications.
- Kuncel, N. R., Klieger, D. M., Connelly, B. S., & Ones, D. S. (2013). Mechanical Versus Clinical Data
Combination in Selection and Admissions Decisions: A Meta-Analysis. *Journal of Applied*
*Psychology*, 98(6), 1060-1072.
- Landeta, J., Barrutia, J., & Lertxundi, A. (2011). Hybrid Delphi: A methodology to facilitate contribution
from experts in professional contexts. *Technological Forecasting & Social Change*, 78, 1629-
1641.
- Lawrence, M., Goodwin, P., O'Connor, M., & Önkal, D. (2006). Judgmental forecasting: A review of
progress over the last 25 years. *International Journal of Forecasting*, 22, 493-518.
- Lee, J. D., & Moray, N. (1992). Trust, control strategies and allocations of functions in human-machine
systems. *Ergonomics*, 35(10), 1243-1270.
- Leuthold, R. M., MacCormick, A. A., Schmitz, A., & Watts, G. (1970). Forecasting Daily Hog Proces
and Quantities: A Study of Alternative Forecasting Techniques. *Journal of the American*
*Statistical Association*, 64(329), 90-107.
- Levy, W. B., & Deliç, H. (1994). Maximum Entropy Aggregation of Individual Opinons. *IEEE*
*Transactions on Systems, Man and Cybernetics*, 24(4), 606-613.
- Lichtendahl Jr., K. C., Grushka-Cockayne, Y., & Pfeifer, P. E. (2013). The Wisdom of Competitive
Crowds. *Operations Research*, 61(6), 1383-1398.
- Lichtendahl, K., & Winkler, R. (2007). Probability elicitation, scoring rules, and competition among
forecasters. *Management Science*, 53(11), 1745-1755.
- Lichtenstein, S., Slovic, P., Fischhoff, B., Layman, M., & Combs, B. (1978). Judged frequency in lethal
events. *Journal of Experimental Psychology*, 4(6), 551-578.
- Linstone, H. A., & Turoff, M. (1975). *The Delphi Method: Techniques and Applications*. Reading:
Addison-Wesley.
- Linstone, H. A., & Turoff, M. (2011). Delphi: A brief look backward and forward. *Technological*
*Forecasting & Social Change*, 78, 1712-1719.
- Logg, J. M., Minson, J. A., & Moore, D. A. (2019). Algorithm appreciation: People prefer algorithmic to
human judgment. *Organizational Behavior and Human Decision Processes*, 90-103.
- Lunsford, D. A., & Fussell, B. C. (1993). Marketing business services in Central Europe: the challenge: a
report of expert opinion. *Journal of Services Marketing*, 7(1), 13-21.
- Makridakis, S., & Hibon, M. (2000). The M3-competition: results, conclusions and implications.
*International Journal of Forecasting*, 16, 451-476.
- Makridakis, S., & Hibon, M. (2000). The M3-Competition: results, conclusions and implications.
*International Journal of Forecasting*, 16, 451-476.
- Makridakis, S., Andersen, A., Carbone, R., Fildes, R., Hibon, M., Lewandowski, R., . . . Winkler, R.
(1982). The Accuracy of Extrapolation (Time Series) Methods: Results of a Forecasting
Competition. *Journal of Forecasting*, 1(2), 111-153.

Makridakis, S., Chatfield, C., Hibon, M., Lawrence, M., Mills, T., Ord, K., & Simmons, L. F. (1993). The
M2-Competition: A real-time judgmentally based forecasting study. *International Journal of*
*Forecasting*, 9, 5-22.
Makridakis, S., Spiliotis, E., & Assimakopoulos, V. (2018). Statistical and Machine Learning forecasting
methods: Concerns and ways forward. *PLoS ONE*, 13(3).
Makridakis, S., Wheelwright, S. C., & Hyndman, R. J. (1998). *Forecasting Methods and Applications*
(3rd Edition ed.). New York: John Wiley & Sons.
Martino, J. P. (1992). *Technological Forecasting for Decision Making*. New York: McGraw-Hill.
Matheson, J. E., & Winkler, R. L. (1976). Scoring Rules for Continuous Probability Distributions.
*Management Science*, 22(10), 1087-1096.
McCulloch, W., & Pitts, W. (1943). A Logical Calculus of Ideas Immanent in Nervous Activity. *Bulletin*
*of Mathematical Biophysics*, 5(4), 115-133.
McNees, S. K. (1987). Consensus forecasts: tyranny of the majority? *New England Economic*
*Review*(Nov), 15-21.
Meehl, P. (1954). *Clinical versus statistical prediction: A theoretical analysis and review of literature*.
Minneapolis, MN: University of Minnesota Press.
Meeuwissen, A. M., & Bedford, T. (1997). Minimally informative distributions with given rank
correlation for use in uncertainty analysis. *Journal of Statistical Computation and Simulation*,
57(1-4), 143-174.
Merkle, E. C., & Steyvers, M. (2013). Choosing a Strictly Proper Scoring Rule. *Decision Analysis*, 10(4),
292-304.
Miyoshi, T., & Matsubara, S. (2018). Dynamically Forming a Group of Human Forecasters and Machine
Forecaster for Forecasting Economic Indicators. *International Joint Conference on Artificial*
*Intelligence*. Stockholm.
Moder, J. J., & Rodgers, E. G. (1968). Judgment estimates of the moments of PERT type distributions.
*Management Science*, 15(2), 76-83.
Moore, C. C. (1987). *Group Techniques for Idea Building*. London: Sage.
Morales Nápoles, O., Kurowicka, D., & Roelen, A. (2008). Eliciting conditional and unconditional rank
correlations from conditional probabilities. *Reliability Engineering & System Safety*, 93(5), 699-
710.
Morales-Nápoles, O., & Worm, D. (2013). *Hypothesis testing of multidimensional probability*
*distributions*. TNO.
Morales-Nápoles, O., Worm, D., Hanea, A. M., & Kalkman, I. (2016). Calibration and combination of
expert's dependence estimates (under review).
Morris, P. A. (1974). Decision Analysis Expert Use. *Management Science*, 20(9), 1233-1241.
Morris, P. A. (1977). Combining Expert Judgements: A Bayesian Approach. *Management Science*, 23(7),
679-693.
Morris, P. A. (1983). An Axiomatic Approach to Expert Resolution. *Management Science*, 29(1), 24-32.
Muir, B. M. (1994). Trust in automation. Part I. Theoretical issues in the study of trust and human
intervention in automated systems. *Ergonomics*, 37(11), 1905-1922.
- Murphy, A. H., & Winkler, R. L. (1970). Scoring Rules in Probability Assessment and Evaluation. *Acta*
Psychologica, 34, 273-286.
- Myung, J., Ramamoorti, S., & Bailey Jr., A. D. (1996). Maximum Entropy Aggregation of Expert
Predictions. *Management Science*, 42(10), 1420-1436.

Nadkarni, S., & Shenoy, P. P. (2004). A causal mapping approach to constructing Bayesian networks.
*Decision Support Systems*, 38(2), 259-281.
- Neiderman, F., Brancheau, J. C., & Wetherbe, J. C. (1991). Information Systems Management Issues for
the 1990s. *MIS Quarterly*, 15(4), 475-500.
- Nelsen, R. B. (1999). *An Introduction to Copulas*. New York: Springer.
- Newbold, P., & Granger, C. (1974). Experience with forecasting univariate time series and the
combination of forecasts. *Journal of the Royal Statistical Society*, A(137), 131-149.
- North, H. Q., & Pyke, D. L. (1969, May-June). Probes of the technical future. *Harvard Business Review*,
p. 68.
- Nunnari, G., & Nunnari, V. (2017). Forecasting Monthly Sales Retail Time Series: A Case Study. *IEEE*
*19th Conference on Business Informatics*. Thessaloniki.
- O'Connor, M., Remus, W., & Griggs, K. (1993). Judgmental forecasting in times of change. *International*
*Journal of Forecasting*, 9, 163-172.
- Ojha, V. K., Abraham, A., & Snasel, V. (2017). Metaheuristic design of feedforward neural networks: A
review of two decades of research. *Engineering Applications of Artificial Intelligence*, 60, 97-
116.
- Olshfski, D., & Joseph, A. (1991). Assessing Training Needs of Executives Using the Delphi Technique.
*Public Productivity & Management Review*, 14(3), 297-301.
- Ottaviani, M., & Sørensen, P. (2006). The strategy of professional forecasting. *Journal of Financial*
*Economics*, 81, 441-466.
- Pankratz, A. (1991). *Forecasting with Dynamic Regression Models*. New York: John Wiley & Sons.
- Peña, J. M. (2017). Causal Effect Identification in Alternative Acyclic Directed Mixed Graphs.
*Proceedings of Machine Learning Research*, 73, 21-32.
- Phillips, L. D. (1984). A theory of requisite decision models. *Acta Psychologica*, 56, 29-48.
- Phillips, L. D. (1987). On the adequacy of judgmental forecasting. In *Judgmental Forecasting* (pp. 11-
30). Chichester, England: Wiley.
- Phillips, P. J., Yates, A. N., Hu, Y., Hahn, C. A., Noyes, E., Jackson, K., . . . O'To. (2018). Face
recognition accuracy of forensic examiners, superrecognizers, and face recognition algorithms.
*PNAS*, 115(24), 6171-6176.
- Pollock, D. S., Green, R. C., & Nguyen, T. (1999). *Handbook of Time Series Analysis, Signal Processing,*
*and Dynamics*. Cambridge, MA: Academic Press.
- Predd, J. B., Osherson, D. N., Kulkarni, S. R., & Poor, H. V. (2008). Aggregating Probabilistic Forecasts
from Incoherent and Abstaining Experts. *Decision Analysis*, 5(4), 177-189.
- Renooji, S. (2001). Probability elicitation for belief networks. *The Knowledge Engineering Review*, 16(3),
255-269.
- Ristovski, K., Radosavljevic, V., Vucetic, S., & Obradovic, Z. (2013). Continuous Conditional Random
Fields for Efficient Regression in Large Fully Connected Graphs. *Twenty-Seventh AAAI*
*Conference on Artificial Intelligence*. Washington.
- Robinson, N. (1999). The use of focus group methodology - with selected examples from sexual health
research. *Journal of Advanced Nursing*, 29(4), 905-913.
- Ross, S. M. (2014). *Introduction to probability models* (11th Edition ed.). Amsterdam, Boston: Elsevier.
- Rowe, G., & Wright, G. (1999). The Delphi technique as a forecasting tool: issues and analysis.
*International Journal of Forecasting*, 15, 353-375.

Sackman, H. (1975). *Delphi Critique: Expert Opinion, Forecasting and Group Process*. Lanham:
Lexington Books.
- Sadilek, A., Kautz, H., & Silenzio, V. (2012). Modeling Spread of Disease from Social Interactions.
Rochester: AAAI.
- Saito, M., & Sinha, K. C. (1991). Delphi Study on Bridge Condition Rating and Effects of Improvements.
*Journal of Transportation Engineering*, 117(3), 320-334.
- Sanders, N. R., & Manrodt, K. B. (2003). The efficacy of using judgmental versus quantitative
forecasting tasks. *Omega*, 31, 511-522.
- Sanders, N. R., & Ritzman, L. P. (1992). The need for contextual and technical knowledge in judgmental
forecasting. *Journal of Behavioral Decision Making*, 5(1), 39-52.
- Schweitzer, M., & Cachon, G. (2000). Decision bias in the newsvendor problem with a known demand
distribution: Experimental evidence. *Management Science*, 46(3), 404-420.
- Secomandi, N. (2000). Comparing neuro-dynamic programming algorithms for the vehicle routing
problem with stochastic demands. *Computers & Operations Research*, 27(11-12), 1201-1225.
- Seong, Y., & Bisantz, A. M. (2008). The impact of cognitive feedback on judgment performance and trust
with decision aids. *International Journal of Industrial Ergonomics*, 38, 608-625.
- Sheridan, T. B. (1988). Trustworthiness of command and control systems. *International Federation of*
*Automatic Control*. IFAC Man-Machine Systems.
- Sineglazov, V., Chumachenko, E., & Gorbatyuk, V. (2014). Using a mixture of experts' approach to solve
the forecasting task. *Aviation*, 18(3), 129-133.
- Slovic, P., & Fischhoff, B. (1977). On the psychology of experimental surprises. *Journal of Experimental*
*Psychology: Human Perception and Performance*, 3(4), 544-551.
- Smith, J. E. (1993). Moment Methods for Decision Analysis. *Management Science*, 39(3), 340-358.
- Soll, J. B., & Larrick, R. P. (2009). Strategies for Revising Judgment: How (and How Well) People Use
Others' Opinions. *Journal of Experimental Psychology*, 35(3), 780-805.
- Spencer, M. H. (1961). *Business and economic forecasting: an econometric approach*. Homewood,
Illinois: R.D. Irwin.
- Spetzler, C., & von Holstein, C.-A. S. (1975). Probability Encoding in Decision Analysis. *Management*
*Science*, 22(3), 340-358.
- Staël von Holstein, C.-A. S. (1970). A Family of Strictly Proper Scoring Rules Which Are Sensitive to
Distance. *Journal of Applied Meteorology*, 9, 360-364.
- Sterman, J. D. (2000). *Business Dynamics - Systems Thinking and Modeling for a Complex World* (1st
ed.). Boston: McGraw-Hill.
- Stiber, N. A., Small, M. J., & Pantazidou, M. (2004). Site-Specific Updating and Aggregation of
Bayesian Belief Network Models for Multiple Expert. *Risk Analysis*, 24(6), 1529-1537.
- Stone, M. (1961). The opinion pool. *Annals of Mathematical Statistics*, 32, 1339-1342.
- Surowiecki, J. (2001). *The Wisdom of Crowds*. New York: Anchor Books.
- Tesfatsion, L. (2002). Agent-Based Computational Economics: Growing Economies From the Bottom
Up. *Artificial Life*, 8(1), 55-82.
- Tesfatsion, L. (2003). Agent-based computational economics: modeling economies as complex adaptive
systems. *Information Sciences*, 149(4), 262-268.
- Tetlock, P. E. (2006). *Expert Political Judgment: How Good Is It? How Can We Know?* (2nd ed.).
Princeton: Princeton University Press.

Tetlock, P. E., & Gardner, D. (2016). *Superforecasting: The Art and Science of Prediction* (1st ed.). New
York: Broadway Books.
- Timmermann, A. (2013). Forecast combinations. In *Handbook of Economic Forecasting* (Vol. 1, pp. 135-
196). New York: Elsevier Science.
Turner, B. M., Steyvers, M., Merkle, E. C., Budescu, D. V., & Wallsten, T. S. (2014). Forecast
aggregation via recalibration. *Machine Learning*, 95(3), 261-289.
Tych, W., Pedregal, D., Young, P., & Davies, J. (2002). An unobserved component model for multi-rate
forecasting of telephone call demand: the design of a forecasting support system. *International*
*Journal of Forecasting*, 673-695.
Wallsten, T. S., & Budescu, D. V. (1983). Encoding subjective probabilities: A psychological and
psychometric review. *Management Science*, 29(2), 151-173.
Wang, D., Khosla, A., Gargeya, R., Irshad, H., & Beck, A. H. (2016, June 18). *Deep Learning for*
*Identifying Metastatic Breast Cancer*. Retrieved from Cornell University:
<https://arxiv.org/abs/1606.05718>
Wathern, C., & Burknell, J. (2002). Believe it or not: Factors influencing credibility on the web. *Journal*
*of the American Society for Information Science and Technology*, 53(2), 134-144.
Werner, C., Bedford, T., Cooke, R. M., Hanea, A. M., & Morales-Nápoles, O. (2017). Expert judgement
for dependence in probabilistic modelling: A systematic literature review and future research
directions. *European Journal of Operational Research*, 258, 801-819.
Wilson, K. J. (2017). An investigation of dependence in expert judgement studies with multiple experts.
*International Journal of Forecasting*, 33, 325-336.
Winkler, J., & Moser, R. (2016). Biases in future-oriented Delphi studies: A cognitive perspective.
*Technological Forecasting & Social Change*, 105, 63-76.
Winkler, R. L. (1968). The consensus of subjective probability distributions. *Management Science*, 15,
361-375.
Winkler, R. L. (1981). Combining Probability Distributions from Dependent Information Sources.
*Management Science*, 27(4), 479-488.
Winkler, R. L. (1994). Evaluating Probabilities: Asymmetric Scoring Rules. *Management Science*,
40(11), 1395-1405.
Winkler, R. L. (1996). Scoring rules and the evaluation of probabilities. *Test*, 5, 1-60.
Winkler, R. L., & Clemen, R. T. (2004). Multiple experts vs. multiple methods: Combining correlation
assessments. *Decision Analysis*, 1(3), 167-176.
Winters, P. R. (1960). Forecasting Sales by Exponentially Weighted Moving Averages. *Management*
*Science*, 6(3), 324-342.
Wisse, B., van Gosliga, S. P., van Elst, N. P., & Barros, A. I. (2008). Relieving the elicitation burden for
Bayesian belief networks. *Proceedings of the sixth conference on uncertainty in artificial*
*intelligence*.
Witkowski, J., Freeman, R., Wortman Vaughan, J., Pennock, D. M., & Krause, A. (2018). Incentive-
Compatible Forecasting Competitions. *AAAI Conference on Artificial Intelligence*. New Orleans.
Worthen, B. (2003). Future results not guaranteed; contrary to what vendors tell you, computer systems
alone are incapable of producing accurate forecasts. *CIO*, 16(19), 1.
Wright, G., Lawrence, M. J., & Collopy, F. (1996). The role and validity of judgment in forecasting.
*International Journal of Forecasting*, 12, 1-8.

Yaniv, I. (2004). Receiving other people's advice: Influence and benefit. *Organizational Behavior and*
*Human Decision Processes*, 93, 1-13.
- Yaniv, I., & Hogarth, R. M. (1993). Judgmental versus statistical prediction: Information asymmetry and
combination rules. *American Psychological Society*, 4(1), 58-62.
- Yaniv, I., & Kleinberger, E. (2000). Advice Taking in Decision Making: Egocentric Discounting and
Reputation Formation. *Organizational Behavior and Human Decision Processes*, 83(2), 260-281.
- Young, P. C. (2011). *Recursive estimation and time-series analysis: An introduction for the student and*
*practitioner*. Berlin: Springer.
- Yuan, J.-L., & Fine, T. L. (1992). Forecasting Demand for Electric Power. *Advances in Neural*
*Information Processing Systems 5*. Denver.
- Zagorecki, A., & Druzdel, M. J. (2004). An empirical study of probability elicitation under noisy-OR
assumption. *Flairs Conference* (pp. 880-886). Miami: Flairs Conference.

Proposal “A Survey of Human and Machine Forecasting Methods”

People and organizations usually employ human experts and / or apply algorithmic procedures to forecast an uncertain quantity or to determine its distribution. With the wide availability of data and advances in computing technology, algorithmic forecasts offer the opportunity to support human forecasters by mining large datasets and learning patterns and trends from data.

Several survey papers have previously reviewed the literature associated with human forecasting alone. For example, Lawrence, Goodwin, O’Conner, & Önköl (2006) offers a comprehensive view of judgmental forecasting. Clemen and Winkler (1999) also review a variety of human aggregation methods. Other literature focuses on comparing human and machine forecasts. For example, Grove, Zald, Lebow, Snitz, & Nelson (2000); Kuncel, Klieger, Connelly, & Ones (2013); Ægisdóttir, et.al. (2006); and Meehl (1954) review findings comparing human and machine predictions in a clinical setting.

This literature review surveys both human and machine-forecasting methods as well as hybrid (human and machine) forecasting. We evaluate research on human forecasting, such as incentivization, scoring, calibration, and group-forecasting, and discuss qualitative methods. The machine methods in this review include approaches such as regression models and smoothing of time-series, neural and Bayesian networks, ARIMA, and simulation. On the intersection of human and machine forecasting, we discuss research issues such as algorithm aversion, belief updating, and human trust in machine forecasts. Aggregation of forecasts, which applies to human and machine methods, is updated and expanded with more recent methods than previously published in other surveys.

The objective of this review is twofold: (i) first, we survey the machine forecasting and the hybrid forecasting literature together with human forecasting and identify current and future trends. (ii) In the process, we revisit and update the previous literature reviews to include new literature in the field of aggregation and forecasting.

Structural outline

Abstract

1. Introduction
2. Research methodology
 - 2.1. Forecasting fields and methods
 - 2.2. Determination of relevant journals
3. Human forecasting
 - 3.1. Individual human forecasting
 - 3.1.1. Probability elicitation
 - 3.1.2. Incentive systems
 - 3.1.3. Forecaster calibration and training
 - 3.1.4. Scoring rules
 - 3.2. Group forecasting
 - 3.2.1. Delphi method
 - 3.2.2. Focus groups
 - 3.2.3. Nominal group technique
4. Machine forecasting
 - 4.1. Time series methods
 - 4.1.1. Moving average
 - 4.1.2. Exponential smoothing
 - 4.1.3. Fourier time series decomposition
 - 4.1.4. Autoregressive-integrated-moving average
 - 4.2. Similarity-Based/ Correlational models
 - 4.2.1. Regression models
 - 4.2.2. Unobserved components model
 - 4.3. Overarching quantitative models
 - 4.3.1. Neural networks
 - 4.3.2. Bayesian networks
 - 4.4. Simulation Models in Forecasting
5. Human/Machine-Forecasting
 - 5.1. Comparison of forecasting methods and motivation for human/machine forecasting
 - 5.2. Human advice taking and belief updating
 - 5.3. Algorithm aversion

6. Aggregation methods

6.1. Bayesian aggregation methods

6.1.1. Combination of point probabilities

6.1.2. Probability copulas

6.1.3. Normal posterior

6.1.4. Maximum entropy aggregation

6.2. Non-Bayesian aggregation methods

6.2.1. Linear opinion pools

6.2.2. Log-linear opinion pools

6.2.3. Democratic opinion pools

6.3. Discussion of accuracy of different mathematical aggregation methods

7. Summary and conclusion

Appendix B

It is not easy to assess reviews. There was nothing here that I disagreed with. The review is a very high-level one: it covers a vast area of literature very concisely. Some of the topics covered in a paragraph or two could easily be the topics of reviews themselves. Conversely, some topics that are of particular interest to one or more of the authors (e.g., opinion pools) are covered relatively expansively. Below, I provide specific comments that the authors may wish to consider.

1. Some of the key search terms seem rather unconventional. In many specific domains, the contrast is between judgmental forecasting and algorithmic forecasting not between human forecasting (forecasting of humans?) and machine forecasting. Different results would have been obtained if terms in more current use had been used.

2. It is weird indeed that economic forecasting did not appear in the distribution of most frequently searched forecasting topics (page 6). It is likely to be more frequent than, say, load or flood forecasting. It might be worth commenting on this anomaly. Also, some terms are near synonyms (e.g., sales forecasting and demand forecasting; load forecasting and electricity forecasting). Other terms are anomalous because they don't refer to domains (e.g., short-term forecasting).

Also, affective forecasting is not forecasting based on behavioural decision making (page 6). It refers to someone's forecasts of their own emotional reactions to some event (e.g., foot amputation).

3. Under forecasting methods (figure 3, page 6), forecasting from time series is not a method but forecasting from a type of data. This type of data can be analysed using the other methods mentioned here. There are a set of methods for forecasting from time series but time series forecasting is not a method in itself.

4. Under relevant journals (page 6), "Judgmental Forecasting" is not a journal. Other journals that I would have expected to see here (e.g., Technological Forecasting and Social Change) do not appear.

5. On page 10, it is odd to use the term 'human opinion' rather than human judgment. When using judgment to extrapolate from time series, one produces a judgment not an opinion. The term 'opinion' may make sense in certain domains (e.g., geopolitical forecasting) but 'judgment' is more universally appropriate.

6. Under 'Individual Human Forecasting' (page 10), the authors launch straight into probability elicitation. But forecasts in many domains rarely involve probability elicitation (e.g. demand forecasting). There is very brief mention of point forecasts in a sentence lower down this page but, given the large literature on this topic (e.g., most of that reviewed by Lawrence et al (2006)), this seems unbalanced.

7. The sections on incentives, training and scoring rules deal mainly with theoretical developments. Mention of evidence indicating that these developments are effective in improving forecast accuracy is notable by its absence. For example, incentives are effective in about a third of tasks, ineffective in about a third of tasks, and damaging to performance in about a third of tasks (Camerer & Hogarth, 1999; Lerner & Tetlock, 1999). They help only when performance can be improved by greater effort or attention. This might be true in some types of forecasting (e.g., geo-political forecasting) but not others (e.g., judgmental forecasting from time series). Similarly, training with feedback is often ineffective or damaging – as research on multiple-cue probability learning has shown: it can be effective in certain types of judgmental forecasting but not others. (These comments are also relevant to the conclusions about training and incentivization presented on page 40.)

8. Group forecasting and aggregation does not improve performance only because it reduces the impact of bias (page 14): it also, arguably more importantly, cancels out random error.

9. Algorithmic forecasting methods are divided between time domain methods and frequency domain methods. But here you have included frequency domain methods (section 4.1.3) under time series methods (section 4). This is incorrect.

10. Under regression models (section 4.2.1), do you include econometric methods. If not, where do you place them?

11. There are more recent surveys than the Sanders & Manrodt (2003) cited here. For example, Fildes & Petropoulos (2015, Foresight). Generally, use of pure judgment has decreased but use of combined algorithmic/judgmental methods has increased since 2003.

12. On page 30, judgment is held to make algorithmic forecasts worse because people see systematic patterns in noise. But there is little evidence of this. In an empirical study of many thousands of forecast triples (original statistical forecast, final adjusted forecast, outcome) obtained from companies, Fildes et al (2009, IJF) showed that the impairments were partially due to motivated reasoning (optimism) and partly due to small random adjustments made possibly to impose 'ownership' on the forecasts.

13. In section 6.1, can you cite any reports of data that throw light on the relative effectiveness of these aggregation methods?

14. There are many typos that need fixing. They include:

Page 8, line 4: Management Sciences -> Management Science

Page 28, line 19: tese -> these

Page 30, line 9: 2015, (2016 -> 2015; 2016

Page 32, line 7: Wefocus -> We focus

Page 32, line 8: easier -> more easily

Page 34, line 14: by(French, 1985) -> by French (1985)

Page 36, last line: 2013); (Fan -> 2013; Fan

Page 40, line 7: in presence -> in the presence

Page 40, line 21: data is -> data are

Page 44, line 31: expert -> experts

Appendix C

Dear Professor Rowe,

Thank you very much for the feedback we received on our paper

“A Survey of Human and Machine Forecasting Methods”.

The reviewers went to great lengths to provide us their excellent viewpoints on this topic, which helped us to improve the paper. As we discuss in the revised version, the topic covers a broad area with intersections in many fields, including Computer Science, Judgment and Decision Making, Operations Research, Psychology, and each field uses its own preferred terminology and has its own priorities and focus. This is one of the main conclusions of the work, and it may explain why the viewers have also provided (well-received) differing views and emphasis. While working on the revision, our goal was to tell a coherent story that emerges from surveying the different fields, and, to summarize those for researchers in the field.

The following points summarize the main feedback we received and discuss how we addressed them when revising our work.

- *Unclear contribution to the field:* We address this issue by clearly articulating it in the abstract, introduction, and conclusion of our revised paper. To our knowledge, there does not exist a comprehensive and detailed survey spanning the breadth of academic disciplines researching forecasting topics, and this input came from the different domains of the authors of the paper (Computer Science, Operations Research, Judgement and Decision Making, Psychology). By incorporating research from Computer Science, Forecasting, Operations Research, Risk Analysis, and Psychology, we believe that our paper spans these different domains, and summarizes supporting and conflicting findings to improve the understanding of the wide field of forecasting.
- *Insufficient discussion of validity of quantitative models:* The revised paper incorporates the sources detailing the invalidity of some quantitative methods and emphasizes that the use of forecasting axioms could constitute a basis for discussion for all fields involved. One reviewer noted that our previous paper fails to assess the validity of quantitative models for forecasting purposes. The reviewer suggests that whether a model can be considered valid appears to be based on how well it conforms with a set of two forecasting axioms (“Golden Rule of Forecasting”, “Occam’s Razor”). While we think that discussing the proposed axioms is desirable, and we have indeed included a discussion of the axioms, we believe that incorporating this discussion of axioms and labelling how each method fairs with the axioms into the current survey paper would exceed the length and is out

of the scope of the current work of surveying the literature in various fields and identifying the main focus points.

- *Terminology inconsistent with the forecasting domain:* This point is well-taken. As we mentioned, different fields use different terminologies for the same concepts, and one objective of this paper is to bring together the different fields. We rectified this issue by adopting consistent terminology. Machine methods and human forecasting were changed to quantitative methods and human judgment respectively. Furthermore, time and frequency domain methods are now used to describe time-series and correlational methods. We also choose the term combination of forecasts, instead of aggregation, which is commonly found in the Computer Science domain.
- *Missing sources:* The mentioned papers by the reviewers and related sources were incorporated into the revised version where possible. In response, we added numerous additional sources e.g. on the axiomatic validity of quantitative methods, evidence against the benefits of training human judges, the role of incentives, and the readiness of human judges to use quantitative methods by adjusting model results. As we noted, because of the breadth of the various fields, we believe the main contribution of the paper is on providing a coherent overview of methods from the various domains and on bringing the various fields together.

Attached you can find a revised version of the paper. We are grateful to the reviewers for their comments and feedback. They helped us to greatly improve the quality of our survey paper, which we resubmit for publication and further review.

Best regards,

Maximilian Zellner, Ali Abbas, David Budescu, Aram Galstyan

Appendix D

Point-by-Point Response to Reviewers

The authors' replies to the reviewers' comments are marked **red**.

Reviewer 1

The survey confuses atheoretical machine learning methods with validated quantitative methods that use theory and evidence from experiments to specify models. The confusion leads to misleading conclusions.

The survey misses key papers providing evidence from comparative studies on which methods do and do not reduce forecast errors. In particular the recent comprehensive review by Armstrong & Green (2018) "Forecasting methods and principles: Evidence-based checklists" would be a good place to start in improving the survey (using snow-balling perhaps) and for identifying the terms used in the forecasting literature for searches for papers on applications that use evidence-based methods.

Reply:

The revised paper considers the papers suggested by Reviewer 1 by including it in the introduction, conclusion, and criticism of some forecasting methods (Neural Networks, Focus Groups). We also emphasize that the axioms (Golden Rule of Forecasting, Occam's Razor) discussed in the suggested papers can be beneficial when choosing a method or designing a novel approach.

However, we decided not to discuss quantitative methods with respect to how well they conform to these axioms, nor which methods are atheoretical or valid, for the following reasons:

1. Given the extent and content of the paper, we consider discussing models based on the axioms out of its scope. Also, it would have shifted the focus of the paper on quantitative models, which was not our original intention.
2. The purpose of this paper is to provide an overview of work being done in all disciplines involved in forecasting. Labeling methods, which are used often in one domain, as atheoretical and invalid for forecasting, goes against our paper's goal.

Reviewer 2

The forecasting field is vast, in terms of both its range of applications and its techniques. Producing a survey that covers this huge area is therefore challenging and involves difficult judgments on which topics to include and which to emphasize. It also requires a clarity of purpose and careful structuring, enabling key findings to be synthesized and new arguments to emerge. While this paper provides an excellent documentation of how relevant papers were identified and selected for the review, I have a number of concerns about its contribution to the literature. These relate to its structure, its clarity, its coverage, and the extent to which some of its conclusions are novel.

First, I found the paper difficult to navigate given the way it is structured. It might be better to delineate forecasting tasks such as point forecasting based on time series data, point forecasting when contextual data is available, prediction interval formation, density forecasting, event forecasting using probabilities and, for each of these tasks in turn, to identify the strengths and limitations of human, machine and combination forecasting and how each approach is best applied where it is appropriate. For example, this could include methods for improving judgmental forecasts (including possibly decomposition and feedback) in relation to each task. A task-orientated structure is especially appropriate for human judgment because its effectiveness is known to be highly sensitive to the nature of the task. Currently, each section mixes tasks, strengths and weaknesses and improvement strategies so it is difficult to see which strategy might be appropriate for a given task. For example, on page 12 the paper refers to Sanders and Ritzman's finding that training forecasters in gathering and handling contextual data was more beneficial than training them in technical and statistical aspects of forecasting -but their paper only related to point forecasting based on time series data. Would this finding also apply to probability forecasters? Indeed, a task-orientated structure would make it easier to contrast human, machine and combination forecasting in relation to each specific task. In addition, some discussion appears to be in the wrong section. For example, in section 3 on individual human forecasting there is reference at the start of in section 3.3.1 to aggregating opinions. Section 4 on machine forecasting, refers to judgmental time series forecasting -citing

Goodwin and Wright (1993). On page 23 SMAPE and MASE are introduced, but shouldn't they, and similar measures, relate to the earlier discussion of incentive schemes? On page 28 much of the discussion of the findings of Onkal et al. 2009 surely belongs to the subsequent section on algorithm aversion.

Reply:

We received multiple suggestions on how to best organize this work given its extent and decided on a compromise solution. Nevertheless, we agree with the reviewer that organizing the survey paper along forecasting tasks would be very helpful to the reader.

Given the different study designs for both human judgment and quantitative models, it would be difficult to objectively compare their applicability for a given task. This lack of comparability of methods along forecasting task is one of the major research thrusts identified by this survey.

Second much of the paper reads simply like a catalogue of forecasting methods, and the discussion is sometimes too brief for an uninitiated reader to understand what a method involves. I appreciate that brevity is essential in a review with a scope as wide as this one, but it is difficult to discern what the paper's purpose is. Is it intended to introduce non-specialists to forecasting (e.g. the discussion of moving averages and exponential smoothing would suggest this) or to update specialists –who would already know about moving averages etc. -on the latest findings? The current treatment falls between these two stools.

Reply:

The paper is a survey, not a book or an introduction to the variety of methods. Therefore, it is aimed at beginners and specialists alike because it provides a holistic view of the forecasting domain. The main contribution of this paper is to bring the disparate disciplines together, and to establish a common basis for future work.

There are also several areas where the discussion is unclear or contradictory. What is strategic behaviour in relation to probability forecasting? And why would an advice seeker benefit from an incentive scheme that rewards strategic behaviour rather than truthful reporting (page 11 -

unless this is a typo). On page 28, line 44 the sentences relating to Onkal et al. (2009) are confusing. On page 31 we are told that “ Opinions and advice originating from highly correlated sources are unlikely to improve forecasting accuracy”, but then “as long as correlation between forecasters is not perfectly positive, adding more forecasters increases forecasting accuracy”. On page 31 , you state that you have excluded behavioral aggregation from your survey, but surely this has already been included in section 3.2?

Reply:

In this particular case, strategic behavior refers to the readiness of an informed agent to reveal his or her information to a less-informed principal. This mechanism design concept was explained in more detail in the revised paper.

If multiple human experts compete under an incentive scheme, they tend to behave strategically in that they emphasize that is only available to them when forming their judgment. As result, the variance of the combined judgment is higher, reducing bias due to overemphasis on joint information.

It is of course easy when reviewing a paper that is as ambitious in its scope as this paper to argue that other works and topics should be included. However, given the importance of weather forecasting, I think that ensemble forecasting should be discussed. On the integration of machine and human forecasting you could consider correction methods (e.g. Theil’s method -see Ahlburg, 1984) where a machine forecasts the errors of a human forecaster and then corrects their forecasts accordingly. I would also like to see some discussion of whether it is better to allow a human to adjust a machine forecast or simply to aggregate human and machine forecasts mechanically (e.g. by taking a simple average). Recent work on judgmental selection of machine methods (Petropoulos et al. 2018, De Baets and Harvey 2020) might also be worth including, though I appreciate the dates of your literature search precluded the selection of these papers.

Reply:

The papers listed have been added to the review. For the sake of brevity, we omitted the specific case of weather forecasting and ensemble methods.

Finally, I have doubts about the novelty of the paper's findings that (1) neither human or machine forecasting is universally superior, and (2) the better method varies as a function of factors such as availability, quality, extent, and format of data, suggesting that (3) the two approaches can complement each other to yield more accurate and resilient models. (1) and (2) are self-evident and (3) was highlighted as early as 1990 in the paper by Blattberg and Hoch.

Reply:

These are not the major contributions of the paper. We aimed to provide an overview and to bring the different disciplines together. The revised paper discusses these contributions in detail. The revised paper clarifies and emphasizes this point.

Minor points

Do figures 3 and 4 include overlapping categories?

Reply:

Yes, the possibility of overlapping publications exists because papers might be tagged with multiple terms. We discuss the possibility of overlap in the paper.

Is there are journal called Judgmental Forecasting as Table II suggests or is this perhaps the edited book by Wright and Ayton?

Reply:

It was the edited book by Wright and Ayton. To avoid confusion, we decided to erase it from this table.

Section 4.2.1 Are regression models always very simple?

Reply:

We chose the term "simple regression" to refer to linear regression. This terminology is consistent with the literature and does not constitute a judgment whether it is a simple model to construct, use, or validate

Page 39, line 43 Are machine models always incapable of using contextual data?

Reply:

We adjusted the terminology of the survey, changing machine models to quantitative models.

This changes the statement mentioned in your comment.

You are correct in pointing out that machine models can include contextual data if one assumes that simulation falls under machine models for example. Using a simulation, one can model causal structures that might not be evident from the data and depend on the context the model is being used in. A purely quantitative model, such a linear regression, only shows a potential association though, requiring human judgment about causation. The human judgment in turn depends on the understanding of the context.

There are lots of typos in the manuscript. Please proof read it.

Reply:

Paper was proof-read and checked for grammatical errors.

References

Ahlburg, D. A. (1984). Forecasting evaluation and improvement using Theil's decomposition. *Journal of Forecasting*, 3, 345– 351.

De Baets, S., & Harvey, N. (2020). Using judgment to select and adjust forecasts from statistical models. *European Journal of Operational Research*.

Petropoulos, F., Kourentzes, N., Nikolopoulos, K., & Siemsen, E. (2018). Judgmental selection of forecasting models. *Journal of Operations Management*, 60, 34-46.

Reviewer 3

It is not easy to assess reviews. There was nothing here that I disagreed with. The review is a very high-level one: it covers a vast area of literature very concisely. Some of the topics covered in a paragraph or two could easily be the topics of reviews themselves. Conversely, some topics

that are of particular interest to one or more of the authors (e.g., opinion pools) are covered relatively expansively. Below, I provide specific comments that the authors may wish to consider.

1. Some of the key search terms seem rather unconventional. In many specific domains, the contrast is between judgmental forecasting and algorithmic forecasting not between human forecasting (forecasting of humans?) and machine forecasting. Different results would have been obtained if terms in more current use had been used.

Reply:

Following the change of terminology, we conducted another search with expanded key terms that include judgmental, algorithmic, and quantitative forecasting. Expanding the key terms, produced several publications that we integrated into the pre-existing work. Given the breadth of the field, we only expanded the key terms by these three, for the sake of brevity.

2. It is weird indeed that economic forecasting did not appear in the distribution of most frequently searched forecasting topics (page 6). It is likely to be more frequent than, say, load or flood forecasting. It might be worth commenting on this anomaly. Also, some terms are near synonyms (e.g., sales forecasting and demand forecasting; load forecasting and electricity forecasting). Other terms are anomalous because they don't refer to domains (e.g., short-term forecasting). Also, affective forecasting is not forecasting based on behavioural decision making (page 6). It refers to someone's forecasts of their own emotional reactions to some event (e.g., foot amputation).

Reply:

1. We agree that one could expect economic forecasting to be of more interest than load or flood forecasting. The prominent forecasting domains were initially obtained by analyzing the number of publications under the related search terms provided by Google Scholar. Given that economic forecasting appears most frequently when searching only for the term "forecasting" we conducted a separate search and appended the results to the

initial findings. However, because economic forecasting is related to forecasting demand, the categories are not mutually exclusive (which we point out in the paper).

2. We removed the incorrect explanation of affective forecasting.

3. Under forecasting methods (figure 3, page 6), forecasting from time series is not a method but forecasting from a type of data. This type of data can be analysed using the other methods mentioned here. There are a set of methods for forecasting from time series but time series forecasting is not a method in itself.

Reply:

This issue was addressed by adopting the terminology used by the reviewer.

4. Under relevant journals (page 6), “Judgmental Forecasting” is not a journal. Other journals that I would have expected to see here (e.g., Technological Forecasting and Social Change) do not appear.

Reply:

These journals were included in the survey paper. We want to point out that because of the initial algorithmic search and then snow-balling from identified publications, there might be outlets that were omitted. Additionally, it is possible that we cited papers in the survey and did not mention the journal name in this table. For example, if a journal published one or only a small number of papers that we considered important for this survey, we did not include it in this table.

5. On page 10, it is odd to use the term ‘human opinion’ rather than human judgment. When using judgment to extrapolate from time series, one produces a judgment not an opinion. The term ‘opinion’ may make sense in certain domains (e.g., geopolitical forecasting) but ‘judgment’ is more universally appropriate.

Reply:

Adjusted terminology where we saw fit.

6. Under ‘Individual Human Forecasting’ (page 10), the authors launch straight into probability elicitation. But forecasts in many domains rarely involve probability elicitation (e.g. demand forecasting). There is very brief mention of point forecasts in a sentence lower down this page but, given the large literature on this topic (e.g., most of that reviewed by Lawrence et al (2006)), this seems unbalanced.

Reply:

We added a few more details and references on point forecasts in human judgment.

7. The sections on incentives, training and scoring rules deal mainly with theoretical developments. Mention of evidence indicating that these developments are effective in improving forecast accuracy is notable by its absence. For example, incentives are effective in about a third of tasks, ineffective in about a third of tasks, and damaging to performance in about a third of tasks (Camerer & Hogarth, 1999; Lerner & Tetlock, 1999). They help only when performance can be improved by greater effort or attention. This might be true in some types of forecasting (e.g., geo-political forecasting) but not others (e.g., judgmental forecasting from time series). Similarly, training with feedback is often ineffective or damaging – as research on multiple-cue probability learning has shown: it can be effective in certain types of judgmental forecasting but not others. (These comments are also relevant to the conclusions about training and incentivization presented on page 40.)

Reply:

The relevant sources outlined by the reviewer were added and their results incorporated in the relevant sections.

8. Group forecasting and aggregation does not improve performance only because it reduces the impact of bias (page 14): it also, arguably more importantly, cancels out random error.

Reply:

This insight has been added to the section.

9. Algorithmic forecasting methods are divided between time domain methods and frequency domain methods. But here you have included frequency domain methods (section 4.1.3) under time series methods (section 4). This is incorrect.

Reply:

We agree with the reviewer and moved the section on Fourier Time Series decomposition to the section on frequency domain methods.

10. Under regression models (section 4.2.1), do you include econometric methods. If not, where do you place them?

Reply:

The most common econometric methods comprise methods such as linear regression, generalized linear models, and ARIMA that we cover in the review. Also, Bayesian Networks can be used to construct an econometric model by modeling the causal structure and implementing the respective probability distributions. While it is important to highlight how these models can be applied in the domain of forecasting economic quantities, it would make the paper even longer than it already is. We decided to mention the use of these methods in economic forecasting but omitted a discussion of their application because of brevity.

11. There are more recent surveys than the Sanders & Manrodt (2003) cited here. For example, Fildes & Petropoulos (2015, Foresight). Generally, use of pure judgment has decreased but use of combined algorithmic/judgmental methods has increased since 2003.

Reply:

The relevant sources outlined by the reviewer were added and their results incorporated in the relevant sections.

12. On page 30, judgment is held to make algorithmic forecasts worse because people see systematic patterns in noise. But there is little evidence of this. In an empirical study of many

thousands of forecast triples (original statistical forecast, final adjusted forecast, outcome) obtained from companies, Fildes et al (2009, IJF) showed that the impairments were partially due to motivated reasoning (optimism) and partly due to small random adjustments made possibly to impose 'ownership' on the forecasts.

Reply:

The relevant sources outlined by the reviewer were added and their results incorporated in the relevant sections.

13. In section 6.1, can you cite any reports of data that throw light on the relative effectiveness of these aggregation methods?

Reply:

There is conflicting evidence of the effectiveness of Bayesian aggregation methods compared to more traditional approaches. We omitted the discussion of their relative effectiveness to keep the survey at a reasonable length, focusing on each method's compatibility with Bayesian statistics.

14. There are many typos that need fixing. They include:

Page 8, line 4: Management Sciences -> Management Science

Page 28, line 19: tese -> these

Page 30, line 9: 2015, (2016 -> 2015; 2016

Page 32, line 7: Wefocus -> We focus

Page 32, line 8: easier -> more easily

Page 34, line 14: by(French, 1985) -> by French (1985)

Page 36, last line: 2013); (Fan -> 2013; Fan

Page 40, line 7: in presence -> in the presence

Page 40, line 21: data is -> data are

Page 44, line 31: expert -> experts

Reply:

Typos and grammatical mistakes were fixed.

Reviewer 4:

This paper presents a critical review of extant work on human, machine, and hybrid forecasting, as well as forecast aggregation. It fits with the journal's scope on reviews of multidisciplinary topics as it provides an extensive summary of the state-of-the-art in forecasting, explores developments in this field and points to promising research avenues/thrusts.

Overall, I believe that the paper provides a detailed review of an important multidisciplinary area with strong implications across a wide variety of domains. It is a very long review, as it aims to be all-encompassing in its scope. Although the length of the paper may not be problematic from the journal's perspective, it may be distracting from the reader's perspective, which may be worth considering. Moving certain tangential subsections into Appendices or use of footnotes could be alternatives, but there could be others.

My comments are as follows:

1. As this is a review of forecasting, I would urge the authors to only cite the work that actually requires making forecasts, instead of involving estimation tasks. As has been shown in repeated studies, people's responses to general knowledge tasks, for example, cannot be generalized into the forecasting domain. This is relevant for probability elicitation section in the paper, as well as the sections on Delphi and advice taking, among potentially others.

Reply:

We tried to make this distinction, to the extent possible.

2. What would set this paper apart would be the final section. In order for this not to feel like a literature review for a dissertation work, it would be fundamental to include (i) a critical overarching evaluation across the findings, and (ii) further venues for promising research. The paper does this to a limited extent and it would be highly valuable to expand on this final section (e.g., to include practical implications across sectors).

Reply:

We re-emphasized the contribution, which is to bring the disparate fields together by offering a comprehensive overview of the work being done by each. The survey shows the disparities between fields, including terminology and methodology, and argues that the forecasting domain can benefit from collaboration. This is emphasized in the abstract, introduction, and conclusion of the revised paper.

Reviewer 5:

This paper reviews forecasting methods, including both human and machine based methods. The relatively wide scope of the review differentiates it from other reviews that consider only human methods or machine methods. Overall, I find the review to be well done. I particularly appreciate the charts that illustrate the growing importance of this field of study and the attention paid to Bayesian methods. I recommend the paper be accepted with minor revision.

Although the paper is generally well written, there are some instances in which past tense is used inappropriately. For example, in the abstract, “The survey started with...” should be revised to, “The survey starts with...” I would suggest the authors take a final look at the manuscript and make minor grammatical and compositional improvements as needed.

Response to Reviewer 5:

We performed a thorough revision of the paper to improve grammar and to eliminate typos.

Appendix E

Second Revision – Reply to Reviewers

We deeply appreciate the time and effort of the reviewers to provide us with such detailed feedback. Our responses to their comments are in **RED**.

Reviewer: 1

Comments to the Author(s)

Might the paper be better titled something along the lines of "A survey of trends in publishing on forecasting methods, by application and method name "

We considered your suggestion carefully and concluded that while the paper mentions trends in forecasting, it also surveys the methods, their shortcomings, performance, etc. As a result we decided to leave the title as is, but if you feel strongly about having it changed we will accommodate your suggestion.

Reviewer: 2

Comments to the Author(s)

This revised paper is improved in some respects -and again I welcome the details of the research methodology. The emphasis on combining forecasts is also appropriate. However, I still have serious concerns about some aspects of the paper's structure and the relative emphasis it places on several forecasting topics, in addition to other issues.

A major structural concern is the placement of subsection 4.1.1 'Regression models' within the section on Frequency Domain models. Frequency domain models are generally used to account for variation in time series through cyclic components at different frequencies. Hence the input variables are typically cosines and sines, but there is no reference to this in the paper. Although least squares estimation can be used to obtain these models, the reader might be led to believe that all regression models are frequency domain, which is very far from the case. A distinction between univariate forecasting methods and explanatory methods -which draw on information from independent variables -would be more appropriate and illuminating than that of frequency versus time domain. I strongly disagree with the statement in the Conclusions (page 42, line 18) that the most common classification distinguishes between time and frequency domain models, and a combination of the two. I certainly cannot understand why logistic

regression models appears in the frequency domain section. Again, reading the paper the reader may infer that these models are based on least squares when they are obtained through maximum likelihood estimation. Section 4 would certainly benefit from an introductory statement of what the frequency domain is rather than the vague statement that models based on it “use highly refined and specific information about relationships between system elements...”

- Adjusted the structure to univariate (time-series and frequency domain methods), explanatory (regression, support vector machines), and overarching methods. Updated the conclusion with this terminology
- Clarified the term frequency models and provided a clearer definition

Another structural oddity is the inclusion of the following sentence in section 4 ‘Quantitative Forecasting Methods’ : “Judgmental time-series models concern humans extrapolating time-series into the future and adjusting the series for contextual data (page 19, line 20). Manifestly, judgmental forecasts are not derived via quantitative methods so why is this sentence placed here? Quantitative models can be applied to judgmental forecasts, for example by using (psychological) bootstrap models, but oddly these are not mentioned at all in the paper. Incidentally, people don’t usually adjust the series, they adjust forecasts.

We agree with the comments. Removed the judgmental time-series models from this section.

I am also not clear why naïve forecasts are included under simulation models (page 27).

We agree with the comments. Moved naïve forecasts to univariate methods -> time-series methods.

In terms of emphasis - there is no mention of prediction intervals -a major way of expressing forecasts and linear regression models -a widely used forecasting method (even discounting econometric applications) merits less than four lines. Yet there is a whole sub-section on focus groups. Focus groups are not a forecasting method -they are designed ‘to explore the dimensions of a topic and the range of conceivable responses rather than achieving a consensus’ (see Ord et al, 2017, page 393).

A brief discussion on prediction intervals and relevant sources was added under the section explanatory methods -> regression methods

There still tends to be a merging of the discussion of different forecasting tasks which gives the paper a shapeless feel. Several parts of the paper would benefit if the forecasting task that was being referred to was made clearer. On page 34 there is a discussion of forecasts of 'point probabilities'. By point probabilities do you mean probabilities for discrete quantities or events (e.g. the probability that it will rain tomorrow) as opposed to estimates of continuous probability distributions?

Clarified and defined point probabilities.

Overall, I think the paper would benefit from a tree diagram early on which maps out the different forecasting methods and provides the paper with a clear structure. A clear definition of the task that is being discussed at the start of each section would also be helpful. I found the conclusions to be unexciting and they ignore a major aspect of recent research -the need to develop methods to support judgmental forecasters, such as decomposition, guidance and the identification and use of analogies.

Inserted tree representation of the paper in the introduction. Added the recommended major aspect of recent research to the conclusion.

Other points

Please tell the reader that the golden rule of forecasting is on page 1

Added

Page 15, bottom. What do you mean by: "Because surveying the multitude of group judgment is not the focus of this review paper.."

Removed this section to avoid confusion

Page 28, line 40. Spelling is: O'Connor.

Corrected

Page 33, line 7. You surely don't mean the opposite result to that found by Ahlburg?

Corrected

Page 34, line 33. forecasters (i.e. plural)

Corrected

Reference

Ord K., Fildes, R. and Kourentzes, N. (2017) Principles of Forecasting 2e, New York: Wessex.

Reviewer: 3

Comments to the Author(s)

The authors' responses have dealt with the points made by the five reviewers. In my view, there are just a few minor issues that remain to be addressed.

1) Additional panels have been added to Figures 2 and 4. I think that this is because the topics have been divided into time-domain and frequency domain searches. But this is not clear. Two things need to be done. First, add to the figure captions to explain what the upper and lower panels of the figures represent. Second, ensure that the axes in the two panels look the same. For instance, in Figure 2, the numbers on the vertical axis are in different sized typefaces, the labels on those axes are bold in one case and not the other, and the axes have a different range of values in the two cases. In Figure 4, the divisions on the vertical axes are different: 50,000 in the upper panel and 100,000 in the lower one.

Reworked the graphical representations and changed the y-axis to be of the same magnitude.

2) In the last paragraph on page 14, point forecasts and pdf forecasts are mentioned. It might be worth adding that interval forecasts (without point forecasts) are also not uncommon, especially in economics.

Mentioned interval forecasts

3) At the bottom of page 14, we are told that studies have found no "clear evidence supporting representing data visually instead of in table format representation when eliciting point forecasts". This statement is misleading. For example, Harvey & Bolger (1996) did find clear evidence that graphical

presentation is superior when data contain trends (most data sets). (There is also mounting evidence that the type of graphical format matters - e.g., Okan et al, QJEP, 2018 – but that does not need to be mentioned here.)

Clarified this sentence

4) Page 31, line 6: intractable -> opaque

Corrected

5) Page 31, line 12: impressive but it -> impressive, it

Corrected

Appendix F

Point-by-Point Response to Reviewers

The authors' replies to the reviewers' comments are marked **red**.

Reviewer: 3

The authors have successfully addressed the issues that I raised (as Referee 3) in my review of their first revision. I do think that they have also gone some way to dealing with the points raised by Referee 2 but I was not convinced that their responses to will fully satisfy that referee (e.g., on focus groups). However, it is up to him/her to make that decision.

One point that could be dealt with later in the publication process (but might be better to address now) is that the new list of references excludes some papers that were in the original reference list and are still cited in the text.

References have been updated to reflect all sources used in the manuscript.

Reviewer: 1

p.2: Suggest revising along the lines of... "Critics of this view point out that the use of machine learning or "big data" methods—such as stepwise regression and neural nets—that use statistical procedures to discover apparent patterns in data without recourse to theory and prior knowledge are akin to alchemy (see, e.g., Einhorn, 1972)."

Roger Penrose is also sceptical on the possibility of "AI" (Shadows of the Mind).

Wording has been adjusted according to suggestion by reviewer.

p.25: The relevant section should mention Gardner's conclusions re the improvements in accuracy provided by "damped trend" exponential smoothing models.

The reasons for why exponential smoothing with damped trend, both multiplicative and additive seasonality, offers an improvement of forecasting accuracy have been added. Including the papers of Gardner & McKenzie (1989) and Fildes (2001).

p.27: There is no single "naïve approach". See Green & Armstrong re the evidence on simple (often could be characterized as "naïve") vs complex methods.

Inserted a paragraph emphasizing the distinction between "simple" and "naïve", explaining that the naïve approach according to our definition is simple but that not every simple forecasting process is naïve.

p. 52: "not grounded in statistical theory". Rather than "statistical theory" should be something along the lines of "theory and prior knowledge on cause and effect".

Adjusted wording according to suggestions made by the reviewer.

p. 53: You mention analogies in the context of “ripe for future research”, but this is not mentioned in the body text and the Green and Armstrong paper on “structured analogies” in the references is not cited in the text.

Added a brief paragraph of structured analogies to the section of human judgment – forecaster calibration and training, describing the term and the accuracy improvement it entails.